# Neuron-based Multifractal Analysis of Neuron Interaction Dynamics in Large Models

**Xiongye Xiao**[1*†]    **Heng Ping**[1†]    **Chenyu Zhou**[1]    **Defu Cao**[1]    **Yaxing Li**[1]

**Yi-Zhuo Zhou**[2]    **Shixuan Li**[1]    **Nikos Kanakaris**[1]    **Paul Bogdan**[1*]

[1]University of Southern California, CA, USA
[2]University of California, Riverside, CA, USA

## Abstract

In recent years, there has been increasing attention on the capabilities of large-scale models, particularly in handling complex tasks that small-scale models are unable to perform. Notably, large language models (LLMs) have demonstrated "intelligent" abilities such as complex reasoning and abstract language comprehension, reflecting cognitive-like behaviors. However, current research on emergent abilities in large models predominantly focuses on the relationship between model performance and size, leaving a significant gap in the systematic quantitative analysis of the internal structures and mechanisms driving these emergent abilities. Drawing inspiration from neuroscience research on brain network structure and self-organization, we propose (i) a general network representation of large models, (ii) a new analytical framework—*Neuron-based Multifractal Analysis (NeuroMFA)*-for structural analysis, and (iii) a novel structure-based metric as a proxy for emergent abilities of large models. By linking structural features to the capabilities of large models, *NeuroMFA* provides a quantitative framework for analyzing emergent phenomena in large models. Our experiments show that the proposed method yields a comprehensive measure of the network's evolving heterogeneity and organization, offering theoretical foundations and a new perspective for investigating emergent abilities in large models.

## 1 Introduction

The advent of foundation models triggered a paradigm shift in artificial intelligence (AI), demonstrating unparalleled capabilities across many domains, particularly natural language processing (Chang et al., 2024; Brown et al., 2020; Chowdhery et al., 2023; He et al., 2020; Duan et al., 2024). Recent studies observed that, compared to traditional small-scale neural networks (NNs), large language models (LLMs) exhibit advanced cognitive and reasoning abilities and a higher level of comprehension (Brown et al., 2020; Bubeck et al., 2023; Achiam et al., 2023). This behavior, where larger models develop qualitatively new abilities absent in smaller ones, is referred to as "emergent abilities" (Srivastava et al., 2022; Wei et al., 2022a). Recent works have focused on using scaling laws and other analytical methods to study the relationship between model size and the development of these new capabilities, suggesting that emergent abilities may arise predictably as models scale up (Kaplan et al., 2020; Hu et al., 2023a;b; Michaud et al., 2023; Schaeffer et al., 2024).

**Existing research, limitations, and motivation.** As depicted in Fig. 1, previous research on the "emergence" in large models and particularly LLMs (Wei et al., 2022a; Fu et al., 2023; Zhong et al., 2024; Schaeffer et al., 2024) focuses on the observation-based analysis and the model size (e.g., number of parameters, model depth). Traditional metrics (Paperno et al., 2016; Sakaguchi et al., 2021; Bisk et al., 2020) have largely assessed LLMs' emergent abilities by capturing the relationship between model's size and output performance. However, these approachs have several limitations: (i) observation-based methods often require exhaustively testing various outputs, making it challenging

---

*Correspondence: Xiongye Xiao <xiongyex@usc.edu>, Paul Bogdan <pbogdan@usc.edu>. †Equal Contribution.

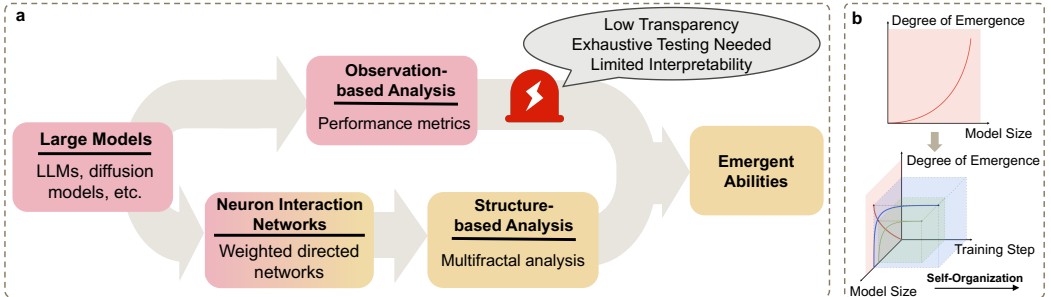

Figure 1: **(a)** The proposed paradigm shift for assessing the degree of emergence in large models. **(b)** Comparison of traditional methods (top) and our approach (down); we study the emergent abilities of large models by analyzing structural dynamics during training.

to establish a unified standard for evaluating the full scope of a large model's emergent abilities; (ii) the approaches treat the model as a black box, limiting the understanding of the internal mechanisms that drive the model's behavior and emergent abilities; (iii) relying solely on model size to measure performance overlooks other critical factors that significantly influence the model's capabilities, such as architecture design, data quality, and training dynamics. To address these limitations, our approach emphasizes structure-based analysis by examining the dynamic changes in a model's internal structure during the training process. The iterative training process is crucial for the improvement and development of capabilities in large models, as it enables the model to transition from having no abilities to exhibiting emergent abilities, as illustrated in Fig. 2. Current work has demonstrated that large models undergo significant structural changes in their internal mechanisms and dynamic characteristics throughout training (Ding et al., 2023). Moreover, inspired by neuroscience research on how the brain's network microstructure supports its functions (Bullmore & Sporns, 2009; Bassett & Sporns, 2017), we aim to develop a similar framework for the structural analysis of large models. This framework seeks to link the internal architecture and interaction patterns of large models with their emergent capabilities, providing deeper insights into the mechanisms that drive their complex behaviors. By adopting a structure-based analysis approach, we can move beyond treating models as black boxes and gain a deeper understanding of how internal structural dynamics connect to the development of advanced functionalities in large models.

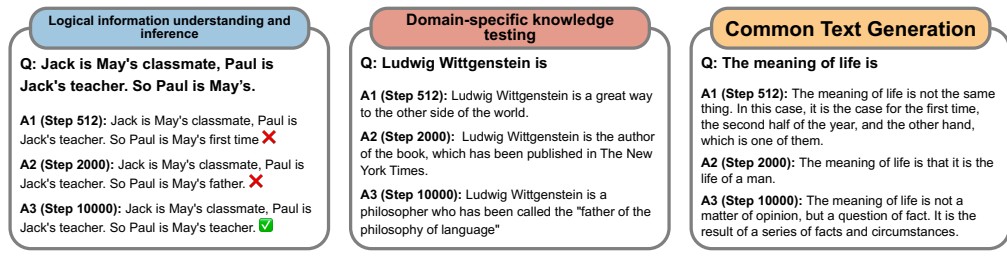

Figure 2: Improvement of Pythia responses with increasing training steps, as indicated by the progression from incoherent to coherent answers across diverse categories (blue for questions with standard answers, red for semi-open-ended questions, and yellow for open-ended questions).

**Our approach and its connection to neuroscience.** To address the above limitations, our research bridges the existing gap by analyzing the internal network structures of large models and linking the structure-based properties to the models' emergent capabilities. In the field of neuroscience, extensive research has established that the brain's network microstructure is fundamental to its functionality, with model organisms such as Drosophila and mice serving as key examples (Mizutani et al., 2013; Shih et al., 2020; Xiao et al., 2021; Yin et al., 2020; Coletta et al., 2020). These studies emphasize the critical role of network structure in supporting and determining functional capabilities, inspiring us to explore the relationship between the structure and the emergent abilities of large artificial models. In the study of human intelligence, the brain is considered a self-organizing complex system (Kelso, 1995; Singer, 1986; Pribram, 2018; Antonello et al., 2022). The manifestation of intelligence and human behavior—from neurons to mind—is governed by the generic process of self-organization. A key aspect of studying intelligence involves quantitatively analyzing and investigating the complex interactions among numerous neurons (Sporns et al., 2004; Bassett & Sporns, 2017). Self-organization

phenomena, widely observed in natural systems, involve numerous micro-level interactions leading to complex macro-level behaviors, and the emergence can be understood by modeling and analyzing the self-organizing micro-structure (Turing, 1990) (see Appendix A.3). Inspired by this, we aim to explore whether the emergent behaviors exhibited by large models can be understood and connected through structural self-organization. By abstracting large models as network representations, we can analyze their interaction structures and correlate them with capabilities. Drawing an analogy to structural self-organization, we propose relevant structure-based metrics that enable us to quantify the *degree of emergence* (this is a structure-based metric that can be viewed as a proxy for the emergent abilities of large models) in large models.

**Our contributions.** In summary, the contributions of our work are three-fold: (i) We propose a **unified network representation** for large models, where large models are represented as *Neuron Interaction Networks (NINs)*, enabling the structural analysis of artificial neuron interactions. (ii) We introduce *neuron-based multifractal analysis (NeuroMFA)*, a framework that quantifies the **regularity and heterogeneity** of NINs by statistically analyzing the fractal properties of neuron interactions. This framework measures the neuron interaction dynamics at the micro-level and it is the first flexible framework that seamlessly integrates techniques for self-organization, fractal, and structural analysis to estimate the *degree of emergence* of large models. (iii) We propose a novel metric based on NeuroMFA to quantify the **degree of structural emergence** in large models by analyzing the interaction dynamics of neurons during the training process, opening a new direction for the study of "emergence" in large models.

## 2 RELATED WORK

The concept of emergent ability has gained prominence in LLM research, initially through the 'chain-of-thought' technique for complex mathematical problem-solving (Wei et al., 2022b). The interpretability and quantification of emergent ability were advanced with the PASSUNTIL evaluation strategy (Fu et al., 2023), utilizing massive sampling for high-resolution analysis (Hu et al., 2023b). Michaud et al. (2023) proposed a quantization model to elucidate neural scaling laws, addressing both the power-law decrease in loss with increasing model and dataset sizes, and the development of new abilities as models scale up. However, Schaeffer et al. (2024)questioned the notion of emergent abilities in LLMs, arguing that the appearance of these abilities may be more a consequence of the evaluation metrics chosen, rather than fundamental changes in the model's behavior. This critique has sparked further inquiry into how evaluation strategies influence our understanding of emergent abilities in large models.

Following this, Chen et al. (2024) proposed a quantitative approach to measuring "emergence" by comparing entropy reductions at semantic and token levels. Gurnee et al. (2024) explored the concept of "universal neurons" in GPT-2 models, discovering that a small subset of neurons exhibits consistent behavior across models trained with different random seeds, suggesting a neuron-level mechanism that may drive emergent phenomena. Most recently, Du et al. (2024) introduced a new perspective, emphasizing pre-training loss as a critical driver of emergent ability in large models. Their study showed that models with lower pre-training losses exhibit emergent abilities, independent of model size or dataset scale.

## 3 METHODOLOGY

In this section, we introduce the motivation behind our approach and outline its three main components. Briefly, we first represent a large model as a neuron interaction network. Then, we propose the neuron-based multifractal analysis (NeuroMFA). Finally, based on NeuroMFA, we propose a new metric as a proxy for the emergent abilities of large models by quantifying and analyzing the dynamic changes in their structure.

### 3.1 MOTIVATION: CONNECTING SELF-ORGANIZATION TO EMERGENCE

Self-organization is a foundational concept for understanding emergent phenomena in natural systems (Goldstein, 1999; Crutchfield, 1994). It describes how systems gradually evolve through local interactions to develop more diverse interaction patterns and achieve overall order, ultimately

leading to emergence (Correia, 2006; Ay et al., 2007). This process involves two key aspects: the development of new interaction patterns (reflecting increased *heterogeneity* in interactions) and the formation of orderly structures (representing enhanced *regularity* in global structure). Emergence, in this context, refers to the novel macro-level properties formed during the process of self-organization—characteristics that arise through collective interactions and cannot be fully explained by the dynamics of individual components. Inspired by these principles, we extend the concept of self-organization to study the degree of emergence in large artificial models (please refer to Appendix A.3.3) and design network-based metrics to quantify the two key aspects of self-organization: *heterogeneity* (capturing the diversity of neuron interactions) and *regularity* (reflecting the order in global network structure). By introducing neuron-based multifractal analysis (NeuroMFA), we provide a unified framework to investigate these dimensions, enabling a quantitative analysis of the relationship between structure and emergent capabilities in LLMs.

## 3.2 NEURON INTERACTION NETWORK

To illustrate our approach, we begin by defining the structure of a neural network (NN) within large models. A large model is usually composed of multiple NN layers, represented as $L_1, L_2, \cdots, L_k$, where $k \in \mathbb{N}^+$ denotes the number of layers. Within each layer $L_j$, the artificial neurons are specified as $n_1^j, n_2^j, \cdots, n_{p_j}^j$, where $p_j \in \mathbb{N}^+$ denotes the number of neurons in layer $L_j$. Neurons between layers are interconnected by weights $w_{ab} \in W$, where $a$ and $b$ denote the indices of the neurons. For instance, $a$ could represent $n_1^j$ and $b$ could represent $n_1^{j+1}$. Notably, neurons in one layer connect only to neurons in the subsequent layer.

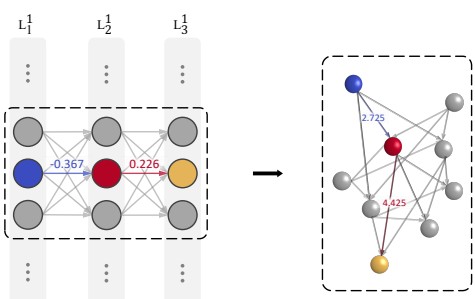

Figure 3: The transformation from a partial neural network (NN) in large models into a neuron interaction network (NIN).

The structure of an NN contains key information about its architecture and neuron connections. However, applying network and structure analysis to NNs presents challenge: (i) the presence of negative weights can create inconsistencies in distance-based metrics, complicating the interpretation of interactions; (ii) defining measures of distance and shortest path is crucial for studying network structures, where stronger connections typically imply shorter distances (Newman, 2001). To address these issues, we first transform NNs into the Neuron Interaction Networks (NINs), defined as follows.

**Definition 3.1** (Neuron Interaction Network (NIN)). NIN is a weighted directed graph $G = (V, E)$, where:

[1] **Node set:** $V$ denotes the set of neurons (nodes) in the NIN, directly inherited from the NN;

[2] **Edge set:** $E \subseteq V \times V$ represents the directed edges between nodes, where each edge $e_{a,b} \in E$ connects nodes $a$ and $b$. Each edge has a non-negative weight $\omega_{ab} = |w_{ab}|^p$, where $w_{ab}$ is the original weight in the NN, $|w_{ab}|$ reflects the interaction strength between neurons, and $p = -1$ is a parameter that inversely maps the original weight to the edge distance (Newman, 2001; Liu et al., 2017).

[3] **Shortest path distance:** the distance $d_{ij} \in \mathbb{R}_0^+$ between two nodes $i$ and $j$ is defined as the shortest path distance, calculated as the minimum sum of edge weights over all possible paths between the nodes. A path $P(i,j) = (v_i, v_1, v_2, \cdots, v_j)$ consists of a valid sequence of neurons and edges connecting $i$ to $j$, spanning multiple layers. The formal expression for the distance is given by:

$$d_{ij} = \min_{P(i,j)} \left( \sum_{(u,v) \in P(i,j)} \omega_{uv} \right) + \lambda \cdot |P(i,j)|^{\gamma}. \tag{1}$$

The term $|P(i,j)|$ is defined as the number of edges in the path $P(i,j)$ connecting nodes $i$ and $j$, representing the length of the shortest path. This distance metric accumulates the edge weights $\omega_{uv}$ along each valid path and incorporates a weighted term based on the path length. The parameters $\lambda \in \mathbb{R}$ and $\gamma \in \mathbb{R}$ balance the influence of edge weights and path lengths, respectively, allowing for a

comprehensive measure of distance that accounts for both interaction strengths and the complexity of the connectivity paths.

[4] **Neighbors** $\mathcal{N}(v_i)$**:** To facilitate the proposed analysis of neuron-based local interaction patterns in the NIN, we define the neighbors of a neuron $v_i \in V$ as the set of neurons whose shortest path distance from $v_i$ is less than or equal to a predefined threshold distance $d_{\text{threshold}}$. Formally, the neighbor set $\mathcal{N}(v_i)$ is defined as:

$$\mathcal{N}(v_i) = \{v_j \in V \mid d_{ij} \leq d_{\text{threshold}}\}, \tag{2}$$

where $d_{ij}$ denotes the shortest path distance between neurons $v_i$ and $v_j$. This definition identifies all neurons that are sufficiently close to $v_i$ in terms of network proximity, enabling us to efficiently and systematically explore local node-based interaction dynamics.

In the subsequent NeuroMFA framework, these neighborhood relationships are leveraged to conduct multifractal analysis, revealing multi-scale structures and complex interaction patterns within the network. Since LLMs contain a huge number of neurons, it is inefficient to apply the fractal analysis to the whole NIN. To this end, we define a Sampled Neural Interaction Network to improve the analysis efficiency and the generalization capabilities of our method.

**Definition 3.2** (Sampled Neural Interaction Network (SNIN))**.** An SNIN represents a sampled subgraph $G' = (V', E')$ of the original NIN. In each layer $L_i$, we sample a subset of nodes $V'_i \in L_i$ and add them to SNIN with weights $E' = \{e_{ab} | e \in E, a, b \in V'\}$.

To enhance the precision and efficiency of the NIN analysis, we adopt a strategy where we average the outcomes from the study of 10 randomly selected SNIN sets in NIN. As substantiated in Appendix A.1.3, this sampling methodology does not alter the estimation of the fractal dimensions. For clarity and simplicity, we consistently use the notation $G = (V, E)$ and refer to the NIN instead of SNIN in subsequent sections.

### 3.3 Neuron-based multifractal analysis

To perform multifractal analysis of LLMs, we extend the box-covering and box-growing methods (Song et al., 2005; Evertsz & Mandelbrot, 1992; Salat et al., 2017; Xiao et al., 2021) to capture the local neuron-based fractal properties of NINs. Specifically, let us consider a neuron $v_i$ as a box with an initial radius of zero at layer $l$, indexed by $i$. We then increase the radius of the box and record the mass distribution $N_{l,i}(r) \in \mathbb{N}^+$ (i.e., the number of nodes covered by the box), as shown in Fig. 4 (a). That is to say, $N_{l,i}(r)$ denotes the number of neighbors of neuron $v_{l,i}$ within a radius $r$.

For a specific neuron $v_i$ at layer $l$ and radius $r$, $N_{l,i}(r)$ is calculated as $N_{l,i}(r) = \sum_{v_j \in \mathcal{N}(v_i)} \mathbf{1}_{\{d_{ij} \leq r\}}$, where $\mathbf{1}_{\{d_{ij} \leq r\}}$ is an indicator function that takes the value 1 when the distance $d_{ij}$ between neighbor neurons $v_i$ and $v_j$ is less than or equal to $r$ and 0 otherwise. The process continues to increase the box radius until all the neighbors of the neuron are covered. Based on this approach, we capture the log-log relationship between the box radius $r$ and the mass distribution $N_{l,i}(r)$ in the box as shown in Fig. 4 (b), indicating the fractal properties of the neuron interactions. The log-log relationship between $N(r)$ and $r$ through the whole NIN presented in Appendix Table 9 leads to the following observation.

**Observation 1** (Fractal Properties)**.** In an NIN, the relation between the mass distribution $N(r)$ and box radius $r$ can be expressed as:
$$N(r) \sim r^D, \tag{3}$$
where $D \in \mathbb{R}$ denotes the fractal dimension characterizing the fractal properties of the interactions among a neuron and its neighbors.

To capture and distinguish the neuron-based multifractality of the NIN, we develop the Neuron-based Multifractal Analysis (NeuroMFA) framework. This method integrates the fractal features of different neurons in the NIN. The distortion factor $q \in \mathbb{R}$ is introduced to differentiate the nuances of fractal structural features. For a given neuron $v_i$ at layer $l$ in the neural network $G = (V, E)$, the mass probability measure $p_{l,i}(r)$ at a radius $r$ is defined as $p_{l,i}(r) = \frac{N_{l,i}(r)}{T_{l,i}}$, where $N_{l,i}(r)$ is the number

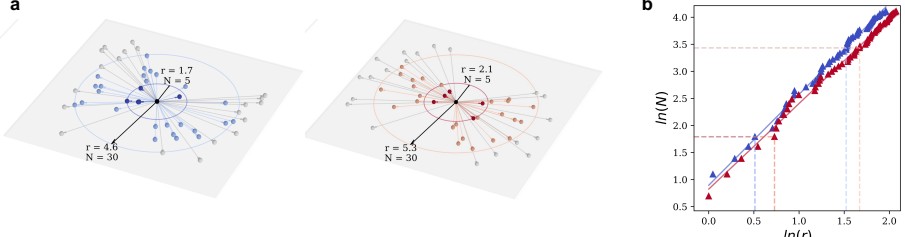

Figure 4: **(a)** Examples of our weighted box growing method, illustrating two neurons randomly selected from the NIN of pythia 1B model at the 143,000[th] epoch. The radius of each box, $r$, is determined by the distance to its neighboring neurons, with $N$ representing the number of neurons contained within the box. **(b)** The power-law relationship between $N$ and $r$.

of neurons within the radius $r$ from neuron $v_i$, and $T_{l,i}$ is the total number of neighbors of neuron $v_i$. To characterize the fractal properties at different scales, the partition function is defined as follows:

**Partition function.** The partition function $Z_q(r)$ for the neuron-based multifractal analysis is defined as the sum of the $q$-th power of the probability measures across all neurons in the network, given by:

$$Z_q(r) = \sum_{l=1}^{L} \sum_{v_i \in L_l} p_{l,i}(r)^q, \tag{4}$$

where $q$ is the distortion factor designed to distinguish different fractal features. We provide the algorithm to obtain the partition function in Appendix Algorithm 2. By capturing the power-law relationship between the partition function $Z_q(r)$ and the scale $\frac{r}{d}$, we have the following observation:

**Lemma 3.1** (Multifractal Scale Invariance). Let $\mu$ be a mass probability measure defined on the network. For any positive number $q$ and scale $\epsilon$, if there exists a constant $C(q)$ such that

$$\sum_i [\mu(B_i(\epsilon))]^q \approx C(q)\epsilon^{\tau(q)}, \tag{5}$$

then the network is said to possess multifractal scale invariance (Falconer, 2014). Here, $B_i(\epsilon)$ represents the box of scale $\epsilon$, and $\tau(q)$ is the mass exponent.

**Observation 2** (Multifractal Properties). A log-log relationship exists between $Z_q(r)$ and the observational scale $r/d$, validated in Appendix Table 9:

$$Z_q(r) \sim \left(\frac{r}{d}\right)^{\tau(q)}, \tag{6}$$

where $d$ is the maximum box radius and $\tau(q)$ is the mass exponent.

$\tau(q)$ is estimated as the slope obtained from the linear regression of $\log Z_q(r_k)$ against $\log\left(\frac{r_k}{d}\right)$:

$$\tau(q) = \frac{\sum_{k=1}^{m} \left(\log Z_q(r_k) - \overline{\log Z_q(r)}\right)\left(\log\left(\frac{r_k}{d}\right) - \overline{\log\left(\frac{r_k}{d}\right)}\right)}{\sum_{k=1}^{m}\left(\log\left(\frac{r_k}{d}\right) - \overline{\log\left(\frac{r_k}{d}\right)}\right)^2}. \tag{7}$$

Here, $r_k$ represents the distance from a neuron to its $k$-th nearest neighbor within the threshold distance $d$, with $k$ ranging from 1 to $m$, where $m$ is the total number of neighbors within the box, and $r_m \leq d$ is the maximum box radius. The terms $\overline{\log Z_q(r)}$ and $\overline{\log\left(\frac{r_k}{d}\right)}$ denote the mean values of $\log Z_q(r_k)$ and $\log\left(\frac{r_k}{d}\right)$, respectively. The mass exponent $\tau(q)$ quantifies the scaling behavior of the partition function $Z_q(r)$ across different observational scales $r$, thereby characterizing the multifractal nature of the neural interactions within the network. The algorithm for computing $\tau$ is provided in Appendix Algorithm 3.

**Multifractal spectrum.** The Legendre transform enables the computation of the the Lipschitz-Hölder exponent $\alpha(q)$ and multifractal spectrum $f(\alpha)$ characterizing the multifractal structural features of the network:

$$\alpha(q) = \frac{d\tau(q)}{dq}, \tag{8}$$

$$f(\alpha) = q\alpha(q) - \tau(q). \tag{9}$$

The detailed algorithm to calculate the multifractal spectrum is provided in Appendix Algorithm 4.

### 3.4 DEGREE OF EMERGENCE

While the Lipschitz–Hölder exponent $\alpha$ provides information about the nature of regularity/order and singularity, the multifractal spectrum $f(\alpha)$ offers insight into the frequency (commonly or rarely occurring pattern) and thus can help us quantify the complex and heterogeneous nature of weighted network topology. Here we extract measures from the multifractal spectrum to quantify the irregularity and heterogeneity of network structures, and use these to construct our measure for the degree of emergence.

**Irregularity metric $\alpha_0$:** The exponent $\alpha_0$ is defined as the Lipschitz–Hölder exponent value $\alpha$ at which the multifractal spectrum $f(\alpha)$ attains its maximum, i.e.,

$$\alpha_0 = \arg\max_\alpha f(\alpha). \tag{10}$$

The exponent $\alpha_0$ represents the most prevalent degree of singularity within the network, and a higher value of $\alpha_0$ indicates a greater level of irregularity and complexity in the network's structure. Therefore, we employ $\alpha_0$ as a metric for assessing the NIN's regularity. A more detailed explanation is provided in Appendix A.2.1.

**Heterogeneity metric $w$:** The width $w$ of the multifractal spectrum is defined as the difference between the maximum and minimum values of the Lipschitz–Hölder exponent $\alpha$, represented by

$$w = \alpha_{\max} - \alpha_{\min}. \tag{11}$$

This width $w$ measures the heterogeneity of the network, with a larger value indicating a broader range of fractal structures within the network. Further details are provided in Appendix A.2.2.

**Proposed metric.** To evaluate the degree of emergence of large models based on structural dynamics of NIN, we estimate two key metrics at a given epoch $t$: the order/regularity $\alpha_0(t)$ and heterogeneity $w(t)$. These metrics are derived from the NIN's evolving network representation, which provides insights into the model's internal structure as it progresses through training.

**Definition 3.3** (Degree of Emergence). The degree of emergence can be calculated through the following formula:

$$E = \frac{w(t)}{w(0)} \log\left(\frac{\alpha_0(0)}{\alpha_0(t)}\right). \tag{12}$$

This formula integrates the changes in heterogeneity and regularity over time, providing a comprehensive measure of the system's structural evolution. Specifically, $\frac{w(t)}{w(0)}$ signifies the relative change in heterogeneity, where values greater than 1 indicate an increase in the diversity of interaction patterns within the network. Similarly, $\log\left(\frac{\alpha_0(0)}{\alpha_0(t)}\right)$ quantifies the shift in regularity, where values greater than 0 represent the process of the network structure becoming more organized. Together, the product of these terms offers a normalized view of the structural shifts in the network, illustrating the degree of self-organization. In this context, self-organization refers to the spontaneous development of new interaction patterns and the increasing regularity within the system, which ultimately leads to the development of skills in large models. A complete explanation is provided in Appendix A.2.

## 4 EXPERIMENTS

In this section, we describe the experiments conducted to analyze the NINs to observe the structural dynamics process during training and evaluate our proposed metrics. Our experiments are conducted

on Pythia from 14M to 2.8B (Biderman et al., 2023) models (except for specific requirements like different architectures). Pythia models provide checkpoints during training phases and different model scales, which is helpful for us to reveal the rules behind these scales of training steps and model size. A detailed description of all the benchmarks used for comparison is provided in Appendix B.6. In the following experiments, we sample 10 networks with 64 nodes per layer and average the results to obtain precise, model-independent calculations. We set parameter $\lambda$ to 1 and $\gamma$ to 5 in Eq. 1 to achieve an appropriate distance between neurons. The variance of the calculations shown in Appendix D.1 demonstrates the reliability of our sampling method.

## 4.1 SELF-ORGANIZATION PROCESS OF THE NIN IN LLMS

We analyze the dynamic LLM networks during their training process. The self-organization process, characterized by the emergence of new patterns and increasingly organized structures, can be quantitatively assessed using the $w$ and $\alpha_0$ metrics from Section 3.4. For each model with different sizes, NeuroMFA is applied to networks at various epochs. The curves showing the changes in the values of $\alpha_0$ and $w$ with epochs are provided in Appendix Fig. 18. The spectra, which vary with the epochs, exhibit distinct behaviors that reveal certain trends, as discussed below.

**Increasing heterogeneity.** The spectrum width $w$ measures the degree of heterogeneity of the fractal interaction patterns within LLMs. Figure 5 shows that $w$ tends to broaden with more training epochs. This widening, evident from the initial spectrum to later stages, signals an increasing heterogeneity in the network's structure. This phenomenon is largely due to the emergence of varied fractal patterns, which are intricate structures formed through neuronal interactions during training. These patterns contribute to the spectrum width's expansion.

The analysis of relatively smaller networks, ranging in size from 14M to 410M parameters, reveals a consistent trend: the spectrum width $w$ increases steadily. However, an interesting deviation occurs in larger models exceeding 1 billion parameters. After approximately 35,000 training epochs, the spectrum width stabilizes, suggesting that the network's fractal diversity reaches a sort of equilibrium at this stage.

**Decreasing irregularity.** The leftward shift of the multifractal spectrum indicates self-organization. This shift suggests an increase in the network's regularity, as marked by a decrease in the $\alpha$ parameter (metric of irregularity). However, Fig. 5 (a) shows for the smaller model (e.g., 14M) an increasing trend of $\alpha_0$. As training progresses through epochs, the spectrum tends to shift rightward slightly instead of leftward. This indicates that, in the 14M model, the network's regularity does not increase. Similarly, the 31M model displays a subtle leftward shift at the 5,000th epoch, but this trend does not continue significantly thereafter.

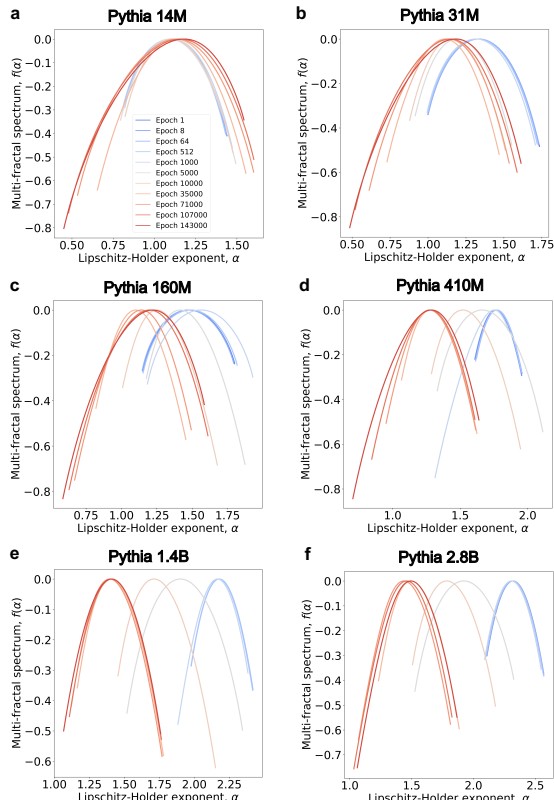

Figure 5: Multifractal spectra of Neuronal Interaction Networks (NINs) for models with different sizes (14M-2.8B) throughout the training process. The gradient of spectral lines, shifting from blue to red, represents the multifractal spectra of the same model at different epochs, with blue indicating early-stage training and red signifying later stages.

In contrast, for larger models (exceeding 100M parameters), a consistent leftward shift of the network spectra is observed from the 0th to the 35,000th epoch. This shift suggests an increase in regularity and a corresponding enhancement in self-organization as training progresses. After reaching the 3,5000th

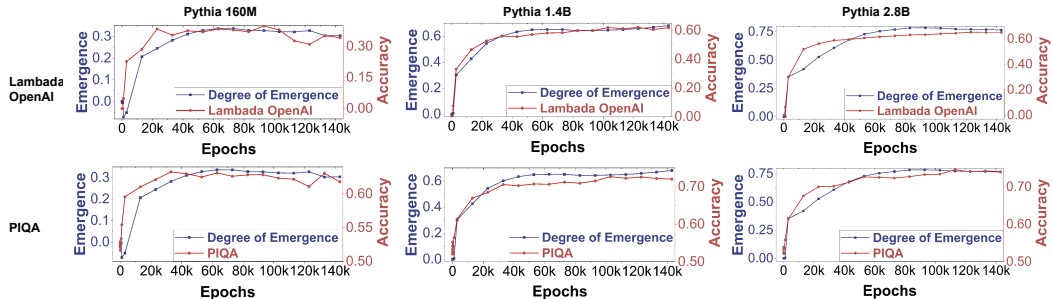

Figure 6: Comparison of the proposed emergence metric $E$ (blue line) with two established metrics (red line). The strong correlation between our proposed metric and existing metrics demonstrates the validity of our approach in quantifying emergent properties.

epoch, this leftward shift in the spectrum stabilizes across these larger models. This stabilization implies that further increases in training epochs do not significantly affect the network's degree of regularity. This could indicate that the model training has reached a certain threshold, beyond which no additional self-organization is observed. Notably, since the degree of emergence is fundamentally tied to self-organization phenomena, we did not compute this metric for the 14M and 31M models in the following experiments.

## 4.2 QUANTITATIVE ANALYSIS OF EMERGENCE

In this section, we assess the degree of emergence $E$ for LLMs during the training process. Figure 6 shows the emergence analysis across models of varying scales (160M, 1.4B, 2.8B) and their performance for different benchmarks. We observe a positive correlation, i.e., higher levels of emergence are associated with enhanced expected model performance, as evidenced by increases in testing accuracy; of note, we acknowledge that this is restricted to these datasets and a more comprehensive evaluation should be done across more datasets (see more results across different metrics in Appendix D.2).

Data from Lambada and PIQA tests reveal a significant increase in model accuracy during the initial training phase, up to approximately the 15,000th epoch. This trend was paralleled by a swift increase in $E$. From around the 15,000th to the 40,000th epochs, we note a gradual enhancement in model performance, aligning with the trajectory of the degree of emergence $E$. Post this phase, models tend to stabilize, exhibiting fluctuating accuracies. These fluctuations mirror the emergence levels, further validating our metric.

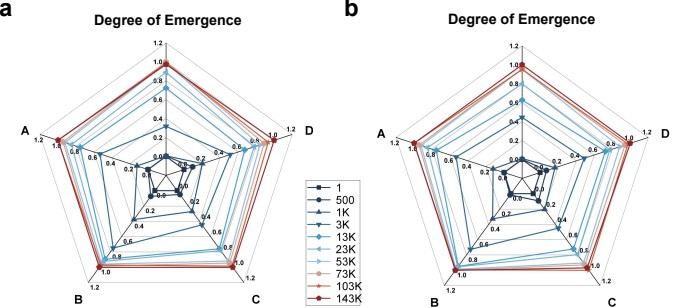

Figure 7: Radar charts showing the performance of **(a)** 1B and **(b)** 1.4B models across cognitive and reasoning benchmarks over different training epochs. A: LAMBADA dataset challenges, B: science exam questions, C: Physical Interaction QA, D: AI2 Reasoning Challenge - Easy.

Moreover, we observe significant performance oscillations in two other datasets (WSC and LogiQA, see Appendix D.2), suggesting their limited utility in assessing model capabilities. This underscores the limitations of relying solely on performance metrics for evaluating the model's emergent abilities. Therefore, our approach to studying emergence focuses on analyzing the model itself, rather than merely its output or performance. This methodology provides a more comprehensive view of the emergence phenomenon in LLMs, encapsulating both the intricacies of the training process and the nuanced evolution of the model. Information about the benchmarks can be found in Appendix B.6.

**Degree of emergence and model capabilities.** Fig. 7 compares our degree of emergence metric with established LLM benchmarks: Lambada, SciQ, PIQA, and ARC-easy. These benchmarks assess different aspects of LLM emergent abilities: Lambada evaluates contextual comprehension, testing

the model's ability to understand long-range dependencies in text. SciQ measures scientific reasoning, challenging models to apply scientific concepts beyond mere fact recall. PIQA assesses physical commonsense, requiring models to reason about everyday physical interactions. ARC-easy tests basic science knowledge and fundamental reasoning across various disciplines. Together, these metrics provide a comprehensive assessment of LLM capabilities, covering language understanding, scientific reasoning, commonsense knowledge, and basic problem-solving skills. We analyze Pythia 1B and Pythia 1.4B models across different training epochs. The radar charts demonstrate that our proposed degree of emergence aligns closely with these established metrics as training progresses.

Our metric reflects the gradual enhancement of LLM's overall capabilities as the training epoch increases, mirroring the trends observed in other LLM benchmarks. This correlation suggests that our degree of emergence, derived from the NeuroMFA method, effectively captures the comprehensive development of LLMs over time. The consistent behavior between our structure-based metric and the diverse set of established performance benchmarks throughout the training process validates the NeuroMFA approach. It also demonstrates the feasibility of assessing LLM capabilities from a structural perspective, offering a new pathway to evaluate the comprehensive abilities of language models based on their internal organization rather than just output performance.

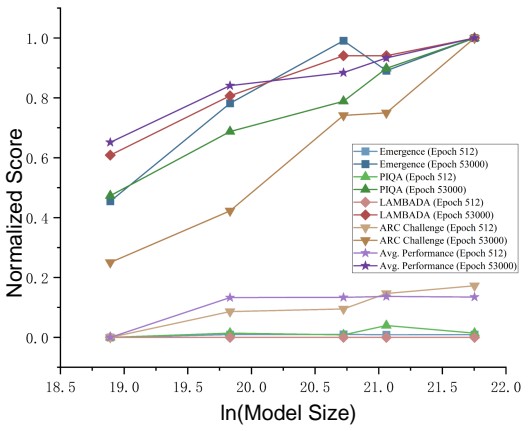

Figure 8: Degree of Emergence vs. Model Size. Avg. Performance means the average score of all the metrics. All metric scores have been normalized between 0 and 1 for comparison.

**Degree of emergence vs. model size.** We evaluate the degree of emergence across models of varying sizes by comparing their performance on different benchmarks, specifically PIQA, LAMBADA, and ARC Challenge, along with the average score across all evaluation metrics. As illustrated in Fig. 8, at an early stage of training (512 epochs), all models, regardless of size, exhibit a low degree of emergence, reflecting limited emergent abilities across the benchmarks. However, at a later stage of training (53,000 epochs), the degree of emergence increases logarithmically with model size, suggesting that larger models demonstrate more pronounced emergent abilities across a range of tasks.

### 4.3  IMPACT EVALUATIONS

We applied NeuroMFA to analyze varying architectures (e.g., diffusion models, CNNs) and training data (e.g., contaminated vs. clean datasets, Pythia-standard vs. Pythia-deduped). The results, detailed in Appendix C, highlight differences in emergent abilities across architectures and the impact of training data on self-organization and emergence.

## 5  DISCUSSION AND CONCLUSION

Our Neuron-based Multifractal Analysis (NeuroMFA) framework addresses a critical gap in quantifying the internal dynamics of large language models (LLMs). By analyzing the multifractal properties of neuron interactions, NeuroMFA offers unique insights into emergent abilities during training. Metrics such as the regularity metric and heterogeneity metric track the evolving complexity and organization within neural networks, providing a cohesive explanation for learning evolution in LLMs. The experimental results demonstrate that emergence in LLMs can be explained by modeling and analyzing the structural and self-organization properties among neurons, aligning with theories of brain function. While accessing human brain neuronal parameters remains challenging, LLMs provide a transparent and accessible platform for studying self-organization phenomena, making this research particularly promising. In future applications, NeuroMFA holds potential not only for enhancing interpretability and assessment but also for forecasting LLMs' behavior and guiding their development. Moreover, by linking structure to the capability of large artificial models, NeuroMFA enables a deeper exploration of potential connections between artificial and biological intelligence.

## ACKNOWLEDGEMENTS

The authors want to thank Ziming Liu for meaningful discussions. The authors acknowledge the support by the U.S. Army Research Office (ARO) under Grant No. W911NF-23-1-0111, the National Science Foundation (NSF) under the Career Award CPS-1453860, CCF-1837131, MCB-1936775, CNS-1932620 and the NSF award No. 2243104 under the Center for Complex Particle Systems (COMPASS), the Defense Advanced Research Projects Agency (DARPA) Young Faculty Award and DARPA Director Fellowship Award under Grant Number N66001-17-1-4044, an Intel faculty award and a Northrop Grumman grant. P.B. is also grateful to National Institute of Health (NIH) for the grants R01 AG 079957 "Interpretable machine learning to synergize brain age estimation and neuroimaging genetics" and RF1 AG 082201 "Neurovascular calcification and ADRD in two nonindustrial Native American populations". It was a wonderful experience designing and writing the grant application entitled "Neurovascular calcification and ADRD in two nonindustrial Native American populations" and awarded under RF1 AG 082201. The views, opinions, and/or findings in this article are those of the authors and should not be interpreted as official views or policies of the Department of Defense, the National Institute of Health or the National Science Foundation.

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

# Appendix

## A THEORIES AND METHODOLOGY DETAILS

### A.1 MULTIFRACTAL ANALYSIS

#### A.1.1 HAUSDORFF MEASURE

The Hausdorff measure is a key concept in fractal geometry and measure theory, extending the notion of Lebesgue measure to irregular sets, and is particularly useful for characterizing the size of fractals. It's named after Felix Hausdorff, a pioneer in set theory and topology.

Formally, consider a set $A$ within $\mathbb{R}^n$. For a metric space $(X, \rho)$, the diameter of $U$ is defined as:

$$\operatorname{diam} U := \sup\{\rho(x, y) : x, y \in U\}, \operatorname{diam} \varnothing := 0, \tag{13}$$

where $U$ is any subset $U \subset X$. For $\delta > 0$, define the $\delta$-approximate $d$-dimensional Hausdorff measure as:

$$\mathcal{H}_\delta^d(A) = \inf\left\{\sum_{i=1}^\infty (\operatorname{diam}(U_i))^d : \bigcup_{i=1}^\infty U_i \supseteq A, \operatorname{diam}(U_i))^d < \delta \; (\{U_i\} \text{ is a } \delta\text{-cover of } A)\right\} \tag{14}$$

where $\operatorname{diam}(U_i)$ is the diameter of the set $U_i$, and a $\delta$-cover is a countable collection of sets $\{U_i\}$ covering $A$ with $\operatorname{diam}(U_i) < \delta$ for all $i$. The $d$-dimensional Hausdorff measure of $A$ is then defined as the limit of $\mathcal{H}_\delta^d(A)$ as $\delta$ approaches zero:

$$\mathcal{H}^d(A) := \sup_{\delta > 0} \mathcal{H}_\delta^d(A) = \lim_{\delta \to 0} \mathcal{H}_\delta^d(A) \tag{15}$$

The Hausdorff dimension $\dim_H(A)$ of a set $A$ is defined as the infimum over all $d$ such that the $d$-dimensional Hausdorff measure of $A$ is zero:

$$\dim_H(A) = \inf\{d \geq 0 : \mathcal{H}^d(A) = 0\}. \tag{16}$$

Equivalently, it can be described as the supremum over all $d$ such that the $d$-dimensional Hausdorff measure of $A$ is infinite:

$$\dim_H(A) = \sup\{d \geq 0 : \mathcal{H}^d(A) = \infty\}. \tag{17}$$

This dimension is a critical value that separates the scales at which the set $A$ appears "large" from those at which it appears "small" in terms of its $d$-dimensional Hausdorff measure. It provides a precise mathematical way to quantify the notion of dimension for irregular sets, like multifractals, which do not fit neatly into the traditional framework of integer dimensions.

#### A.1.2 MULTIFRACTAL ANALYSIS BY BOX-COUNTING METHOD

In the domain of network analysis, multifractal analysis (Song et al., 2005; 2007; Furuya & Yakubo, 2011) employs the box-counting method to elucidate the intricate, multifractal structures inherent in complex networks. This analytical approach aligns with the renormalization procedure, essential for understanding multifractality within these systems. The process commences with covering the network using a series of boxes of variable sizes, denoted as $\epsilon$. The count of boxes, $N(\epsilon)$, required to encompass the network at each scale is meticulously recorded. This methodology resonates with the principles of renormalization, highlighting the dynamic interplay between different scales within the network. This approach is akin to assessing the Hausdorff measure A.1.1, a foundational concept in fractal geometry that quantifies the size of a fractal object. Here we provide the process of the calculation of the generalized fractal dimension and the multifractal spectrum.

The fractal structure is divided into boxes (or elements) of size $\epsilon$. Each box covers a part of the fractal, with a total of $N(\epsilon)$ boxes. For each box, we calculate the proportion of the mass probability measure within that box, denoted as $p_i$, where $i$ represents the $i$-th box. By applying a $q$-th power weighting

to the probability $p_i$ in each box, we obtain the partition function $\sum_{i=1}^{N(\epsilon)} p_i^q$. The distortion factor $q$ can adjust the relative importance of different box probabilities. Considering the scale factor as $\epsilon$ approaches 0 to capture the fractal, the generalized fractal dimension is calculated as:

$$D_q = \frac{1}{q-1} \lim_{\epsilon \to 0} \frac{\log \sum_{i=1}^{N(\epsilon)} p_i^q}{\log \epsilon} \tag{18}$$

Subsequently, through Legendre transform, the Lipschitz-Hölder exponent $\alpha(q)$ is determined through the differential equation:

$$\alpha(q) = \frac{d}{dq}((q-1)D_q) \tag{19}$$

This exponent $\alpha(q)$ provides insights into the local regularity of the network structure at various scales. Advancing this analytical framework, the multifractal spectrum $f(\alpha)$ is derived from the relationship:

$$f(\alpha) = q\alpha - ((q-1)D_q) \tag{20}$$

The multifractal spectrum $f(\alpha)$, in conjunction with $\alpha(q)$, offers a comprehensive understanding of the network's scaling behaviors, revealing the multifractal and heterogeneous nature of its structure. This spectrum delineates the distribution of singularities across the network, thus encapsulating the essence of multifractality. The utilization of these calculations in network analysis is not merely computational; they provide profound insights into the network's heterogeneity, unraveling the multifractal core of its structure and the dynamics that shape its evolution.

### A.1.3 POWER LAW IN FRACTAL DIMENSION

The definition of fractal dimension is based on unconventional views of scaling and dimension. In conventional geometry, dimension can be defined by the intuitive space scaling law. For example, a line with finite length can contain 3 lines with 1/3 its size. But for a square, with a box of side length 1/3 the size of a square, one will find 9 times as many squares as with the original. In general, mathematical definition of such a scaling law can be given by

$$N = \varepsilon^{-D}, \tag{21}$$

where $N$ stands for the number of measurement units, $\varepsilon$ is the scaling factor. Thus, this gives the definition of dimension D:

$$D = -\log_\varepsilon N = -\frac{\ln N}{\ln \varepsilon}. \tag{22}$$

For lines, we can see $N = 3$ when $\varepsilon = \frac{1}{3}$. Then we get $D = 1$. In the case of a square $D = 2$ because $N = 9$ when $\varepsilon = \frac{1}{3}$.

Then the definition can be generalized into fractal geometry. For example, the Koch snowflake shown in Fig.9, $N = 4$ when $\varepsilon = \frac{1}{3}$. Thus the fractal dimension $D = \log_3 4 \approx 1.2618595$, which is not a integer. Even though it is not intuitive any more, the scaling law $N = \varepsilon^{-D}$ still exists. Then for fractal objects embedded in the Euclidean space we can use box-covering method to define fractal dimensions, which is useful for fractal networks:

$$N_B(r_B) \sim r_B^{-D}, \tag{23}$$

where $N_B(r_B)$ is the minimum number of boxes needed to tile a given fractal network with the box radius $r_B$. Then we can define the fractal dimension in another way. The mass distribution is defined by the average number of vertices $\langle N(r_C) \rangle$ within a box with radius $r_C$. The mass-radius relation can be given when the relation $M \sim N_B(r_B)\langle N(r_C)\rangle$ holds for $r_B = r_C = r$. we can then characterize the power law between the mass distribution $N(r)$ and the box radius $r$ as

$$N(r) \sim r^D, \tag{24}$$

where $D$ is the fractal dimension.

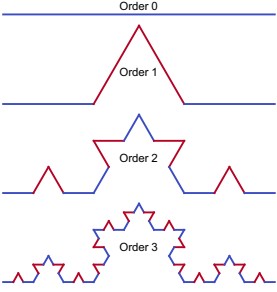

Figure 9: Fractal pattern.

**Lemma A.1** (Scale Invariance). Let $\mu$ be a mass measure defined on the network. For any radius $\epsilon$ around a specific node, if there exists a constant $C$ and a real number $D$ such that the following relationship holds:

$$\mu(B(\epsilon)) \approx C\epsilon^D, \tag{25}$$

where $B(\epsilon)$ is the set of nodes within radius $\epsilon$ of the specified node, then the network exhibits scale invariance at the local scale around the node. $D$ represents the local fractal dimension.

**Definition A.1** (Neuron-based Fractal Dimension). Consider a neuron $v_i$ in layer $l$ of the NIN. The neuron-based fractal dimension $D_{l,i}$ is defined to capture the fractal characteristic based on each neuron. Given the set of distances $\{r_1, r_2, ..., r_m\}$ from $n_{l,i}$ to its neighbors in the next layer $l+1$, the fractal dimension $D_{l,i}$ is calculated as follows:

$$D_{l,i} = \frac{\sum\limits_{k=1}^{m} (\log(r_k) - \bar{\log}(r))(\log(N_{l,i}(r_k)) - \log(\bar{N}_{l,i}(r)))}{\sum\limits_{k=1}^{m} (\log(r_k) - \bar{\log}(r))^2} \tag{26}$$

where $r_k$ represents each distances in the set, $N_{l,i}(r_k)$ is the number of neurons within the distance $r_k$, and $\bar{\log}(r), \log(\bar{N}_{l,i}(r))$ are the mean values of $\log(r_k)$ and $\log(N_{l,i}(r_k))$, respectively.

In practice, to calculate the fractal dimension of a large network we do not need to count all nodes. A practical method is to randomly sample the nodes in the box. Note that, when we plot a $\ln r - \ln N$ graph, the slope is the fractal dimension $D$:

$$\ln N = D \ln r. \tag{27}$$

When we randomly sample the nodes with a constant ratio $k$, it only generates a constant intercept compared to the original:

$$\ln N' = \ln kN = \ln N + \ln k = D \ln r + \ln k, \tag{28}$$

where $\ln k$ is a constant. Then we can still calculate the fractal dimension from its slope.

## A.2 ELABORATION FOR NEUROMFA

### A.2.1 LIPSCHITZ-HÖLDER EXPONENT

In the context of multifractal analysis, the Lipschitz-Hölder exponent, denoted as $\alpha$, is crucial for characterizing the local scaling properties of a dataset. Mathematically, $\alpha$ is defined as the derivative of the mass exponent $\tau(q)$ with respect to $q$, expressed as $\alpha(q) = \frac{d\tau(q)}{dq}$. Here, $\tau(q)$ represents the mass exponent that characterizes the scaling behavior of the data at different moments ($q$).

The value of $\alpha$ provides insights into the degree of irregularity at various scales within the dataset. Lower values of $\alpha$ suggest a higher level of regularity, where the data exhibits smoother transitions and less variability in its local structure. In contrast, higher values of $\alpha$ are representative of regions with more irregular and disordered scaling behavior. This distinction is fundamental to multifractal analysis, offering a detailed perspective on the heterogeneous nature of the data and its scaling properties across various scales.

Therefore, the distribution and range of $\alpha$ values within the multifractal spectrum offer valuable insights into the underlying scaling dynamics of the dataset, revealing the intricate interplay between uniformity and complexity that defines its multifractal nature.

### A.2.2 SPECTRUM WIDTH

The spectrum width, denoted as $w$, in multifractal analysis plays a pivotal role in quantifying the heterogeneity of a network's structure. It is defined as the difference between the maximum and minimum values of the Lipschitz-Hölder exponent $\alpha$, mathematically expressed as $w = \alpha_{\max} - \alpha_{\min}$.

This width $w$ captures the range of scaling behaviors exhibited by the network across different regions. A larger $w$ indicates a broader spectrum of $\alpha$ values, signifying a network with a wide variety of scaling behaviors. Such diversity in scaling properties points to a network with a rich mixture of regular and irregular structures, highlighting the complexity and variability in its composition.

In essence, the spectrum width $w$ serves as a crucial metric for assessing the structural heterogeneity of a network. It reflects the degree to which the network deviates from uniform scaling behavior, offering insights into the multifaceted and complex nature of the network's fractal characteristics. The broader the spectrum, the more pronounced the heterogeneity, revealing a network structure that is rich in its diversity.

### A.2.3 DEGREE OF EMERGENCE

In the study of self-organizing networks, the phenomenon of emergence is a key concept, representing the evolution from simpler initial states to more complex and ordered structural patterns. The measure of emergence, denoted as $E$, aims to quantify this transition from disorder to order, capturing the key characteristics of the network's self-organization process.

The measure of emergence $E$ considers two principal aspects: the increase in network regularity and the growth in heterogeneity. Regularity, indicated by the parameter $\alpha_0$, reflects the uniformity and predictability of the network's structure. As the process of self-organization progresses, changes in $\alpha_0$ suggest a transition of the network structure towards greater regularity and consistency. This transition is typically marked by an increase in uniform patterns, indicating a move from a more irregular to a more regular state.

Heterogeneity, represented by the parameter $w$, illustrates the diversity of the network's fractal structure. A larger $w$ value points to a network exhibiting a variety of structural characteristics across different regions. The emergence of new patterns and structures during self-organization leads to an increase in $w$, reflecting a rise in the diversity and heterogeneity of the network's structure.

Thus, the emergence measure $E$ is defined as:

$$E = \frac{w(t)}{w(0)} \log \left( \frac{\alpha_0(0)}{\alpha_0(t)} \right) \tag{29}$$

This metric accounts for the relative changes in regularity and heterogeneity over time, offering a comprehensive perspective to quantify the phenomenon of emergence in the network's self-organization. In this formula, $\frac{w(t)}{w(0)}$ represents the relative change in heterogeneity, while $\log \left( \frac{\alpha_0(0)}{\alpha_0(t)} \right)$ captures the relative change in regularity. The utilization of the logarithm function is crucial, as it allows the emergence measure to have positive or negative values, reflecting whether the multifractal spectrum shifts left or right, a critical indicator of emergence in the model. By integrating these dimensions, $E$ effectively describes the network's evolution from its initial state to a more advanced level of organization, revealing the complex dynamics and characteristics of emergence in the self-organizing process.

### A.2.4 ILLUSTRATION EXAMPLES

We provide concrete examples to illustrate the multifractality and monofractality in Fig. 10. Cantor sets are classical fractal structures created by iteratively removing the middle part of a line segment. Fig. 10 (a) shows the patterns with self-similar statistical properties characterized by a single fractal dimension across all scales while Fig. 10 (b) shows the patterns characterized by different fractal

dimensions across multiple scales. Here, the mass probability $p_i$ is defined as the ratio of the number of segments in each subinterval to the total number of segments. By applying the multifractal analysis, as shown in Fig. 10 (c), the spectrum of a monofractal Cantor set results in a single point, indicating uniform scale characteristics. In contrast, the multifractal Cantor set's spectrum forms a bell-shaped curve, reflecting the diversity of scale characteristics.

In our NeuroMFA, the mass probability is represented as the ratio of nodes in a box to the total number of nodes, with the observation scale being the radius of the box (see Figs. 10 (f-g)). By measuring and distorting the mass distribution in different boxes, we can capture the node-based multifractal characteristics in the network. In Figs. 10 (d-e), we provide an example of a Watts-Strogatz (WS) network model transitioning from a regular network to a completely random network. A completely regular network exhibits monofractal properties, with the multifractal spectrum being a single point. As edges are randomly rewired, the network structure's diversity increases, leading to an increased spectrum width. Simultaneously, the network structure becomes more irregular, indicated by a right-shifted spectrum.

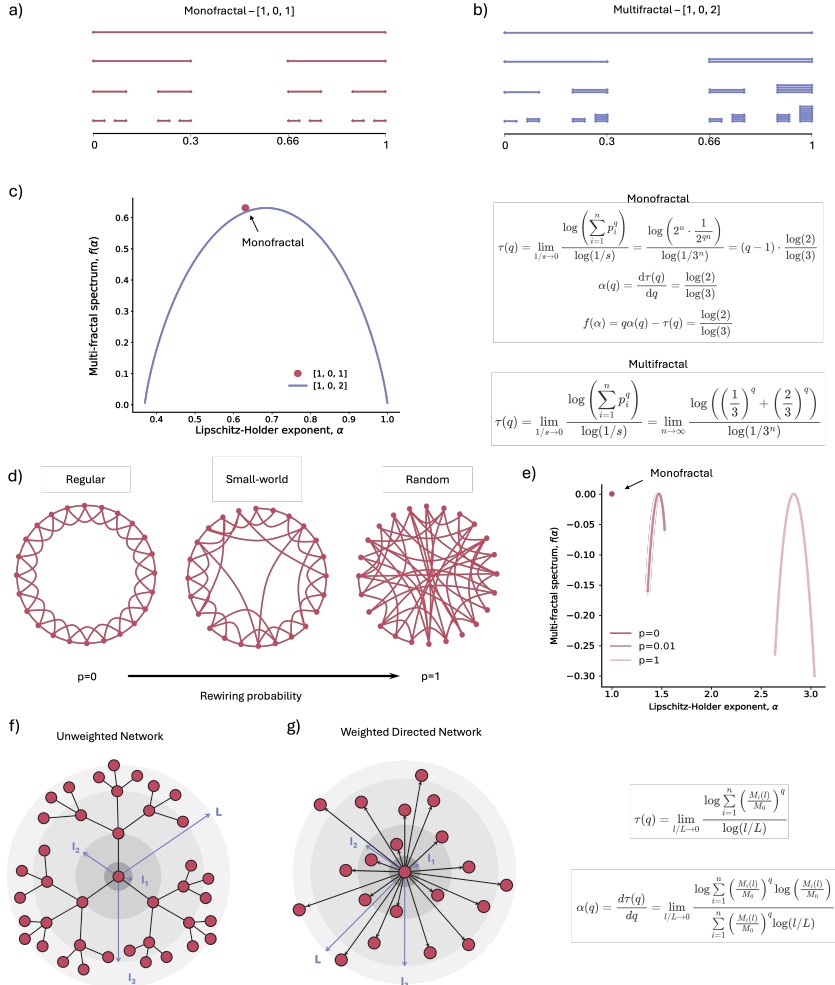

Figure 10: Concrete examples of monofractal and multifractal. Fig. (a) illustrates self-similar patterns with a single fractal dimension across all scales, while Fig. (b) depicts patterns with varying fractal dimensions across multiple scales. Fig. (c) shows the monofractal Cantor set yielding a single-point spectrum, contrasted by the bell-shaped multifractal spectrum for a multifractal Cantor set. In Figs. (d-e), a Watts-Strogatz (WS) network model transitions from a regular to a random network, where rewiring edges increases spectrum width and irregularity, shifting the multifractal spectrum. Figs. (f-g) demonstrate the node-based multifractal analysis in a network, using box-counting techniques.

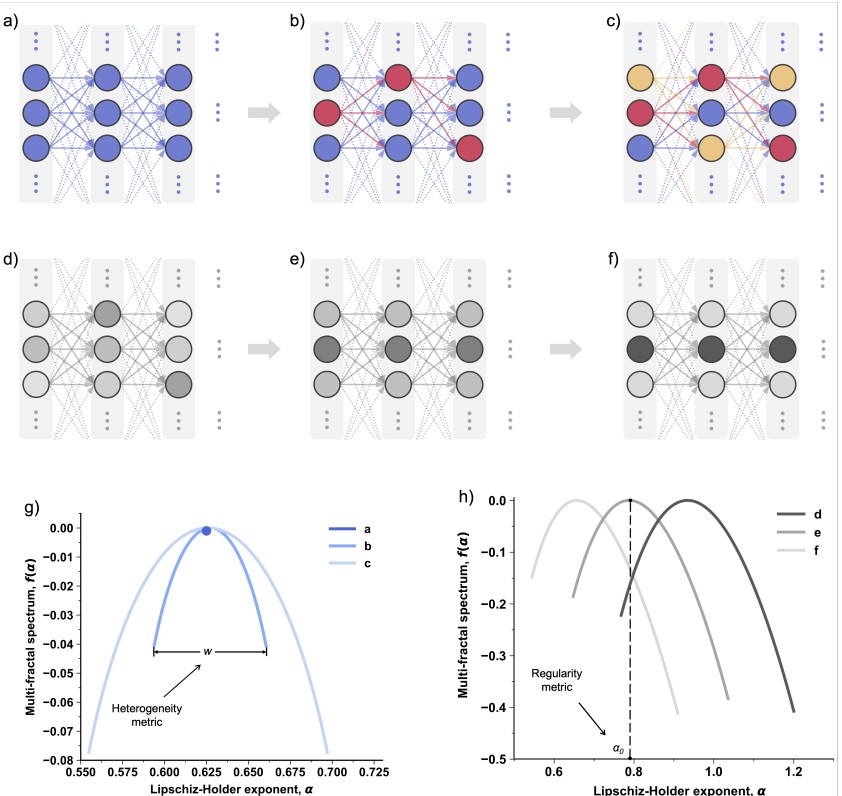

Figure 11: Illustration of the proposed metrics. (a-c) Experiment 1 illustrates the transformation from a homogeneous to a more heterogeneous network, as shown by (g) broadened spectral width. (d-f) Experiment 2 demonstrates increasing regularity, starting from random configurations and progressing through multiple Weighted Preferential Attachment Mechanism (wPAM) iterations, reflected in (h) the shift and decrease in $\alpha_0$.

To intuitively illustrate the metrics, we apply NeuroMFA to toy examples. These metrics provide quantitative insights into the heterogeneity and regularity of neuronal networks. Using simplified models consisting of three layers with 32 nodes each, we demonstrate how network transformations influence these metrics through two experiments:

- Experiment 1: Focused on the effects of increasing network heterogeneity. Starting with a homogeneous network, we systematically varied node characteristics to simulate increased complexity.

- Experiment 2: Explored the introduction of regularity into a network using Algorithm 1 . We began with a randomly structured network and used the wPAM to gradually increase regularity.

The results are visualized in Fig. 11, showing distinct phases of the experiments and how NeuroMFA metrics quantify changes in network heterogeneity and regularity.

---

**Algorithm 1** Multiple Rounds of Weight Adjustment

---

**Require:** $N$ (number of iterations)
 1: **for** $n = 1$ to $N$ **do**
 2:     **for** $i = 1$ to 128 **do**
 3:         **for** $j = 1$ to 32 **do**
 4:             **if** $W_{ij} < \text{median}(W)$ **then**
 5:                 $W_{ij} \leftarrow W_{ij} \cdot \delta$                           ▷ Decrease weight
 6:             **else**
 7:                 $W_{ij} \leftarrow W_{ij} \cdot \iota$                           ▷ Increase weight
 8:             **end if**
 9:         **end for**
10:     **end for**
11: **end for**

---

## A.3 DISCUSSION ON SELF-ORGANIZATION

### A.3.1 EMERGENCE AND SELF-ORGANIZATION

Emergence and self-organization are fundamental concepts that span across multiple disciplines, including physics, biology, computer science, and sociology, among others. They describe the way complex systems and patterns arise out of a multiplicity of relatively simple interactions.

**Emergence.** Emergence refers to the phenomenon where larger entities, patterns, or regularities arise through interactions among smaller or simpler entities that themselves do not exhibit such properties. The essential point about emergent properties is that they are not properties of any component of the system but of the system as a whole.

**Self-organization.** Self-organization is closely related to emergence and is often seen as a process leading to emergent properties. It is the process by which a system, without external guidance, spontaneously forms a coherent and ordered structure.

Emergence and self-organization underpin complex systems across physics, biology, computer science, and sociology. Emergence captures how complex systems and patterns spring from simple interactions. Self-organization, integral to emergence, describes systems spontaneously forming ordered structures without external direction. In Physics, emergence is seen in phenomena like thermodynamics, where macroscopic properties such as temperature and pressure emerge from the collective behavior of particles(Epstein et al., 2006). Self-organization is observed in non-equilibrium processes such as the formation of crystal structures or the spontaneous formation of cyclones in the atmosphere. In Biology, emergence explains how complex biological phenomena, such as life, arise from the interactions of non-living molecules in the right conditions(Kauffman, 1993). The behavior of ant colonies, where complex organization and problem-solving abilities emerge from simple rules followed by individual ants, is another example. The development of an organism from a fertilized egg, where highly ordered structures and functions emerge, is a classic example of self-organization. These concepts highlight the transition from simple interactions to complex behaviors, also exemplified by Turing patterns. Turing patterns serve as a model, illustrating how basic reaction-diffusion processes can create intricate patterns, mirroring the self-organization leading to emergence.

### A.3.2 TURING PATTERN

Turing pattern is a well-known example of how self-organization occurs in various systems. It was first proposed by Alan Turing when he tried to explain the patterns in morphogenesis. His reaction–diffusion theory has provided insights for understanding pattern formation of various complex objects ranging from chemical molecules to sandpiles.

In general, we examine the reaction-diffusion equations

$$\frac{\partial C_i}{\partial t} = F_i(\{C_j\}) + D_i \nabla^2 C_i + \eta(\vec{x}, t). \tag{30}$$

Here, $D_i$ is the diffusion coefficient for object species $i$ (which could be totally different things in different systems, for example, molecular in chemical process) with concentration $C_i$, and reactions

are described by local non-linear terms included in $\{C_i\}$. A stochastic term $\eta(\vec{x}, t)$ is also included to describe the general case with stochastic noise: In the microscopic system it could result from thermal fluctuations, whereas for macroscopic ones it could stand for variations in the environment.

Now we demonstrate the simplest example. The key idea is that simple chemical processes, when subject to diffusion and reaction, can give rise to complex and diverse patterns. The necessary conditions for the formation of Turing patterns in the simplest case are:

- **Morphogens**: There must be at least two components **Activator** and **Inhibitor**. For example, in biological systems, they can be signaling molecules.
- **Reaction**: For the activator it must be autocatalytic (positive feedback), while the inhibitor could suppress the activator. In short, the reactions (auto-catalysis and cross-catalysis) must occur between them.
- **Diffusion**: Both activator and inhibitor diffuse spatially. And the inhibitor diffuse much faster than the activator.

The simplest form is like

$$\begin{cases} \frac{\partial a}{\partial t} = D_a \nabla^2 a + f_a(a, b), \\ \frac{\partial b}{\partial t} = D_b \nabla^2 b + f_b(a, b), \end{cases} \tag{31}$$

where $C_1 = a$ is the concentration of activator, $C_2 = b$ is the concentration of inhibitor, $D_{a,b}$ are diffusion coefficients, $f_{a,b}(a, b)$ are the reaction functions.

In Turing's mathematical analysis, such a reaction-diffusion system could yield six potential steady states, depending on the dynamics of reaction term and wavelength of the pattern(Kondo & Miura, 2010). When the diffusion of the inhibitor is much faster than that of the activator, that is, $D_a \ll D_b$, the steady state becomes a Turing pattern, a kind of nonlinear wave maintained by the dynamic equilibrium of the system.

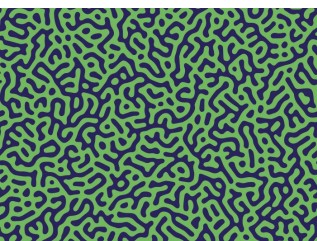

Figure 12: Turing pattern(Turing, 2022).

The process of pattern formation is a symmetry-breaking one: at the beginning all spices diffuse in an isotropic way but finally the patterns become like the one shown in Fig.12. In principle, Turing patterns possess an intrinsic wavelength, which is determined by interactions between molecules and their rates of diffusion. More specifically, it depends only on the ratio of the parameters in the equation, which means it is a scale-independent property.

To demonstrate this mathematically, we try to examine the Eq.30. We need to find a stable fixed point $\{C_i^*\}$ as a solution. First, we apply the linear expansion on the reaction-diffusion equations around the fixed point

$$C_i(\vec{r}, t) = C_i^* + c_i(\vec{r}, t),$$
$$\Rightarrow \frac{\partial c_i}{\partial t} = \sum_j M_{ij} c_j + D_i \nabla^2 c_i + \eta, \ M_{ij} = \frac{\partial F_i}{\partial C_j}\Big|_{C^*}. \tag{32}$$

The stability of the solution requires all eigenvalues of the matrix $M_{ij}$ have negative real parts. We introduce Fourier transforms

$$c_i(\vec{r}, t) = \int d\vec{k} e^{i\vec{k}\cdot\vec{r}} \tilde{c}_i(\vec{k}, t), \tag{33}$$

then Eq.32 becomes

$$\frac{d\tilde{c}_i(\vec{k},t)}{dt} = \sum_j \left(M_{ij} - \delta_{ij}D_i k^2\right)\tilde{c}_j(\vec{k},t) + \tilde{\eta}(\vec{k},t). \tag{34}$$

To get the Turing instability, the eigenvalues question after the transformation becomes whether the matrix $M_{ij}(k) = M_{ij} - \delta_{ij}D_i k^2$ can have a positive eigenvalue at a finite wave-vector $\vec{k}$. Then we come back to the simplest case. Let us examine the $2 \times 2$ linear-stability matrix

$$M(k) = \begin{pmatrix} M_{11} - D_1 k^2 & M_{12} \\ M_{21} & M_{22} - D_2 k^2 \end{pmatrix}. \tag{35}$$

Here we denote two eigenvalues of the matrix by $\varepsilon_\pm(k)$, and we can get their sum

$$\varepsilon_+(k) + \varepsilon_-(k) = \mathrm{Tr}M(k) = (M_{11} + M_{22}) - (D_1 + D_2)k^2, \tag{36}$$

and the product of them

$$\varepsilon_+(k)\varepsilon_-(k) = \det M(k) \tag{37}$$

$$= (M_{11}M_{22} - M_{12}M_{21}) - (M_{11}D_2 + M_{22}D_1)k^2 + D_1 D_2 k^4. \tag{38}$$

When $k = 0$ it is obviously a stable state (uniform state), so we have $\varepsilon_+(0) < 0, \varepsilon_-(0) < 0$. Thus the first term $\det M(0) = \varepsilon_+(0)\varepsilon_-(0)$ is positive at $k = 0$. And the last term $D_1 D_2 k^4$ is always positive. Within the unstable state $(\varepsilon_+(0) > 0, \varepsilon_-(0) < 0)$ we must have $\varepsilon_+(k)\varepsilon_-(k) < 0$, thus the second term is the only route to instability, that is

$$(M_{11}D_2 + M_{22}D_1) > 0, \quad \text{while} \quad (M_{11} + M_{22}) < 0. \tag{39}$$

where $(M_{11} + M_{22}) < 0$ is because of the stability at $k = 0$. The sum requires $\varepsilon_+(0) + \varepsilon_-(0) = \mathrm{Tr}M(0) = (M_{11} + M_{22}) < 0$. Then we can give a curve to describe Eq.38:

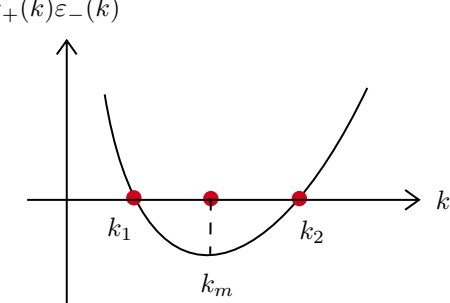

It crosses zero at two points $k_1, k_2$ and there is a minimum $k_m$ requiring

$$k_1^2 + k_2^2 = 2k_m^2 = \frac{M_{11}D_2 + M_{22}D_1}{D_1 D_2}. \tag{40}$$

The band of unstable modes spans wave numbers from $k_1$ to $k_2$. Now recall the necessary conditions for instability in Eq.39, it is clear that when $M_{11}, M_{22}$ are all negative the conditions can not be satisfied. Without loss of generality, we can choose $M_{11} > 0, M_{22} < 0$, which means $C_1$ stands for the activator and $C_2$ stands for the inhibitor. The requirement for instability now becomes

$$|M_{22}|\frac{D_1}{D_2} < M_{11} < |M_{22}|,$$

$$\Rightarrow \frac{D_2}{D_1} > \frac{|M_{22}|}{M_{11}} > 1. \tag{41}$$

Also, we must require

$$\varepsilon_+(k_m) + \varepsilon_-(k_m) = \det M(0) - \frac{(M_{11}D_2 + M_{22}D_1)^2}{4D_1 D_2} < 0 \tag{42}$$

$$\Rightarrow (M_{11}D_2 + M_{22}D_1) > 2\sqrt{D_1 D_2 \det M(0)},$$

which gives a more restrictive condition on the ratio $D_2/D_1$. For example, in the case of the matrix

$$M(k) = \begin{pmatrix} 1 - D_1 k^2 & 1 \\ -5 & -1 - D_2 k^2 \end{pmatrix}, \tag{43}$$

the condition in Eq.41 requires $D_2/D_1 > 1$, while Eq.42 requires $D_2/D_1 > 9 + 4\sqrt{5} \approx 17.9$.

Let us come back to talk about the stochastic term in Eq.32. For chemical reactions, fluctuations obey typical $\sqrt{N}$ rules for a number of $N$ molecules. We assume the stochastic fluctuations are described. by white noise with covariance,

$$\langle \eta(\vec{x}, t)\eta(\vec{x}', t')\rangle = 2N\delta(t - t')\delta^d(\vec{x} - \vec{x}'). \tag{44}$$

After the Fourier transformation, it becomes

$$\langle \eta(\vec{k}, t)\eta(\vec{k}', t')\rangle = \frac{2N}{(2\pi)^d}\delta(t - t')\delta^d(\vec{k} + \vec{k}'). \tag{45}$$

For many variable cases in Eq.32 the power spectrum of fluctuations in steady state is proportional to $1/\det M(k)$ in Eq.38. When $D_2/D_1 > |M_{22}|/M_{11}$, the peak in the power spectrum for noisy fluctuations shifts to $k_m$.

### A.3.3 EMERGENCE AND SELF-ORGANIZATION IN LARGE MODELS

*"Emergence refers to the arising of novel and coherent structures, patterns, and properties during the process of self-organization"* (Goldstein, 1999). Emergence can be described as the formation of new properties - "something new appears" - in the evolution of a collective system that cannot be fully captured by its dynamical equations. These emergent properties arise from local interactions and can be harnessed by the system to develop additional functionalities or capabilities (Crutchfield, 1994). In the context of large models, emergence is the complex behavior through which new learning capabilities are acquired by the LLM from the collective interactions among numerous simple elements (i.e., the neurons following consistent rules within the neural network).

*"Self-organization is the process of forming patterns and achieving overall order through internal local interactions"* (Correia, 2006; Ay et al., 2007). In the human brain, intelligence and complex behaviors arise from the collective behavior of neurons, with the brain operating as a self-organizing system (Kelso, 1995; Singer, 1986; Pribram, 2018). In the context of large models, self-organization refers to the process where the interactions of artificial neurons during training evolve from initial randomness, developing new patterns and orderly structures, thus leading to emergent abilities in large models.

### A.4 COMPARISON WITH TRADITIONAL METRICS

In this section, we further discuss why traditional network analysis methods fail to effectively meet the requirements of an analytical framework for large models, while also highlighting how our proposed NeuroMFA method enables efficient analysis of neural networks in large models, representing a key contribution of our work.

Here are some key challenges in applying network analysis to large-scale models:

1. **Unique Structural Characteristics of Neural Networks:** Neural networks exhibit **directed, weighted, and hierarchical** structures, which make traditional metrics designed for unweighted or undirected networks less applicable.

2. **Scalability to Large Models**: Large models have billions of parameters, creating networks of enormous size. Many traditional metrics become computationally infeasible for networks exceeding 10 million nodes or edges.

3. **Need for Multiscale Analysis:** Our objective is to bridge local (microscopic) structural changes in neural networks with their emergent macroscopic capabilities. This requires metrics capable of capturing both local interactions and global multiscale properties.

Based on these points, we analyzed the potential applications of traditional network metrics in neuronal networks and neural networks, as well as their limitations when applied to large-scale models as below.

1. **Centrality Metrics:** Centrality measures such as degree, closeness, and betweenness centrality quantify the importance of nodes within a network.

   - **Potential Applications:** Useful for identifying key neurons in neuronal networks that are critical for information propagation or specific task processing.
   - **Limitations:** Centrality metrics focus on **individual node-level properties**, making them less suitable for capturing global network dynamics or multiscale interactions.

2. **Small Worldness:** Small worldness quantifies the balance between local clustering and global efficiency using metrics such as local clustering coefficients and average shortest path lengths.

   - **Potential Applications:** In neuronal networks, it helps understand the trade-off between local processing and global information integration, as well as network efficiency.
   - **Limitations:** Primarily designed for undirected and unweighted networks, small worldness (especially for **the calculation of average shortest path**) is computationally expensive for large, weighted networks like those in LLMs.

3. **Modularity:** Modularity measures the degree to which a network can be divided into distinct modules or communities.

   - **Potential Applications:** In neuronal networks, modularity can be used to identify functionally distinct sub-networks or quantify modular brain region connections.
   - **Limitations:** Modularity identifies large-scale structures but fails to capture **node-level interactions** or multiscale properties. Additionally, it was initially designed for undirected and unweighted networks. Although adaptations exist, they are often computationally expensive to analyze the directed and weighted structures in neural networks.

4. **Entropy-Based Metrics:** Entropy metrics quantify the diversity or randomness of network structures, often focusing on degree distributions or edge weights.

   - **Potential Applications:** Useful for measuring the diversity of synaptic connections or activity patterns in neuronal networks.
   - **Limitations:** Entropy metrics analyze **distributions** rather than interactions, making them unsuitable for multiscale analysis. Additionally, entropy calculations for large, directed, and weighted networks are resource-intensive.

5. **Ricci Curvature:** Ricci curvature provides geometric insights into connectivity and robustness by measuring edge-based properties.

   - **Potential Applications:** Applied in neural networks to identify critical pathways for signal transmission or bottlenecks in artificial neural networks.
   - **Limitations:** Ricci curvature is computationally prohibitive for large-scale, weighted networks and focuses on **edge properties**, offering limited insights into global network structure.

**Why NeuroMFA?** Compared to the metrics above, NeuroMFA offers several distinct advantages:

1. **Node-Based Design:** NeuroMFA analyzes nodes directly, making it well-suited for studying local interactions and scaling up to **multiscale analyses**.

2. **Efficiency:** In NeuroMFA, the analysis starts from **node-based local properties** and leverages multiscale characteristics to derive the overall network's multifractal features. This method, which relies on local properties rather than direct computation over the entire network, significantly improves efficiency. Additionally, by leveraging scaling laws, NeuroMFA allows for network sampling without compromising the accuracy of fractal dimension estimation (see Appendix A.1.3, revised version, page 19), making NeuroMFA **computationally efficient**.

3. **Multiscale Analysis:** NeuroMFA captures both local and global properties, enabling us to capture the **heterogeneity** (diversity of local interactions) and **irregularity** (global structure) of the network structure across multiple scales, aligning with natural systems' self-organization and emergence rationale.

In summary, although traditional metrics can be useful for analyzing smaller neural networks, they are often limited by computational inefficiency or lack of the ability for multiscale analysis when applied to large-scale neural networks. NeuroMFA overcomes these challenges by providing an efficient, scalable, and multiscale framework tailored to the unique characteristics of large neural models.

# B IMPLEMENTATION DETAILS

## B.1 CODE AVAILABILITY

The source code is available at `https://github.com/joshuaxiao98/Neuron_LLM`. Here we provide the hyper-parameters we used to produce our experiments in Table 2.

## B.2 EVALUATION LLMS

We experiment on Pythia Biderman et al. (2023) dataset and models. The detailed information can be found in the table 1.

| Model Size | Non-Embedding Params | Layers | Model Dim | Heads | Learning Rate | Equivalent Models |
|---|---|---|---|---|---|---|
| 70 M | 18,915,328 | 6 | 512 | 8 | $10.0 \times 10^{-4}$ | — |
| 160 M | 85,056,000 | 12 | 768 | 12 | $6.0 \times 10^{-4}$ | GPT-Neo 125M, OPT-125M |
| 410 M | 302,311,424 | 24 | 1024 | 16 | $3.0 \times 10^{-4}$ | OPT-350M |
| 1.0 B | 805,736,448 | 16 | 2048 | 8 | $3.0 \times 10^{-4}$ | — |
| 1.4 B | 1,208,602,624 | 24 | 2048 | 16 | $2.0 \times 10^{-4}$ | GPT-Neo 1.3B, OPT-1.3B |
| 2.8 B | 2,517,652,480 | 32 | 2560 | 32 | $1.6 \times 10^{-4}$ | GPT-Neo 2.7B, OPT-2.7B |

Table 1: Models in the Pythia suite and select hyperparameters from Biderman et al. (2023).

And the model being used is based on GPT-NeoXAndonian et al. (2023). It is an open source library GPT-NeoX developed by EleutherAI. The model checkpoints are saved at initialization, at the first 1, 2, 4, 8, 16, 32, 64, 128, 256, 512 iterations, and then every 1000 iterations, making it a total of 154 checkpoints.

The design of GPT-NeoX is based on currently most wide-spread design of open source GPT models. To be more specific, it has the follow features:

- The model uses sparse and dense attention layers in alternation introduced by Brown et al.. It is used in the fully dense layers of the model.
- The model uses Flash AttentionDao et al. (2022) during training for improved device throughput.
- The model uses rotary embeddings introduced by Su et al.. Now it is widely used as a positional embedding type of choice.
- The model uses a parallelized attention and feed-forward technique.
- The model's initialization methods are introduced by Wang & Komatsuzaki and adopted by Black et al. and Chowdhery et al., because they improve training efficiency without losing performance.
- Aside from rotary embeddings, the model uses untied embedding / unembedding matrices Belrose et al. (2023). This makes interpretability research easier, which is very important for our research.

| Model Name | Blocks | Model Dim | Heads | Sample1 ( x 10) | | Sample2 ( x 10) | | Sample3 ( x 10) | |
|---|---|---|---|---|---|---|---|---|---|
| | | | | Nodes Per Layer | Total Nodes | Nodes Per Layer | Total Nodes | Nodes Per Layer | Total Nodes |
| Pythia-14M | 6 | 128 | 8 | 128 | 5504 | 64 | 2752 | 32 | 1376 |
| Pythia-31M | 6 | 256 | 8 | 128 | 5504 | 64 | 2752 | 32 | 1376 |
| Pythia-70M-deduped | 6 | 512 | 8 | 128 | 5504 | 64 | 2752 | 32 | 1376 |
| Pythia-160M-deduped | 12 | 768 | 12 | 128 | 10880 | 64 | 5440 | 32 | 2720 |
| Pythia-410M-deduped | 24 | 1024 | 16 | 128 | 21632 | 64 | 5440 | 32 | 2720 |
| Pythia-1B-deduped | 16 | 2048 | 8 | 128 | 14464 | 64 | 7232 | 32 | 3616 |
| Pythia-1.4B-deduped | 24 | 2048 | 16 | 128 | 21632 | 64 | 10816 | 32 | 5408 |
| Pythia-2.8B-deduped | 32 | 2560 | 32 | 128 | 28800 | 64 | 14400 | 32 | 7200 |

Table 2: SNIN sample parameters

## B.3 Algorithms for NeuroMFA

The detailed algorithms of the calculation of the mass exponent $\tau(q)$ and the multifractal spectrum $f(\alpha)$ are shown in Algorithms 3 and 4.

---

**Algorithm 2** Calculation of the Partition Function $Z(q)$

---

**Require:** $S$ (a dictionary with nodes as keys and lists of distances from the node to its neighbors as values), $q_{\text{list}}$ (a list of distortion factors)
1: $M_{\text{list}} \leftarrow []$
2: $R \leftarrow \{\}$
3: **for all** $s$ in $S$ **do**
4:     $num \leftarrow$ empty map
5:     **for all** $d$ in $S[s]$ **do**
6:         $rdist \leftarrow round(d)$
7:         $num[rdist] \leftarrow num[rdist] + 1$
8:     **end for**
9:     $R \leftarrow R \cup \text{keys}(num)$
10:     append $num$ to $M_{\text{list}}$
11: **end for**
12: $r_{\text{sorted}} \leftarrow \text{sort}(R)$
13: $len_M \leftarrow$ length of $M_{\text{list}}$
14: $len_r \leftarrow$ length of $r_{\text{sorted}}$
15: $W \leftarrow$ matrix of ones with dimensions $len_M \times len_r$
16: **for** $i = 0$ **to** $len_M - 1$ **do**
17:     **for** $j = 0$ **to** $len_r - 1$ **do**
18:         $W[i,j] \leftarrow \sum_{k \leq r_{\text{sorted}}[j]} M_{\text{list}}[i][k]$
19:     **end for**
20: **end for**
21: $Zq_{\text{list}} \leftarrow []$
22: **for all** $q$ in $q_{\text{list}}$ **do**
23:     $Zq \leftarrow \sum_{i=0}^{len_M} \left( \frac{W[i,:]}{W[i,-1]} \right)^q$
24:     append $Zq$ to $Zq_{\text{list}}$
25: **end for**
26: **return** $(r_{\text{sorted}}, Zq_{\text{list}})$

---

**Algorithm 3** Calculation of the Mass Exponent $\tau(q)$ in Eq. 6

---

**Require:** $r_{\text{sorted}}$ (a list of sorted box radius, output from Algorithm 2), $Zq_{\text{list}}$ (a list of Partition functions, output from Algorithm 2), $d$ (maximum of box radius)
1: **for all** $Zq$ in $Zq_{\text{list}}$ **do**
2:     Calculate the slope of the linear regression of $\log(Zq)$ on $\log\left(\frac{r_{\text{sorted}}}{d}\right)$
3:     Save the slopes to an empty list: $\tau_{\text{list}}$
4: **end for**
5: **return** $\tau_{\text{list}}$

---

**Algorithm 4** Calculation of the Multifractal Spectrum $f(\alpha)$ in Eq. 9

---

**Require:** $\tau_{\text{list}}$ (a list of mass exponents, output from Algorithm 3), $q_{\text{list}}$ (a list of distortion factors)
1: Calculate the $\alpha_{\text{list}}$ using Eq.8
    $\alpha_{\text{list}} \leftarrow \frac{d\tau_{\text{list}}}{dq_{\text{list}}}$
2: Calculate the $f(\alpha)_{\text{list}}$ using Eq.9
    $f(\alpha)_{\text{list}} \leftarrow q_{\text{list}} \cdot \alpha_{\text{list}} - \tau_{\text{list}}$
3: $\alpha_0 \leftarrow \alpha_{\text{list}}$ [index of maximum value in $f(\alpha)_{\text{list}}$]
4: width $\leftarrow \max(\alpha_{\text{list}}) - \min(\alpha_{\text{list}})$
5: **return** $(\alpha_0, \text{width}, \alpha_{\text{list}}, f(\alpha)_{\text{list}})$

---

### B.4    TIME COMPLEXITY OF NEUROMFA

Here are the full expressions for each algorithm's complexity:

- Algorithm 2: $O(|S| \cdot |S[s]| + |R| \log |R| + |S| \cdot |R| + |q_{list}| \cdot |S|)$
- Algorithm 3: $O(|Z_{qlist}| \cdot |r_{sorted}|)$
- Algorithm 4: $O(|\tau_{list}|)$

We simplify these expressions based on certain considerations:

- $|\tau_{list}|$ and $|Z_{qlist}|$ are equal to $|q_{list}|$, which is a predefined constant.
- $|r_{sorted}|$ is equal to $|R|$. Since $R$ represents positive integers and we set an upper limit for computational convenience, so $|R|$ becomes a constant.

After applying these simplifications:

- Algorithm 2: $O(|S| \cdot |S[s]|)$
- Algorithm 3: $O(1)$
- Algorithm 4: $O(1)$

To connect Algorithm 2's complexity to neural network parameters, we consider:

- $L$: number of layers
- $D$: average neurons per layer
- $N$: number of neurons of the whole network
- $\alpha$: sampling ratio

The number of sampled nodes $|S| = L \cdot \alpha \cdot D$ and each sampled node considers only nodes in the next layer, so $|S[s]| = \alpha \cdot D$. Therefore, Algorithm 2's complexity is expressed as:

$$O(|S| \cdot |S[s]|) = O((L \cdot \alpha \cdot D) \cdot (\alpha \cdot D)) = O(\alpha^2 \cdot L \cdot D^2)$$

Since $\alpha$ is a constant, we simplify to:

$$O(L \cdot D^2) = O(N \cdot D)$$

This final expression represents the overall time complexity of NeuroMFA, as it is dominated by Algorithm 2. This analysis shows that NeuroMFA's complexity scales linearly with the number of layers ($L$) and quadratically with the average number of neurons per layer ($D$).

### B.5    NETWORK CONSTRUCTION

**Shortest Path Calculation.** The time complexity of the algorithm of calculating the shortest paths among each node in different layers is $O(kn^2)$, here $k$ denotes how many layers we crossed, and $n$ denotes the number of nodes in each layer. This algorithm calculates the passing closure with a limitation of steps to calculate the shortest paths with an allowance of $k$ layers. Here since the network is a leverage graph, we can reduce the number of nodes we need to compute in each iteration to the number of nodes in each layer. The algorithm is as follows:

However, the time cost of the calculation of shortest paths in neural networks is unacceptable, because in large language models, the total number of nodes and edges is extremely large. So in real-world situation we need to sample some nodes from each layer to build a manageable network for analysis. We will explain the sample and shortest path estimation in the Appendix.

Even though we have already sampled nodes in each layer, calculating the shortest paths across multiple layers is still very expensive. The complexity of what we mentioned in Section 4 is $O(kn^2)$. Here the $k$ is the number of layers we cross and $n$ is the number of nodes in each layer. This is a

---

**Algorithm 5** Calculation of the Precise Shortest-Path in Neural Networks

---

**Require:** $Dis$ (A map from node pairs $(u, v)$ to distance value $w$), $Nodes[2...N]$ (An array containing nodes in each layer),$N$ (Number of layers)
1: **for all** $x$ in $Nodes[0]$ **do**
2:     **for** $l = 2$ to $N$ **do**
3:         **for all** $y$ in $Nodes[l]$ **do**
4:             **for all** $i$ in $Nodes[l-1]$ **do**
5:                 $Dis[x, y] \leftarrow \min\{Dis[x, y], Dis[x, i] + Dis[i, y]\}$
6:             **end for**
7:         **end for**
8:     **end for**
9: **end for**
10: **return** $Dis$

---

modified version of Floyd-Warshall algorithm calculating the passing closure of the whole graph. However, since the number of each layer is still very large, it is still too expensive to calculate the distance. Here we cannot calculate the distance on the sampled subgraph as it will be too inaccurate.

To address this problem, we use a binary search based method to estimate the shortest paths among nodes. We will give a budget to sample a passing node and calculate the distance passing by this node at every binary checking. The historical biggest value is also recorded to accumulate historical trials. The detailed process of the algorithm can be found at Algorithm 6.

**Weight Matrices in Transformers.** In our network construction of transformer attention mechanisms, we focus specifically on the static weight matrices ($W_Q$, $W_K$, and $W_V$) rather than the dynamic attention weights that are generated during inference. While the interaction between query and key matrices produces dynamic attention weights that vary with input data, these static weight matrices represent the model's fundamental learned parameters and therefore better reflect the inherent structural properties of the trained model.

---

**Algorithm 6** Calculation of the Estimated Shortest-Path in Neural Networks

---

**Require:** $Dis$ (A map from node pairs $(u, v)$ to distance value $w$), $Nodes[2...N]$ (An array containing nodes in each layer), $N$ (Number of layers)
1: $Nodes \leftarrow$ Sample($Nodes$)
2: **for all** $x$ in $Nodes[0]$ **do**
3:     **for** $l = 2$ to $N$ **do**
4:         **for all** $y$ in $Nodes[l]$ **do**
5:             $l \leftarrow 0$
6:             $r \leftarrow \inf$
7:             $hisest \leftarrow 0$
8:             $mid \leftarrow 0$
9:             **while** $l < r$ **do**
10:                 $mid \leftarrow \frac{l+r}{2}$
11:                 $IntNodes \leftarrow$ Sample($Nodes[l-1]$)
12:                 $est \leftarrow$ ShortestAmong($IntNodes$)
13:                 $hisest, est \leftarrow \min(est, hisest)$
14:                 **if** $est \leq mid$ **then**
15:                     $r \leftarrow mid$
16:                 **else**
17:                     $l \leftarrow mid$
18:                 **end if**
19:             **end while**
20:             $dis[x, y] \leftarrow mid$
21:         **end for**
22:     **end for**
23: **end for**
24: **return** $Dis$

---

## B.6 Benchmarks description

In this subsection we introduce the metrics used to evaluate the Pythia model's performance. Here the traditional metrics are all testing-based metrics, each focusing on different aspects.

- **Lambada-OpenAI**. LAMBADAPaperno et al. (2016) is a specialized metric developed by OpenAI to evaluate the capabilities of Large Language Models (LLMs). It stands for "Language Modeling Broadened to Account for Discourse Aspects." This metric is designed to assess an LLM's ability to understand and generate text within the context of broader discourse, going beyond simple sentence-level understanding. LAMBADA specifically focuses on testing the model's proficiency in handling long-range dependencies and context in text. This is achieved through a set of challenging tasks that require the model to make predictions or generate responses based on extended passages of text, rather than just individual sentences or short snippets. By employing LAMBADA, researchers can gain deeper insights into an LLM's understanding of complex linguistic structures and its capacity to maintain coherence over longer stretches of discourse, which are critical for more advanced natural language processing applications.

- **PIQA**. PIQA (Physical Interaction: Question Answering)Bisk et al. (2020) is a dataset created for assessing commonsense reasoning in natural language processing (NLP) models, particularly focusing on their understanding of physical knowledge. PIQA challenges these systems with questions that require physical commonsense, such as choosing the most sensible physical action among various options. While humans exhibit high accuracy (95%) in answering these questions, pretrained models like BERT struggle, achieving around 77% accuracy. This dataset exposes the gap in AI systems' ability to reliably answer physical commonsense questions and provides a vital benchmark for advancing research in natural language understanding, especially in the realm of physical knowledge.

- **WinGrande**.WinoGrandeSakaguchi et al. (2021) is a large-scale dataset designed to evaluate neural language models' commonsense reasoning capabilities. Comprising 44,000 problems and inspired by the original Winograd Schema Challenge, WinoGrande was developed through a meticulous crowdsourcing process and systematic bias reduction using the AfLite algorithm. This dataset addresses the overestimation of machine commonsense in previous benchmarks by reducing dataset-specific biases and providing a more challenging set of problems. While state-of-the-art methods on WinoGrande achieve accuracy rates between 59.4% and 79.1%, they fall short of human performance at 94%, highlighting the gap in true commonsense reasoning capabilities of AI models. WinoGrande not only serves as a crucial benchmark for transfer learning but also underscores the importance of algorithmic bias reduction in evaluating machine commonsense.

- **WSC**. The Winograd Schema Challenge (WSC)Kocijan et al. (2020) is a test of artificial intelligence that focuses on evaluating a system's ability to perform commonsense reasoning. WSC consists of a series of questions based on Winograd schemes, pairs of sentences that differ in only one or two words and contain a highly ambiguous pronoun. The challenge requires deep understanding of text content and the situations described to resolve these pronouns correctly. Initially containing 100 examples constructed manually by AI experts, the dataset has since expanded to 285 examples, with the WSC273 variant often used for consistency in model evaluations. Despite its initial design to be difficult for machines, recent advances in AI, such as the BERT and GPT-3 models, have achieved high levels of accuracy on the WSC, raising questions about the true extent of their commonsense reasoning capabilities. The challenge highlights the importance of knowledge and commonsense reasoning in AI and serves as a benchmark for evaluating progress in natural language understanding.

- **ARC**.The AI2 Reasoning Challenge (ARC)Clark et al. (2018) dataset, introduced by Clark et al., is a multiple-choice question-answering dataset featuring questions sourced from science exams spanning grades 3 to 9. ARC is divided into two sections: the Easy Set and the Challenge Set, with the latter containing more difficult questions that necessitate higher levels of reasoning. The dataset primarily consists of questions with four answer choices, although a small percentage have either three or five options. Accompanying ARC is a supporting knowledge base (KB) of 14.3 million unstructured text passages. This dataset is used to assess the reasoning capabilities of language models, particularly in the

context of science and general knowledge. Models like GPT-4 have shown remarkable performance on the ARC dataset, reflecting advancements in AI's ability to handle complex question-answering tasks that require a blend of knowledge retrieval and reasoning skills.

- **SciQ**. The SciQ Welbl et al. (2017) dataset, developed by the Allen Institute for Artificial Intelligence (AI2), consists of 13,679 crowdsourced science exam questions covering subjects such as Physics, Chemistry, and Biology. These questions are formatted as multiple-choice queries, each offering four answer options. A distinctive feature of the SciQ dataset is that for the majority of its questions, an additional paragraph is provided, offering supporting evidence for the correct answer. This dataset is a valuable tool for evaluating language models' ability to perform in reading comprehension, question generation, and understanding complex scientific content. It challenges models to not only select the correct answer from multiple choices but also to utilize supporting evidence effectively, thereby testing their comprehension and reasoning skills in the scientific domain.

- **LogiQA**. LogiQALiu et al. (2020) is a dataset consisting of 8,678 QA instances that focus on evaluating machine reading comprehension with an emphasis on logical reasoning. It is derived from the National Civil Servants Examination of China and covers various types of deductive reasoning. This dataset presents a significant challenge for state-of-the-art neural models, which perform notably worse than humans in these tasks. LogiQA serves as a unique benchmark for testing logical AI under deep learning NLP environments, requiring models to demonstrate a blend of language understanding and complex logical reasoning. It includes different types of logical problems, such as categorical reasoning, sufficient and necessary conditional reasoning, disjunctive reasoning, and conjunctive reasoning, all key to deductive reasoning. This dataset provides a rigorous test of AI's logical reasoning capabilities and its ability to handle problems similar to those faced by human experts.

- **HendrycksTest**. The HendrycksTestHendrycks et al. (2020), also known as the Measuring Massive Multitask Language Understanding (MMLU) test, is a massive multitask test that includes multiple-choice questions from a wide range of knowledge domains, covering 57 tasks in areas such as elementary mathematics, US history, computer science, and law. This test aims to measure the multitask accuracy of text models, requiring them to demonstrate extensive world knowledge and problem-solving ability. The results show that while most recent models have near random-chance accuracy, larger models like GPT-3 have shown improvement, but still fall short of expert-level accuracy across all tasks. The HendrycksTest serves as a comprehensive tool for evaluating the breadth and depth of models' academic and professional understanding, identifying significant shortcomings, and highlighting areas needing substantial improvement, especially in socially important subjects like morality and law.

## C IMPACT STUDY

### C.1 VARYING ARCHITECTURE

Although in the area of computer vision (CV), several generative models (Rombach et al., 2022) providing personalized image generation exhibit emergent abilities, no analysis of their emergence exists. Thus, we apply NeuroMFA to the diffusion model. The experimental results are shown in Appendix C.1, Fig. 13. The degree of emergence for the diffusion model (CV task) is lower than the LLM (NLP task) at the same parameter size, which indicates a lower level of emergent abilities.

We extend our analysis to CNNs, testing both ResNet-18 and ResNet-152 models. As shown in Appendix Fig. 14, ResNe-t18's spectral graph lacks a regular bell-shaped structure, indicating no clear multifractal structure or self-organization. While ResNet-152 displays a noticeable multifractal structure, its spectral graph shows irregular horizontal shifts, suggesting that a stable self-organized structure did not form. These results indicate that CNN structures alone may not produce emergent abilities comparable to transformer-based models. Our findings from both CNN-based and transformer-based models, presented in Appendix C.1, provide preliminary validation for applying multifractal and self-organization research to study model emergent abilities. The contrasting results between these architectures highlight the potential of our method in differentiating and understanding various model structures in terms of their capacity for emergent abilities.

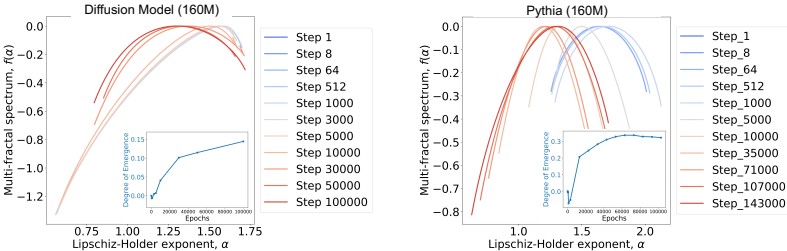

Figure 13: NeuroMFA results on stable diffusion model and the Pythia model of equivalent size.

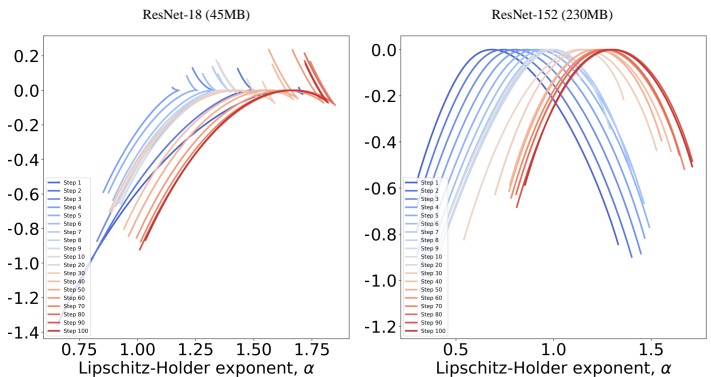

Figure 14: NeuroMFA results on ResNet-18 model and the ResNet-152 model.

The analysis results for different training epochs on a Computer Vision (CV) task using the Diffusion Model are depicted in Fig. 13 (Left). On the right, analysis outcomes for a LLM of equivalent size (in terms of the number of parameters) are presented. Furthermore, we extend the NeuroMFA analysis to two typical CV models: ResNet-18 and ResNet-152, as shown in Fig. 14.

### C.2 VARYING TRAINING DATA

We explore the potential impact of different types of training data on neuron interaction dynamics. Our study primarily focuses on Pythia-deduped models, but we also conduct comparative experiments

using both Pythia-standard and Pythia-deduped models to explore the impact of different training datasets, whose results are shown in Appendix C.2. The Pythia project offers two types of models with identical architectures but trained on different datasets: Pythia-standard models (trained on a 300B token dataset) and Pythia-deduped models (trained on a 207B token dataset with duplicates removed). Applying our NeuroMFA method to both model types reveals that the critical point for emergence occurs at the same training step. However, the Pythia-standard model shows a slightly higher rate of increase in the degree of emergence after this transition. This aligns with Pythia's official statement that the effect of training a model for one epoch on the standard dataset is approximately equivalent to training for 1.5 epochs on the deduped dataset. These findings suggest that our method can capture the influence of different datasets on model training, reflecting how the nature of training data affects self-organization and emergence phenomena in neural networks.

Fig. 15 illustrates the degree of emergence of two sizes of Pythia models trained on different data by using our NeuroMFA method.

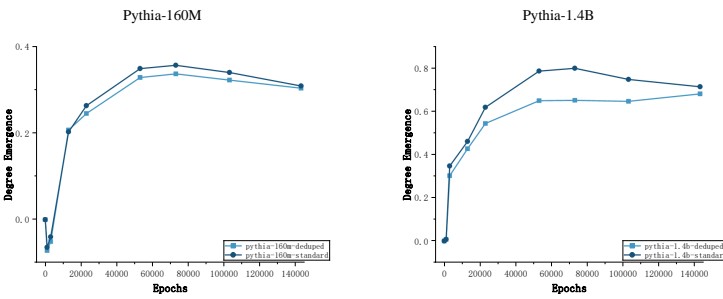

Figure 15: Emergence of Pythia 160M model and Pythia 1.4B model trained on two different data.

## C.3 VARYING TRAINING DIRECTIONS

To validate our proposed metric's effectiveness in capturing model performance changes, we conduct experiments using contaminated training data with BERT. We create a controlled experimental setup by deliberately contaminating the training dataset, combining 20% of Wikipedia dumps data with 80% randomly generated data following a normal distribution.

Using a pre-trained BERT-large-uncased model from HuggingFace (24 layers, 336M parameters, 1024 hidden dimensions, 16 attention heads), we conduct fine-tuning experiments with this contaminated dataset. To evaluate model performance, we employ two widely-used metrics: SQUAD 1.1 F1 score for reading comprehension and Multi NLI Accuracy for natural language inference.

As shown in Figure 16, our proposed metric effectively captures the pattern of model ability degradation caused by severe data contamination, highlighting its capability to track structural changes in the model under different training scenarios. Moreover, this consistency further validates that our structure-based metric can reflect the evolution of emergent properties in LLMs throughout the training process, regardless of training quality or the direction of change.

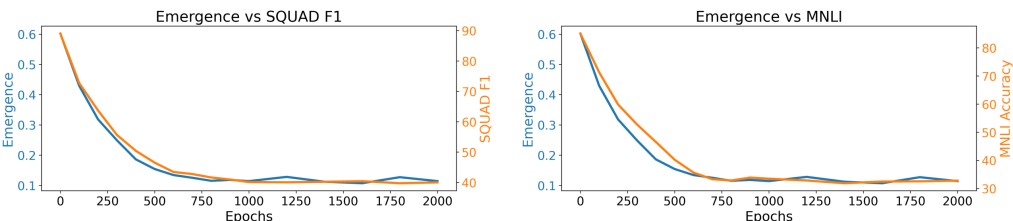

Figure 16: Changes in the degree of emergence during training with contaminated data.

# D    ADDITIONAL RESULTS

## D.1    NEUROMFA SPECTRA WITH VARIANCE

Figure 17 displays the averaged (with variation) multifractal spectra for each model, derived from analyses across 10 SNINs under NeuroMFA. We provide how the $\alpha_0$ and $width$ vary with the training steps in Fig. 18.

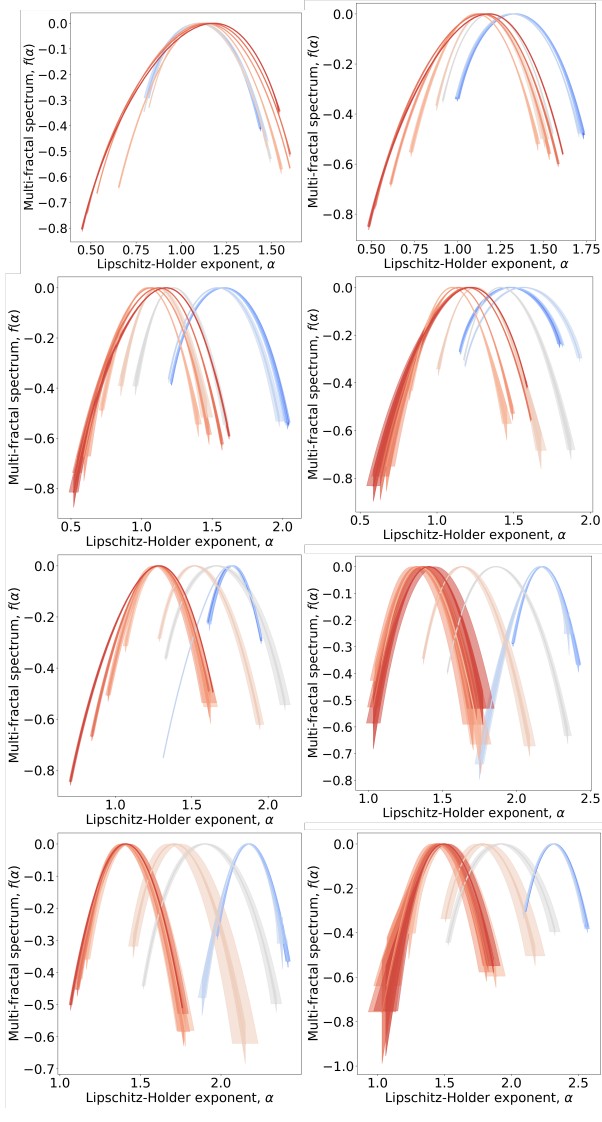

Figure 17: Multifractal analysis spectra of 10 Sampled Neuronal Interaction Networks (SNINs), illustrating the average value of the spectra with shadows indicating the standard deviation. From upper left to lower right, the model sizes are 14M, 31M, 70M, 160M, 410M, 1B, 1.4B, and 2.8B respectively.

Here, we also provide the variance across 10 sampled NINs. In Table 3 and Table 4 below, we present the variance of $\alpha$ and $f(\alpha)$ from 10 samples of the spectra shown in Fig. 5. Furthermore, in Table 5 and Table 6, we present the variance for both the regularity metric ($\alpha_0$) and the heterogeneity metric ($width$).

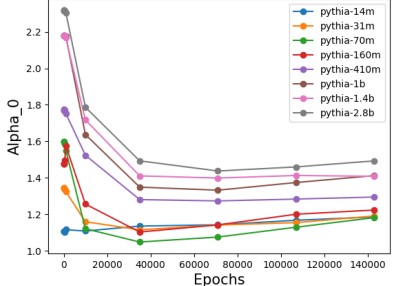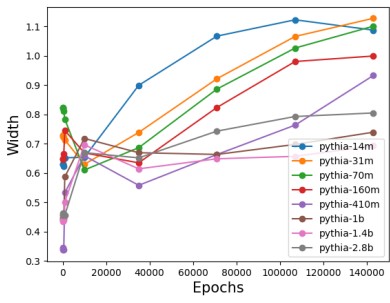

Figure 18: $\alpha_0$ (left) and width $w$ (right) of each epoch in different models.

| Epochs
Model | | 1 | 8 | 64 | 512 | 1000 | 5000 |
|---|---|---|---|---|---|---|---|
| pythia-14m | $\alpha$ | 4.76e-06 | 1.89e-06 | 2.47e-06 | 1.10e-06 | 1.05e-06 | 4.60e-06 |
| | $f(\alpha)$ | 9.41e-06 | 5.68e-06 | 5.42e-06 | 3.86e-06 | 2.12e-06 | 2.24e-06 |
| pythia-31m | $\alpha$ | 5.19e-05 | 5.11e-05 | 3.71e-05 | 6.07e-05 | 5.95e-05 | 3.45e-05 |
| | $f(\alpha)$ | 4.69e-05 | 3.05e-05 | 3.61e-05 | 5.32e-05 | 3.83e-05 | 2.53e-05 |
| pythia-70m | $\alpha$ | 3.90e-06 | 2.35e-05 | 4.96e-05 | 6.71e-05 | 9.35e-06 | 4.88e-05 |
| | $f(\alpha)$ | 6.47e-06 | 1.60e-05 | 5.30e-05 | 6.96e-05 | 1.28e-05 | 4.31e-05 |
| pythia-160m | $\alpha$ | 4.55e-05 | 9.21e-06 | 1.14e-05 | 1.14e-04 | 4.70e-06 | 1.10e-04 |
| | $f(\alpha)$ | 1.49e-05 | 1.74e-05 | 2.18e-05 | 3.17e-05 | 9.40e-06 | 2.10e-05 |
| pythia-410m | $\alpha$ | 3.12e-06 | 7.38e-05 | 6.86e-05 | 6.21e-05 | 2.34e-06 | 4.38e-05 |
| | $f(\alpha)$ | 3.61e-06 | 1.11e-05 | 8.61e-06 | 1.62e-05 | 3.75e-06 | 6.59e-06 |
| pythia-1b | $\alpha$ | 2.14e-05 | 2.83e-05 | 2.42e-05 | 3.03e-05 | 3.84e-05 | 7.84e-05 |
| | $f(\alpha)$ | 2.23e-05 | 4.90e-05 | 3.67e-05 | 3.85e-05 | 6.23e-05 | 1.12e-04 |
| pythia-1.4b | $\alpha$ | 3.99e-05 | 4.57e-05 | 3.98e-05 | 2.01e-05 | 2.18e-05 | 3.69e-05 |
| | $f(\alpha)$ | 4.19e-05 | 5.21e-05 | 3.91e-05 | 2.32e-05 | 4.22e-05 | 4.18e-05 |
| pythia-2.8b | $\alpha$ | 5.20e-06 | 4.19e-05 | 5.79e-05 | 3.20e-05 | 3.75e-05 | 5.47e-05 |
| | $f(\alpha)$ | 7.56e-06 | 2.19e-05 | 2.76e-05 | 2.34e-05 | 1.94e-05 | 2.83e-05 |

Table 3: Variance of $\alpha$ and $f(\alpha)$ in the first six epochs

| Epochs
Model | | 10000 | 35000 | 71000 | 107000 | 143000 |
|---|---|---|---|---|---|---|
| pythia-14m | $\alpha$ | 1.03e-04 | 6.58e-05 | 1.03e-05 | 8.35e-06 | 5.03e-06 |
| | $f(\alpha)$ | 2.28e-05 | 2.48e-05 | 3.19e-05 | 2.48e-05 | 2.29e-05 |
| pythia-31m | $\alpha$ | 5.48e-06 | 3.56e-06 | 8.55e-06 | 6.22e-06 | 1.01e-05 |
| | $f(\alpha)$ | 1.07e-05 | 7.16e-06 | 2.29e-05 | 1.46e-05 | 2.28e-05 |
| pythia-70m | $\alpha$ | 3.75e-05 | 5.72e-06 | 2.02e-04 | 1.78e-04 | 1.92e-04 |
| | $f(\alpha)$ | 2.29e-05 | 3.17e-06 | 8.42e-05 | 8.36e-05 | 2.45e-04 |
| pythia-160m | $\alpha$ | 6.84e-05 | 6.18e-05 | 1.40e-04 | 1.47e-04 | 1.34e-04 |

| Model \ Epochs | | 10000 | 35000 | 71000 | 107000 | 143000 |
|---|---|---|---|---|---|---|
| pythia-410m | $f(\alpha)$ | 1.28e-05 | 7.15e-06 | 1.96e-05 | 2.98e-05 | 2.09e-04 |
| | $\alpha$ | 6.00e-06 | 6.27e-05 | 3.57e-05 | 1.67e-04 | 7.35e-04 |
| pythia-1b | $f(\alpha)$ | 2.15e-05 | 5.12e-05 | 1.05e-04 | 7.49e-04 | 1.08e-04 |
| | $\alpha$ | 1.92e-04 | 2.38e-04 | 8.94e-04 | 2.52e-04 | 3.25e-04 |
| pythia-1.4b | $f(\alpha)$ | 1.44e-03 | 3.76e-03 | 5.56e-03 | 4.39e-03 | 3.09e-03 |
| | $\alpha$ | 1.07e-03 | 7.00e-04 | 2.62e-04 | 3.15e-04 | 1.88e-04 |
| pythia-2.8b | $f(\alpha)$ | 1.17e-03 | 2.22e-03 | 1.97e-03 | 1.61e-03 | 1.24e-03 |
| | $\alpha$ | 4.34e-03 | 1.85e-03 | 9.02e-04 | 9.56e-04 | 8.59e-04 |
| | $f(\alpha)$ | 4.74e-03 | 7.56e-04 | 6.77e-03 | 8.09e-03 | 8.00e-03 |

Table 4: Variance of $\alpha$ and $f(\alpha)$ in the last five epochs

| Model \ Epochs | | 1 | 8 | 64 | 512 | 1000 | 5000 |
|---|---|---|---|---|---|---|---|
| pythia-70m | $\alpha_0$ | 1.12e-05 | 1.12e-05 | 9.66e-05 | 2.73e-04 | 2.67e-05 | 8.31e-05 |
| | $width$ | 4.21e-04 | 4.27e-04 | 3.40e-04 | 5.77e-04 | 4.98e-04 | 1.41e-03 |
| pythia-160m | $\alpha_0$ | 9.72e-05 | 1.21e-04 | 3.87e-05 | 3.60e-04 | 3.07e-04 | 1.37e-04 |
| | $width$ | 1.18e-04 | 1.16e-04 | 9.18e-05 | 2.40e-04 | 4.02e-04 | 7.24e-04 |
| pythia-410m | $\alpha_0$ | 9.41e-05 | 9.41e-05 | 1.04e-04 | 9.37e-05 | 3.76e-04 | 4.61e-04 |
| | $width$ | 1.65e-04 | 1.65e-04 | 1.95e-04 | 4.17e-04 | 2.27e-03 | 3.05e-04 |
| pythia-1b | $\alpha_0$ | 2.09e-05 | 2.19e-05 | 3.67e-05 | 2.18e-05 | 7.47e-04 | 2.34e-04 |
| | $width$ | 2.16e-04 | 2.15e-04 | 2.28e-04 | 6.56e-05 | 2.62e-03 | 3.12e-04 |
| pythia-1.4b | $\alpha_0$ | 3.07e-05 | 1.36e-05 | 1.30e-05 | 1.78e-05 | 1.53e-05 | 3.42e-04 |
| | $width$ | 3.36e-04 | 3.36e-04 | 3.32e-04 | 1.93e-04 | 1.38e-03 | 1.63e-04 |
| pythia-2.8b | $\alpha_0$ | 2.42e-05 | 5.81e-05 | 8.53e-05 | 5.70e-05 | 3.24e-05 | 6.81e-04 |
| | $width$ | 4.78e-04 | 4.78e-04 | 4.84e-04 | 4.19e-04 | 1.21e-04 | 1.00e-04 |

Table 5: Variance of $\alpha_0$ and $width$ in the first six epochs

| Model \ Epochs | | 10000 | 35000 | 71000 | 107000 | 143000 |
|---|---|---|---|---|---|---|
| pythia-70m | $\alpha_0$ | 8.32e-05 | 5.01e-05 | 1.26e-05 | 1.26e-04 | 2.38e-05 |
| | $width$ | 1.57e-03 | 1.11e-03 | 3.71e-03 | 1.95e-03 | 2.67e-03 |
| pythia-160m | $\alpha_0$ | 4.11e-05 | 1.48e-04 | 2.29e-05 | 5.83e-06 | 7.27e-05 |
| | $width$ | 2.08e-03 | 1.53e-03 | 4.43e-03 | 2.72e-03 | 1.57e-03 |
| pythia-410m | $\alpha_0$ | 1.63e-04 | 4.14e-04 | 8.14e-04 | 3.92e-04 | 5.17e-05 |
| | $width$ | 8.36e-04 | 1.47e-03 | 1.71e-03 | 8.93e-04 | 2.58e-04 |
| pythia-1b | $\alpha_0$ | 1.92e-04 | 2.38e-04 | 8.94e-04 | 2.52e-04 | 3.25e-04 |
| | $width$ | 1.44e-03 | 3.76e-03 | 5.56e-03 | 4.39e-03 | 3.09e-03 |

| Epochs
Model | | 10000 | 35000 | 71000 | 107000 | 143000 |
|---|---|---|---|---|---|---|
| pythia-1.4b | $\alpha_0$ | 1.07e-03 | 7.00e-04 | 2.62e-04 | 3.15e-04 | 1.88e-04 |
| | $width$ | 1.17e-03 | 2.22e-03 | 1.97e-03 | 1.61e-03 | 1.24e-03 |
| pythia-2.8b | $\alpha_0$ | 4.34e-03 | 1.85e-03 | 9.02e-04 | 9.56e-04 | 8.59e-04 |
| | $width$ | 4.74e-03 | 7.56e-04 | 6.77e-03 | 8.09e-03 | 8.00e-03 |

Table 6: Variance of $\alpha_0$ and $width$ in the last five epochs

## D.2  ASSESSMENT OF THE DEGREE OF EMERGENCE ACROSS VARIOUS DATASETS

Fig. 23, Fig. 20, Fig. 22, Fig. 19, and Fig. 21 show the result for degree of emergence under our metric.

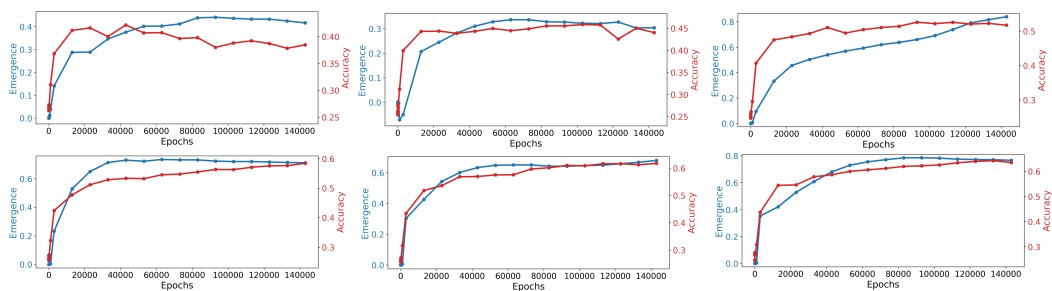

Figure 19: Results on ARC-Easy benchmark. From upper left to lower right, the model sizes are 70M, 160M, 410M, 1B, 1.4B, and 2.8B respectively.

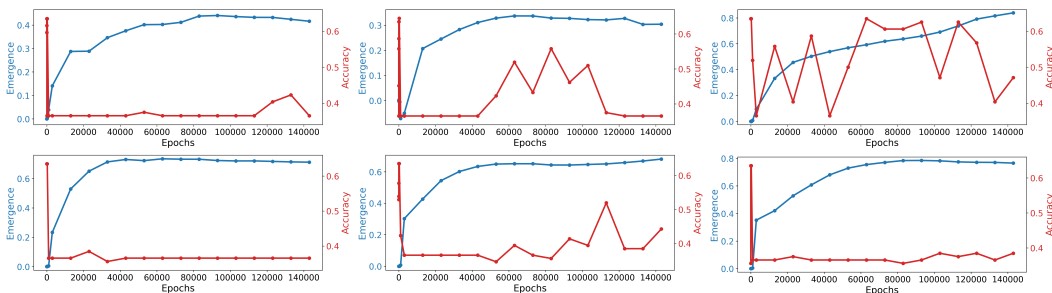

Figure 20: Results on WSC benchmark. From upper left to lower right, the model sizes are 70M, 160M, 410M, 1B, 1.4B, and 2.8B respectively.

Graphs with all metrics are shown in Fig. 24. (a) and (b) represent the metric performance of Pythia-1B and Pythia-1.4B, respectively. Large language models (LLMs) can be evaluated comprehensively using metrics A-H. The seven specific metrics A-H can be categorized as follows: LAMBADA (A) assesses language understanding; SciQ (B) measure scientific knowledge; LogiQA (C) evaluates logical reasoning; ARC-challenge (D), ARC-easy (H) and PIQA (G) examines physical commonsense reasoning; WSC (E) and Winogrande (F) test commonsense reasoning. Detailed performance data are provided in Tables 8 and 7, respectively.

## D.3  LOG-LOG RELATIONSHIP EVALUATION

We use $R^2$ (the coefficient of determination) to evaluate the log-log relationship between $N(r)$ and $r$, between $Z_q(r)$ and $\frac{r}{d}$ at different samples of SNIN. The detailed evaluation results can be found in Table 9.

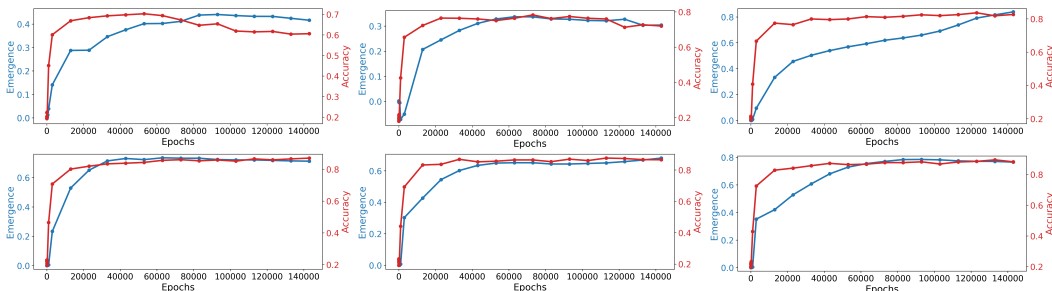

Figure 21: Results on SciQ benchmark. From upper left to lower right, the model sizes are 70M, 160M, 410M, 1B, 1.4B, and 2.8B respectively.

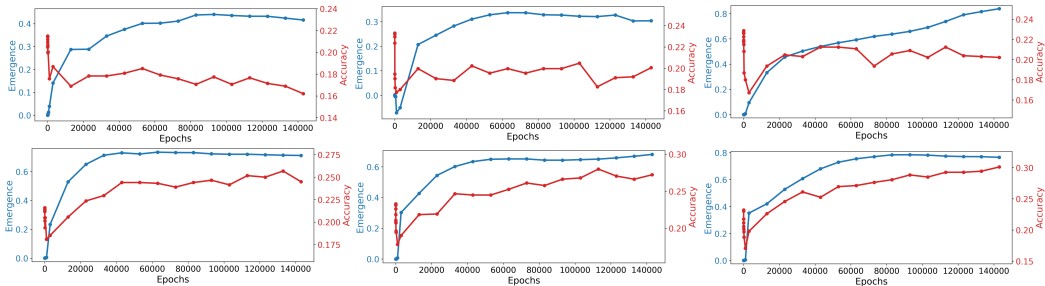

Figure 22: Results on ARC-Challenge benchmark. From upper left to lower right, the model sizes are 70M, 160M, 410M, 1B, 1.4B, and 2.8B respectively.

According to the coefficient of determination detailed in Appendix E.2, the log-log relationships are all above 0.7, demonstrating strong log-log relationships between $N(r)$ and $r$, between $Z_q(r)$ and $\frac{r}{d}$.

### D.4 AVERAGE WEIGHTED DEGREE DISTRIBUTION

Previous work has reported the existence of heavy-tail degree distributions in the node connectivity of neural networks, which suggests the presence of hubs, playing an important role in the network (Barabási & Bonabeau, 2003; Broido & Clauset, 2019). These phenomena are commonly considered as outcomes of self-organization (Park et al., 2005; Evans & Saramäki, 2005).

In our work, we first construct NINs for two versions of the pythia model: a small-scale 14M parameter model (pythia 14M) and a larger 1.4B parameter model (pythia 1.4B). Dealing with NINs, we compute the weighted degree of each node, defined as the average value of distances from connections. This allows us to analyze the average weighted degree distributions that emerge during training.

Figure 25 shows a clear distinction between the two models. During the training process, the average degree distribution for pythia 14M exhibits a dispersed trend, with lower kurtosis values (where kurtosis is commonly used to measure distribution characteristics, with higher kurtosis indicating a heavier-tailed distribution, see Appendix E.1), the distribution for pythia 1.4B with left shifting indicates the trend of appearing a heavy-tail distribution over multiple epochs of pertaining, which is also indicated by the increasing Kurtosis value. Please refer to Fig. 26, Fig. 27, and Fig. 28 for more information and results of Kurtosis value on pythia 14M, 160M, and 1.4B. This indicates that as the scale of the model increases, the average distance from most neurons to the neighbors in the next layer decreases, signifying the establishment of stronger interactions. It might imply that an organizational structure with a heavy-tailed connectivity pattern systematically self-organizes. Therefore, further in-depth research into the network's connectivity structure is required to observe the self-organization process, leading us to the multifractal analysis of the network.

Fig. 26, Fig. 27, and Fig. 28 provide more results on different scales of models.

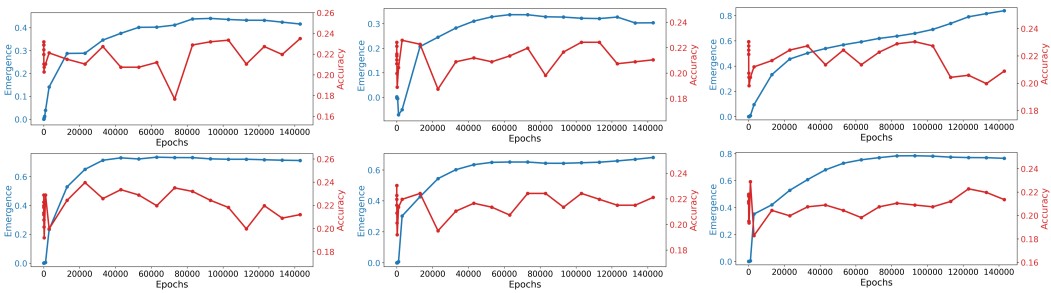

Figure 23: Results on LogiQA benchmark. From upper left to lower right, the model sizes are 70M, 160M, 410M, 1B, 1.4B, and 2.8B respectively.

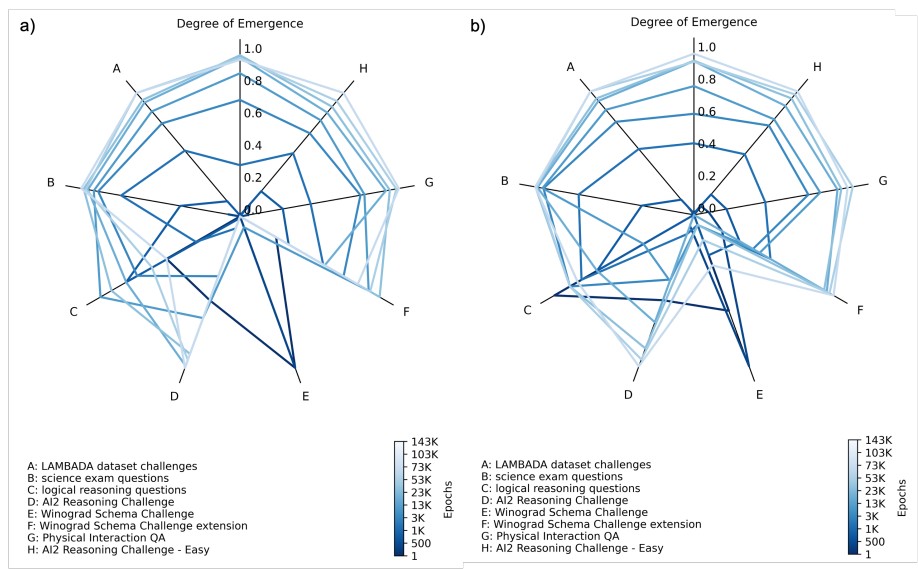

Figure 24: Full radar chart comparing the degree of emergence metric with conventional benchmarks.

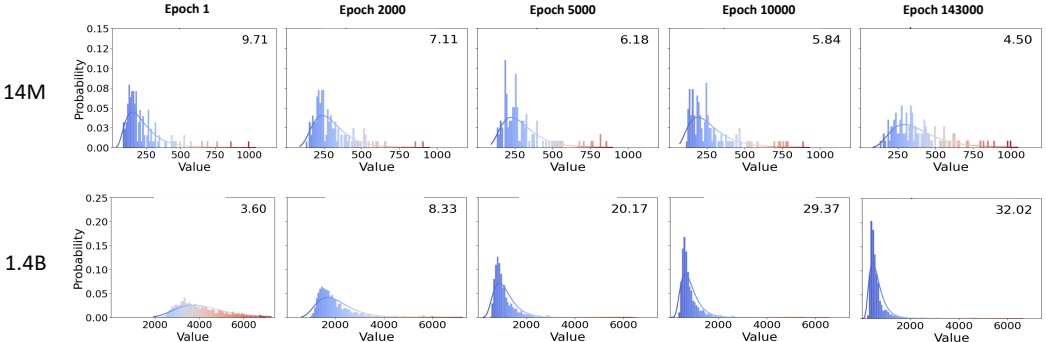

Figure 25: The distributions of average weighted degree for two models of different sizes (i.e., 14M and 1.4B pythia model) during the training process. The numbers in the figures represent Kurtosis, indicating the tail weight and peak sharpness of the distributions. High kurtosis implies heavier tails and a sharper peak, whereas low kurtosis suggests lighter tails and a flatter peak.

| Epochs | 0 | 512 | 1000 | 3000 | 13000 | 23000 | 53000 | 73000 | 103000 | 143000 |
|---|---|---|---|---|---|---|---|---|---|---|
| Degree of Emergence | 0.00 | 0.00 | 0.00 | 0.23 | 0.53 | 0.65 | 0.72 | 0.73 | 0.72 | 0.71 |
| WSC | 0.63 | 0.63 | 0.37 | 0.37 | 0.37 | 0.38 | 0.37 | 0.37 | 0.37 | 0.37 |
| LogiQA | 0.21 | 0.18 | 0.23 | 0.20 | 0.22 | 0.24 | 0.23 | 0.24 | 0.22 | 0.21 |
| SciQ | 0.22 | 0.27 | 0.47 | 0.71 | 0.80 | 0.82 | 0.84 | 0.86 | 0.85 | 0.87 |
| LAMBADA | 0.00 | 0.00 | 0.08 | 0.33 | 0.47 | 0.53 | 0.57 | 0.58 | 0.62 | 0.62 |
| ARC-easy | 0.26 | 0.29 | 0.32 | 0.42 | 0.48 | 0.51 | 0.53 | 0.55 | 0.56 | 0.58 |
| ARC-challenge | 0.22 | 0.18 | 0.18 | 0.19 | 0.21 | 0.22 | 0.24 | 0.24 | 0.24 | 0.24 |
| PIQA | 0.53 | 0.54 | 0.57 | 0.60 | 0.66 | 0.66 | 0.69 | 0.69 | 0.70 | 0.70 |
| Winogrande | 0.49 | 0.47 | 0.49 | 0.51 | 0.52 | 0.53 | 0.51 | 0.54 | 0.54 | 0.53 |

Table 7: Detailed performance data of Pythia-1B across different epochs and metrics.

| Epochs | 0 | 512 | 1000 | 3000 | 13000 | 23000 | 53000 | 73000 | 103000 | 143000 |
|---|---|---|---|---|---|---|---|---|---|---|
| Degree of Emergence | 0.00 | 0.00 | 0.01 | 0.30 | 0.43 | 0.54 | 0.65 | 0.65 | 0.65 | 0.68 |
| WSC | 0.53 | 0.63 | 0.42 | 0.37 | 0.37 | 0.37 | 0.35 | 0.37 | 0.39 | 0.44 |
| LogiQA | 0.23 | 0.18 | 0.21 | 0.22 | 0.22 | 0.20 | 0.21 | 0.22 | 0.22 | 0.22 |
| SciQ | 0.23 | 0.24 | 0.44 | 0.69 | 0.83 | 0.84 | 0.86 | 0.86 | 0.86 | 0.87 |
| LAMBADA | 0.00 | 0.00 | 0.08 | 0.33 | 0.47 | 0.53 | 0.57 | 0.58 | 0.62 | 0.62 |
| ARC-easy | 0.26 | 0.28 | 0.32 | 0.43 | 0.52 | 0.54 | 0.57 | 0.60 | 0.61 | 0.62 |
| ARC-challenge | 0.23 | 0.19 | 0.18 | 0.19 | 0.22 | 0.22 | 0.24 | 0.26 | 0.27 | 0.27 |
| PIQA | 0.52 | 0.54 | 0.56 | 0.61 | 0.67 | 0.68 | 0.71 | 0.71 | 0.73 | 0.72 |
| Winogrande | 0.48 | 0.50 | 0.51 | 0.53 | 0.52 | 0.52 | 0.56 | 0.56 | 0.56 | 0.57 |

Table 8: Detailed performance data of Pythia-1.4B across different epochs and metrics.

| Model | Step | $R^2$ of $lgN(r) \sim lgr$ | | $R^2$ of $lgZ_q(r) \sim lg\frac{r}{d}$ | |
|---|---|---|---|---|---|
| | | Mean | Std | Mean | Std |
| pythia-14m | 1 | 0.8249 | 0.0291 | 0.9125 | 0.0018 |
| | 2000 | 0.8308 | 0.0376 | 0.9160 | 0.0026 |
| | 5000 | 0.8370 | 0.0330 | 0.8943 | 0.0047 |
| | 10000 | 0.8421 | 0.0248 | 0.8613 | 0.0118 |
| | 143000 | 0.8948 | 0.0300 | 0.9763 | 0.0012 |
| pythia-160m | 1 | 0.9155 | 0.0594 | 0.9585 | 0.0004 |
| | 2000 | 0.8874 | 0.0419 | 0.9546 | 0.0005 |
| | 5000 | 0.8431 | 0.0195 | 0.9248 | 0.0023 |
| | 10000 | 0.8132 | 0.0168 | 0.8750 | 0.0093 |
| | 143000 | 0.8556 | 0.0391 | 0.9800 | 0.0014 |
| pythia-1.4b | 1 | 0.9005 | 0.0002 | 0.9279 | 0.0024 |
| | 2000 | 0.9364 | 0.0424 | 0.9287 | 0.0020 |
| | 5000 | 0.9112 | 0.0307 | 0.9700 | 0.0011 |
| | 10000 | 0.8805 | 0.0205 | 0.9336 | 0.0035 |
| | 143000 | 0.8558 | 0.0190 | 0.9489 | 0.0018 |

Table 9: Log-log relationship evaluation.

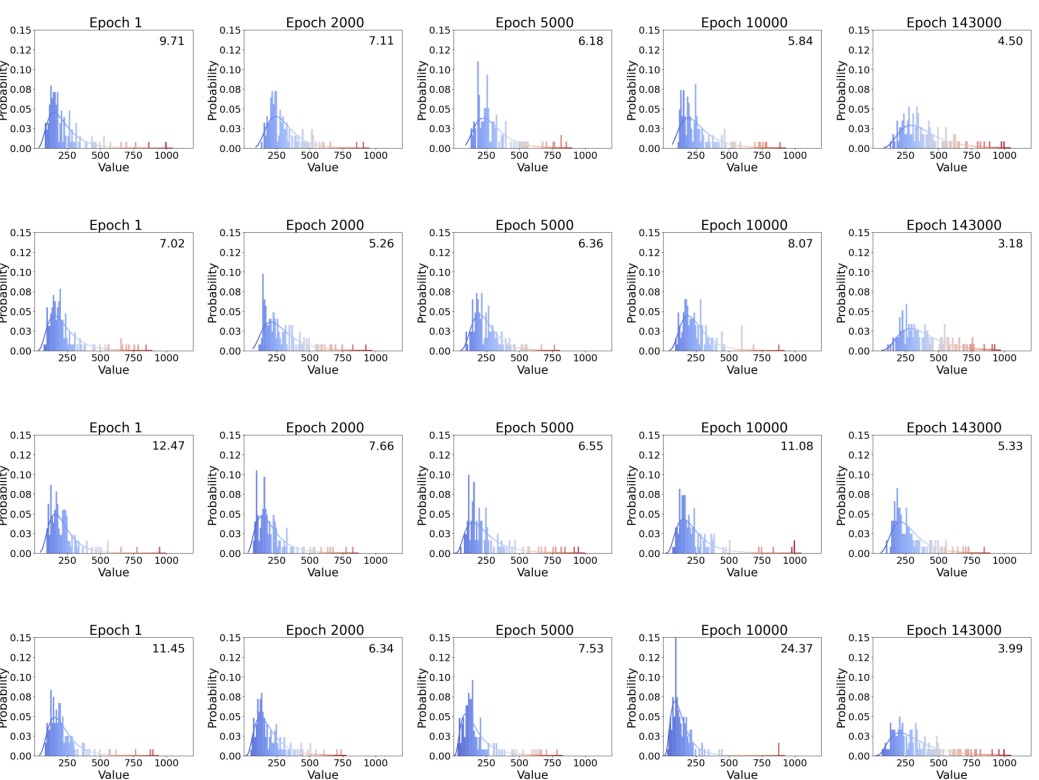

Figure 26: The distributions of average weighted degree for 14M model of different sizes in different layers during the training process.

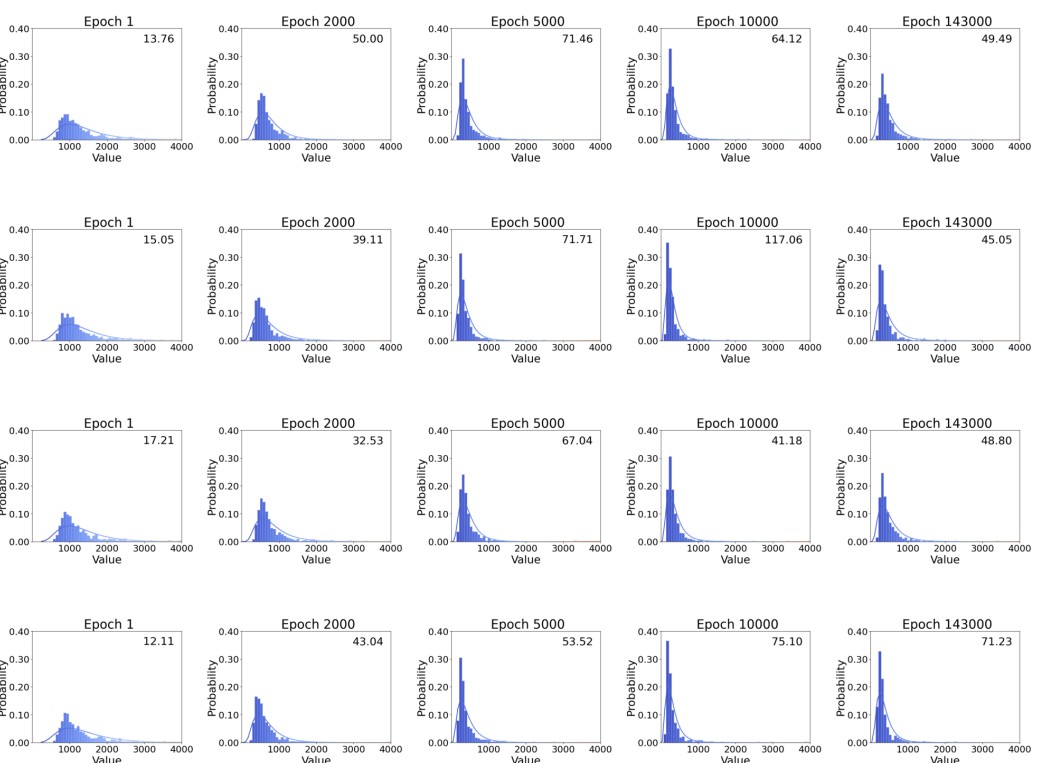

Figure 27: The distributions of average weighted degree for 160M model of different sizes in different layers during the training process.

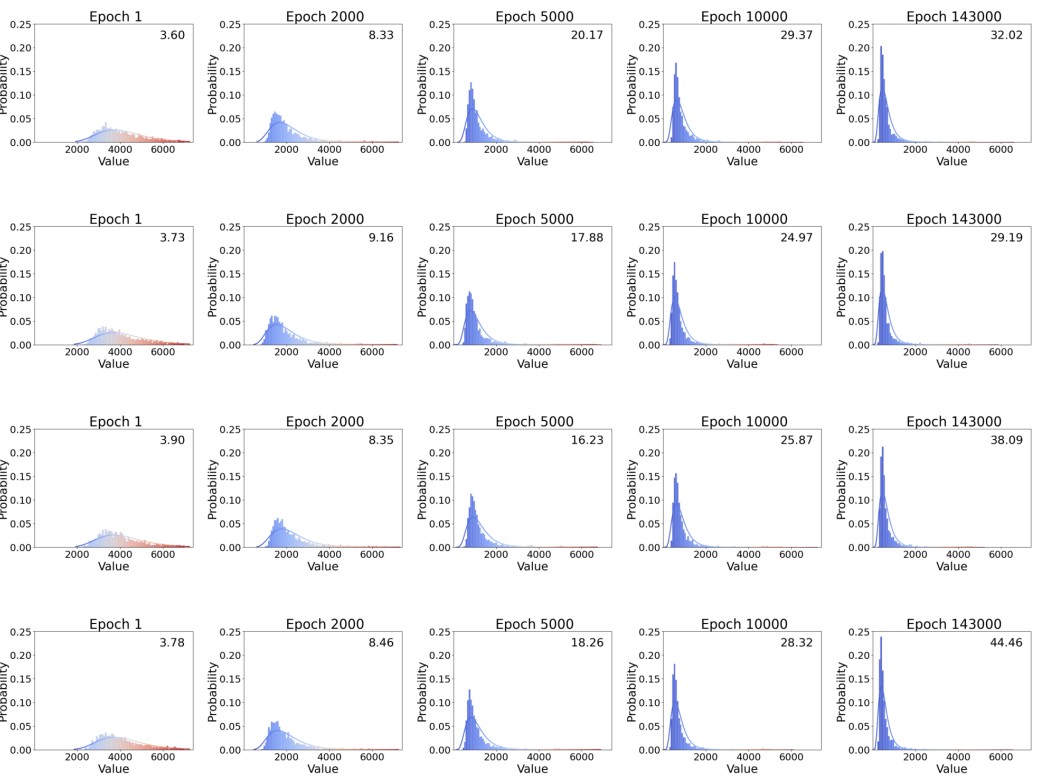

Figure 28: The distributions of average weighted degree for 1.4B model of different sizes in different layers during the training process.

# E    EVALUATION METRICS

## E.1    KURTOSIS ($\beta_2$)

Kurtosis is a measure of the "tailedness" of the probability distribution of a real-valued random variable. It provides insights into the shape of the distribution's tails and peak. High kurtosis in a data set suggests a distribution with heavy tails and a sharper peak (leptokurtic), while low kurtosis indicates a distribution with lighter tails and a more flattened peak (platykurtic). Kurtosis is often compared to that of a normal distribution, which has a kurtosis of 3.

The formula for kurtosis is:

$$\beta_2 = E\left[\left(\frac{X - \mu}{\sigma}\right)^4\right] \tag{46}$$

where the variables represent the same as in the skewness formula.

These statistical measures, skewness ($\gamma_1$) and kurtosis ($\beta_2$), are crucial for quantifying and analyzing the non-Gaussianity in image data. They provide valuable insights into the distribution characteristics of image pixel intensities, particularly in highlighting deviations from the normal distribution.

The higher the kurtosis, the greater the degree of non-Gaussianity in the distribution, indicating a distribution with heavier tails than a normal distribution.

## E.2    COEFFICIENT OF DETERMINATION ($R^2$)

The coefficient of determination, represented by $R^2$, quantifies the extent to which the variance in the dependent variable can be predicted from the independent variable(s) in a regression model. It offers a measure of how well observed outcomes are replicated by the model, based on the proportion of the total variation in outcomes explained by the model.

The formula to calculate $R^2$ is given by:

$$R^2 = 1 - \frac{\sum_i (y_i - \hat{y}_i)^2}{\sum_i (y_i - \bar{y})^2} \tag{47}$$

where $y_i$ denotes the observed values, $\hat{y}_i$ represents the predicted values from the model, and $\bar{y}$ is the mean of the observed values.

$R^2$ is interpreted as follows:

- $R^2 < 0.3$: Weak correlation. The model accounts for a small fraction of the variance in the data.
- $0.3 \leq R^2 < 0.5$: Moderate correlation. The model provides a moderate level of explanation for the data's variance.
- $0.5 \leq R^2 < 0.7$: Strong correlation. The model explains a substantial portion of the variance in the data.
- $R^2 \geq 0.7$: Very strong correlation. The model offers a high degree of explanation for the variance in the data.

While a higher $R^2$ value indicates a model that can better explain the variance observed in the dependent variable, it does not necessarily imply a causal relationship between the dependent and independent variables. It is also critical to consider other statistical metrics and tests when assessing the performance of a regression model, as $R^2$ alone may not provide a complete picture of the model's effectiveness.

References that delve into the concept and applications of $R^2$ in regression analysis, emphasizing its significance in evaluating model fit and understanding the variability in data, including foundational texts and studies in the field of statistics and econometrics.

**Limitations.** While our NeuroMFA framework demonstrates correlations between the emergence metric and downstream performance by studying the self-organization of LLMs, it has not yet established a clear causal relationship. Future work should focus on identifying specific linguistic

phenomena captured by LLMs and demonstrating how our analysis methods can reveal the learning process of these phenomena during training, thereby establishing a stronger causal connection.

