# OpenReview forum: "Neuron-based Multifractal Analysis of Neuron Interaction Dynamics in Large Models"
_ICLR.cc/2025/Conference — ICLR 2025 Poster_

### Official Review · Reviewer_VqLs · 2024-10-29

**Soundness:** 2
**Presentation:** 3
**Contribution:** 3
**Rating:** 5
**Confidence:** 3

**Summary:**

This paper presents a new approach to studying the internal property of large language models. I got really excited by the discussion but a bit disappointed by the results. I believe the efforts in this paper should be generalized across the field.

The parallel between the brain and LLMs in terms of their complexity and emergent characteristics is fascinating. This paper aims to establish a connection with network analysis in neuroscience by characterizing the evolution of organizational patterns within the network during training and linking these to different (emerging) performances.

I disagree with the article’s use of “emergence,” although I understand the rationale. You state that your measure is intended to quantify network regularity. The main conclusion of the article should be that this measure appears sufficient as a proxy for the emergent properties of LLMs.

The results are interesting but quite weak and merit further development. In particular I felt like many network properties could correlate (positively or negatively) with the training stages of LLMs. Nonetheless, this is an intriguing direction towards understanding network local properties and their link to emergent properties.

**Strengths:**

The paper is extremely well contextualized and the presentation of the ideas and results are solids. The quality of the writing makes it really pleasant to read.

It gives a really interesting direction of research by trying to bridge a gap with neuroscience and trying to create a new framework to link emerging properties and network topologies.

**Weaknesses:**

Some mathematics would have deserved better explanation in the article itself and not the appendix.

 You are measuring the degree of organisation in the networks and show that :
1. it increases during training
2. it correlates with previous definitions of emergence

Your are yourself comparing it with the averaged weighted degree distribution of the network in the appendix which seem to correlates as well with emergence.

I feel like your "degree of emergence" is itself one dimensional as well, which makes it way less interesting as if it was not. It would be a bit outrageous if we were able such a complex reorganisation with only one global metric. It would much more interesting if you were able to link certain network properties with specific emerging properties.

**Questions:**

* Your description of the neural network as blocks and layers seems unnecessary. Simply stating that the neural network can be described as a graph with  w_{ij}  values between neurons is sufficient for your further definitions. The layer and block definitions are not used later, except to describe weights that ultimately serve as connections between two nodes.

* A reference is missing in the appendix on page 39.

* Emergence is, by definition, a property observable at a higher scale that is not explainable by its components. You are measuring network properties, which is interesting, but is it truly the appropriate scale to discuss emergence? I am not convinced the term “emergence” is wisely used in this article. Instead, the focus should be on providing a quantification of network structure that enables or correlates with emergent properties.

On the more serious problems :

* You don’t explain how  |P(i,j)|  is calculated. Do you add +1 for each edge along the shortest path? This leaves it unclear whether your distance is calculated exactly.

* I feel that many regularity properties would undergo the same type of increase as the one you noted, such as an irregularity, heterogeneity metric or average weighted degree distribution . What were the results (Figure 5 and 24) ? If simpler metrics correlate with emergence similarly, what additional value does your metric provide?

---

> ### Author Response · Authors · 2024-11-26
> **Thank you for your thoughtful and constructive feedback.**
>
> We would like to thank you for recognizing the novelty and merit of our method. We provide details answers to your questions and concerns and look forward to a fruitful discussion. We hope that our detailed explanation clarifies all your questions and resolves your concerns.
>
> >**W 1:** Some mathematics would have deserved better explanation in the article itself and not the appendix.
>
> **A:** Thank you for your suggestions. We understand the importance of presenting key mathematical explanations clearly within the main body of the paper. In light of the reviewer’s suggestion, we have also recognized the critical need to elucidate the mathematical rationale behind the metrics. To address this, we have included a dedicated "Motivation" section in the main text to provide further clarity. For additional details, please refer to **General Response Answer A.2**.
>
> ---
>
> >**W2:** You are measuring the degree of organisation in the networks and show that :
> >1. it increases during training
> >2. it correlates with previous definitions of emergence
> >Your are yourself comparing it with the averaged weighted degree distribution of the network in the appendix which seem to correlates as well with emergence.
>
>
> > **Q5:** I feel that many regularity properties would undergo the same type of increase as the one you noted, such as an irregularity, heterogeneity metric or average weighted degree distribution . What were the results (Figure 5 and 24) ? If simpler metrics correlate with emergence similarly, what additional value does your metric provide?
>
> **A:** Thank you for your specific question about the importance of the proposed metrics. To clarify, individual metrics such as irregularity, heterogeneity, or average weighted degree distribution **cannot** reliably align with model performance. As illustrated in Figure 19 (revised version, page 39), irregularity and heterogeneity, when used as standalone metrics, can only reveal trends in self-organization but fail to effectively measure emergence. Furthermore, the average weighted degree distribution also reflects some self-organization trends but is insufficient for accurately capturing emergence. As shown in Figure 28 (revised version, page 44), the metric demonstrates significant numerical variability. For instance, from epoch 10,000 to epoch 143,000, the values decline, contradicting actual observations. This discrepancy underscores the necessity of integrating both **heterogeneity** and **irregularity** dynamics grounded in the principles of self-organization and emergence in natural systems to develop a robust metric capable of measuring the emergence of large models effectively. For a detailed explanation, please refer to **General Response A.2**.

---

> ### Author Response · Authors · 2024-11-26
> **Cont.**
>
> >**W3:** I feel like your "degree of emergence" is itself one dimensional as well, which makes it way less interesting as if it was not. It would be a bit outrageous if we were able such a complex reorganisation with only one global metric. It would much more interesting if you were able to link certain network properties with specific emerging properties.
>
> **A:** Thank you for raising this insightful question. The network properties we utilize—**heterogeneity** and **regularity**—are closely tied to emergent properties, as they directly correspond to the key characteristics of self-organization and emergence commonly observed in natural systems. Specifically, (i) heterogeneity reflects the **diversity of interaction patterns** within the system, while (ii) regularity captures the degree of global **structural order**. These properties serve as the foundation and motivation for our metric design, directly inspiring our approach. For a more detailed response, please refer to **General Response Answer A.2**.
>
> ---
>
> > **Q1:** Your description of the neural network as blocks and layers seems unnecessary. Simply stating that the neural network can be described as a graph with w_{ij} values between neurons is sufficient for your further definitions. The layer and block definitions are not used later, except to describe weights that ultimately serve as connections between two nodes.
>
> **A:** Thank you for your suggestion! We completely agree with your point. While our initial description was inspired by the structure of LLMs, the use of "blocks" indeed appears unnecessary for the introduction of our methods. Following your suggestion, we have revised the paper to simplify the notation, removing the concept of "blocks" and referring only to "**layers**" in the updated version (revised version, page 4).
>
> ---
>
> > **Q2:** A reference is missing in the appendix on page 39.
>
> **A:** We appreciate your careful reading and for pointing this out. This issue has been fixed in the revised version (revised version, page 40).
>
> ---
>
> > **Q3:** Emergence is, by definition, a property observable at a higher scale that is not explainable by its components. You are measuring network properties, which is interesting, but is it truly the appropriate scale to discuss emergence? I am not convinced the term “emergence” is wisely used in this article. Instead, the focus should be on providing a quantification of network structure that enables or correlates with emergent properties.
>
> **A:** Thank you for the reviewer's insightful comments. As you noted, "Emergence is, by definition, a property observable at a higher scale that is not explainable by its components." Similarly, our work aims to study the emergence phenomenon in large models not through components but through their complex interaction structures and dynamics, much like how emergence in natural systems is examined via **self-organization**. By investigating the relationship between *network structure* and *self-organization*, we relate this metric to observable *emergent capabilities*. For further details on the connection, please refer to **General Response Answer A.2**.
>
> ---
>
> > **Q4:** You don’t explain how |P(i,j)| is calculated. Do you add +1 for each edge along the shortest path? This leaves it unclear whether your distance is calculated exactly.
>
> **A:**: Thank you for your insightful question regarding the calculation of $|P(i,j)|$. To clarify, $|P(i,j)|$ is defined as the number of edges in the path $P(i,j)$ connecting nodes $i$ and $j$, which essentially represents the length of their shortest path. We have incorporated this definition into the revised version of the paper (see page 5).
>
> **Why we need the penalty term:** Given the hierarchical structure of neural networks, where single layers have fully connected patterns, calculating all possible shortest distances in the network can be **extremely time-consuming**. Additionally, this process often results in calculating **redundant interaction patterns across layers**. For example, a three-layer connection following the first layer and a two-layer connection following the second layer are structurally identical. To address these issues, we introduce the penalty term $|P(i,j)|$ combined with a threshold (see Equation 2) to filter out deep multi-layer connections. This approach allows us to prioritize local interactions while effectively distinguishing overlapping patterns in multilayer connections, avoiding unnecessary repetition.
>
> ---
>
> **Update:** During further discussions with the other reviewers regarding **the use of “emergence,”** we carefully considered your comments alongside their suggestions and have made adjustments and clarifications to address this issue. Please refer to **General Response Answer A.3** for a detailed explanation. We greatly appreciate your time and effort, which have significantly contributed to improving our manuscript, and we welcome any further discussion.

---

> ### Comment · Reviewer_VqLs · 2024-12-01
> **Feedback on Revised Manuscript**
>
> Thank you for the extra work you put in that paper. You improved the motivation of the work and provided the necessary details to understand your work, I increase the "Contribution score" from 2 to 3. I still find the topic and direction of your work highly interesting and potentially impactful.
>
> However, after further consideration, I believe the manuscript requires additional iterations to reach the level of clarity and rigor necessary for acceptance. Here are my main concerns:
> *  As noted in my initial review and echoed by other reviewers, the use of the term “emergence” in the paper is inconsistent with its accepted definition. Your metric, as described, characterizes the degree of self-organization rather than directly quantifying emergence. For example, your conclusion correctly notes that self-organization properties can lead to emergence, but the title and much of the text do not make this distinction clear. For instance, the title (“Assessment of the Degree of Emergence Across Various Datasets”) is misleading. What does it even mean since your "degree of emergence" is the **same for every datasets**  (since you are showing the exact same curve of degree of emergence here) . I realize that this misuse of the term make the whole paper rather confusing.
> II also notice that you have replaced some mentions of accuracy with “emergent abilities,” but then the two terms are the same for really different aspects. This is detrimental to the clarity of your paper. and this inconsistency makes the manuscript confusing.  For all those reasons, I am downgrading the Presentation score from 4 to 3.
>
> * While the proposed structural metric is interesting, its practical utility as a proxy for model performance requires stronger evidence. For instance:
>    * Figure 16 for instance. After epoch 6000 or 8000, your metric decreases. Does this imply a decline in model performance ?
>    * Same Figure 16, the metric values differ between two models. Does this mean one model outperforms the other, based solely on this metric and their parameter counts?.
> If the goal is to propose your metric as a proxy for accuracy (what other people would call emergence), it is essential to provide a clear comparative analysis and explicitly connect metric trends with performance outcomes.
>
> * A consolidated graph that overlays performance across all datasets would also help make this connection clearer, rather than presenting results in five separate figures. (For presentation of the results, should do a single graph that shows all those datasets rather 5 different ones. ).
>
> In summary, while the paper has many strengths and is headed in an exciting direction, it is not yet aligned with its stated goals. Focusing on the metric’s strengths without overextending its implications would make the paper more credible. For all these reasons, I cannot recommend this work for acceptance at this time, and I will not change my original grading.
>
> That said, I genuinely believe that with additional work, this could become an excellent paper. Thank you again for your hard work, and I look forward to seeing how your research progresses.

---

> ### Author Response · Authors · 2024-12-02
> **Further Discussion**
>
> Thank you for recognizing the contributions of our work and for further highlighting areas where clarity could be improved. We sincerely appreciate the time and effort you dedicated to improving our manuscript. Based on our understanding of your comments, we believe that your concerns primarily stem from certain **terminology** and phrasing within the manuscript. These are areas that we have worked on and continue to improve, and we believe they can be addressed with relatively small adjustments and refinements in the final version.
>
> As highlighted in **General Response Answer A.3**, we noticed that the core concerns regarding the term "emergence" stem from the nuanced differences in its definition between natural complex systems and the LLM domain. Recognizing this distinction, we acknowledge that fully reconciling these differing definitions across the two domains is challenging and beyond the immediate scope of this work. Our goal is to delineate the two usages within the manuscript to better explain our methodology while avoiding confusion, as outlined in General Response Answer A.3, and we also provide detailed explanations in the following one-to-one responses.
>
> Moreover, we are grateful for the opportunity to clarify the content of the manuscript and to address your remaining concerns. We are confident that, through minor revisions, we can address these issues while preserving the core contributions of our work. Below we provide the **one-to-one response** regarding your questions:

---

> ### Author Response · Authors · 2024-12-02
> **Cont.**
>
> ### Q1.1 The use of “Emergence”
>
> > **Q1.1:** As noted in my initial review and echoed by other reviewers, the use of the term “emergence” in the paper is inconsistent with its accepted definition.
>
> **A:**  Thank you for your further questions regarding the use of the term "emergence." First, we would like to point out that, in our further discussion with Reviewer 2tKM, we realized and addressed the potential misunderstandings that may arise from the **cross-disciplinary use** of the term "emergence" in our manuscript, particularly across the fields of complex systems and LLMs. The definition of "emergence" in complex systems differs subtly from its usage in the LLM domain. In our work, the metrics are inspired by the definitions and principles of self-organization and emergence in natural complex systems, where "emergence" refers to the **appearance of novel, system-level properties** that arise from the **interactions and self-organization** of components. However, in the context of LLMs, "emergence" is often used to mean simply the "appearance of new capabilities" that are absent in smaller models. This dual usage could potentially lead to misunderstandings. Therefore, as clarified in **General Response Answer 3**, we have made several key adjustments to address these concerns in our revised manuscript:
> 1. **Abstract adjustment:** We have revised the abstract from “a novel structure-based metric for evaluating the degree of emergence in large models” to “a novel structure-based metric as a proxy for emergent abilities of large models.”
>
> 2. **Clarification of the "degree of emergence" definition:** We have clarified that the "degree of emergence" in our method refers specifically to emergence following structural self-organization. This clarification has been added where the term  "degree of emergence" first appears (see revised version, page 3), explicitly stating that:“ degree of emergence (this is a structure-based metric that can be viewed as a proxy for the emergent abilities of large models)”.
>
> 3. **Terminology modification:** To further reduce ambiguity, we have unified the references to “emergence” associated with large model capabilities, replacing them with “**emergent abilities**” or “**development of skills**” as the reviewer oHwU suggested. For structural aspects, we continue to use "emergence" (e.g. “degree of emergence”) to align with the definitions in complex systems, where the term specifically describes the arising of novel, coherent properties or patterns resulting from structural self-organization.
>
> We are glad that these clarifications and adjustments have addressed the concerns about the use of "emergence" raised by Reviewer 2tKM and Reviewer oHwU. Based on your feedback, we will explicitly incorporate this discussion and briefly explain the use of "emergence" at the **very beginning of the manuscript** in the final version to better guide readers. We sincerely appreciate the efforts and insightful suggestions of you and other reviewers, please let us know if you have any further specific concerns or suggestions regarding the term usage.

---

> ### Author Response · Authors · 2024-12-02
> **Cont.**
>
> ### Q 1.2: The fundamental representation of our metrics
>
> > **Q 1.2:** Your metric, as described, characterizes the degree of self-organization rather than directly quantifying emergence. For example, your conclusion correctly notes that self-organization properties can lead to emergence, but the title and much of the text do not make this distinction clear. For instance, the title (“Assessment of the Degree of Emergence Across Various Datasets”) is misleading. What does it even mean since your "degree of emergence" is the same for every datasets (since you are showing the exact same curve of degree of emergence here) .
>
> **A:** Thank you for raising the question about the fundamental representation of our metric and the relationship between self-organization and emergence. First, we would like to point out that we have emphasized this relationship multiple times in the main text of the manuscript, for example:
>
> “Self-organization phenomena, widely observed in natural systems, involve numerous micro-level interactions leading to complex macro-level behaviors, and the emergence can be understood by modeling and analyzing the self-organizing micro-structure (Turing, 1990) (see Appendix A.3).” (Line 101-104, page2)
>
> “Self-organization is a foundational concept for understanding emergent phenomena in natural systems (Goldstein, 1999; Crutchfield, 1994). It describes how systems gradually evolve through local interactions to develop more diverse interaction patterns and achieve overall order, ultimately leading to emergence (Correia, 2006; Ay et al., 2007).” (Line 178-181, page 4)
>
> In simple terms, self-organization refers to the process of interaction and structural changes at the **micro level**, while emergence highlights the resulting novel properties and behaviors observed at the **macro level**. The two metrics **regularity** and **heterogeneity**, indeed characterize the self-organization process to measure **micro-level interactions**. The multifractal spectrum, as structural features that capture interaction patterns at the micro level, also serve as indicators of self-organization. As we discuss in Section 4.1 (“Self-Organization Process of the NIN in LLMs”), these metrics observe the evolution of interaction patterns at the neuron level (micro-level), regarded as metrics that characterize the self-organization process.
>
> However, as explained in General Response Answer A.2, we bridge these structural features of self-organization, measured by **regularity** and **heterogeneity**, to **macro-level** properties (abilities) by proposing the metric degree of emergence.
> Because this metric serves as a proxy for **macro-level** properties (abilities), we consider it relevant to emergence. Thus, while regularity and heterogeneity measure self-organization, the proposed metric serves as a proxy for evaluating macro-level abilities, which we frame in terms of the degree of emergence.
>
> **“Assessment of the Degree of Emergence Across Various Datasets”** Thank you for pointing this out. The term "datasets" mentioned here does not refer to the training datasets but rather to the datasets used for evaluating the performance of LLMs, as detailed in Appendix B.6. In this section, we compare the structural metric, degree of emergence, with other metrics designed to assess various aspects of model performance (tested using different datasets).  The model itself remains fixed, while only the test datasets vary; therefore, the structural metric, degree of emergence, remains constant. We acknowledge that this phrasing is not precise, and we sincerely appreciate you bringing this issue to our attention. To address this, we will revise the term "datasets" to "benchmarks" and ensure that this change is accurately reflected in the final version.
>
> We appreciate your insightful question and hope this clarification addresses your concern. If you have further questions or suggestions, we would be happy to discuss them.
>
> ---
>
> ### Q 1.3: The use of “Emergent ability”
>
> > **Q 1.3:** I also notice that you have replaced some mentions of accuracy with “emergent abilities,” but then the two terms are the same for really different aspects.
>
> **A:** Thank you for raising this question. As mentioned in General Response Answer 3, we have replaced "emergence" with "development of skills" or "emergent abilities" when referring to the capabilities of large models, to further eliminate potential confusion arising from the cross-disciplinary usage of the term "emergence." If you identify any instances of imprecise wording, we would appreciate your feedback and will be more than happy to further enhance the clarity of our manuscript. We will also continue to conduct thorough reviews before the final version to address any potential issues with term usage or wording.

---

> ### Author Response · Authors · 2024-12-02
> **Cont.**
>
> > **Q 2**:While the proposed structural metric is interesting, its practical utility as a proxy for model performance requires stronger evidence. For instance:
> Figure 16 for instance. After epoch 6000 or 8000, your metric decreases. Does this imply a decline in model performance ?
> Same Figure 16, the metric values differ between two models. Does this mean one model outperforms the other, based solely on this metric and their parameter counts?. If the goal is to propose your metric as a proxy for accuracy (what other people would call emergence), it is essential to provide a clear comparative analysis and explicitly connect metric trends with performance outcomes.
>
> **A:** Thank you for raising this insightful question. We noticed that there are still some misunderstandings regarding the **goals and contributions** of our work. Our goal is not solely to “propose the metric as a proxy for accuracy.” Our primary objective is to develop a comprehensive **analytical framework for large-scale AI models**, with a particular focus on establishing a **theoretical foundation** for exploring the multifractal structure of neural networks. This framework holds significant potential for broad applications.
> Thus, we recognize the need to further validate our approach and clarify our contributions from three key aspects:
> 1. The relationship between the metric and model performance, including its behavior during performance degradation.
> 2. Our goals and the potential applications of the method.
> 3. Why existing network-based analysis methods are almost infeasible for large models and why NeuroMFA is particularly suited for this context.
>
>
> # 1. The relationship between the metric and model performance, including its behavior during performance degradation.
>
>
> To answer your questions about (1) After epoch 6000 or 8000, your metric decreases. Does this imply a decline in model performance? (2) Does this mean one model outperforms the other, based solely on this metric and their parameter counts? We further elaborate on the relationship and correlation between structure and performance from both **theoretical and experimental** perspectives.
>
>
> ## Theoretical Perspective
>
> From a theoretical standpoint, a higher degree of emergence typically corresponds to greater structural heterogeneity and regularity, while a decrease in the degree of emergence indicates the opposite. Better structural organization suggests greater potential for better performance, as performance can be viewed as a macro-level manifestation of underlying structural features. In principle, **micro-level structural support** often correlates with **macro-level performance** in natural systems to some extent. However, performance is a broad and multi-faceted concept, and test results can exhibit fluctuations. While performance and structure are **generally consistent** in overall trends, specific performance metrics and specific structural features are not necessarily directly correlated. This aligns with our discussion in the Limitations section, where we outline future work to explore the causal relationship between structure and performance. This will involve extensive experiments to observe how specific structural characteristics at different scales correspond to specific performance metrics.
>
> ## Experimental Analysis
>
> Experimentally, we observed that the Pythia-standard model shows rapid growth in the metric at the beginning of training. After epoch 6000 or 8000, the metric decreases. To investigate this, we tracked the Lambada OpenAI metric for the Pythia-160M-standard model. Interestingly, as shown in [Performance](https://anonymous.4open.science/r/Neuron_LLM-4DC8/Graphs/standard.png), Pythia-160M-standard indeed reaches its peak performance at around epoch 6000, followed by a gradual decline. Due to the sparse sampling of epoch points, it is possible that fluctuations between consecutive epochs were not fully captured, and what appears to be a slight decline may instead reflect small-scale oscillations in performance. Given the constraints on time, we are unable to conduct additional experiments to further substantiate this connection at present. However, the experiments previously completed and shared with reviewers **2tKM** and **oHwU** provide evidence that our metric can capture trends associated with performance degradation, as described below.
>
>
>
>
> ### **Experiment: Performance Degradation with Contaminated Data**
>
>
> We utilized the Hugging Face pre-trained BERT-large-uncased model (24 layers, 336M parameters, 1024 hidden dimensions, 16 attention heads) [Source] (https://huggingface.co/google-bert/bert-large-uncased) [1].
>
> [1] Kenton JD, Toutanova LK. Bert: Pre-training of deep bidirectional transformers for language understanding. InProceedings of naacL-HLT 2019 Jun 2 (Vol. 1, p. 2).

---

> > ### Author Response · Authors · 2024-12-02
> > **Cont.**
> >
> > **Dataset Construction:**
> >
> >
> > To simulate performance degradation, we created a deliberately contaminated dataset:
> >
> >
> >   * Retaining 20% of the original Wikipedia dumps dataset.
> >
> >
> >    * Replacing the remaining 80% with random data generated from a normal distribution.
> >
> >
> > **Evaluation Framework:**
> >
> >
> >  * Structural Metric: we assessed the model's degree of emergence throughout training using **NeuroMFA**.
> >
> >
> >  * Performance Metrics: model performance was evaluated using two standard metrics: (i) SQUAD 1.1 F1 score for reading comprehension; (ii) Multi NLI Accuracy for natural language inference.
> >
> >
> > **Results:**
> >
> >
> > As shown in Figure [Decreasing Performance](https://anonymous.4open.science/r/Neuron_LLM-4DC8/Graphs/Decreasing_Performance.png), the structure-based metric captured this pattern of performance degradation, decresing from 0.6 to less than 0.1. This experiment demonstrates that when a large model experiences performance degradation due to severely contaminated datasets, its behavior becomes increasingly random, and the ordered structure of its network is also disrupted. This finding emphasizes the intrinsic connection between a model's performance and its **network structure**.
> >
> >
> > # 2. Our goals and the potential applications of the method.
> >
> > To clarify, our goal is not solely to “propose the metric as a proxy for accuracy.” Our primary objective is to develop a **comprehensive analytical framework** for large-scale AI models, focusing on establishing a **theoretical foundation** for exploring the multifractal structure of neural networks.
> >
> > We are **the first** to propose a systematic investigation into the dynamic behavior of neuron networks within large models, bridging this micro-level structural analysis to macro-level model capabilities. The challenges of applying traditional network analysis to large models will be discussed in detail in the third point below. Importantly, this framework is not merely intended to serve as a proxy for accuracy but rather as a tool to provide an entirely new perspective on large artificial model analysis.
> >
> > As we mentioned in further discussions with Reviewer **2tKM**, the concept can be explained through an **Intuitive Analogy**, designed to be more accessible and engaging, albeit not strictly adhering to scientific rigor:
> >
> > **Intuitive Analogy:** To provide a more straightforward and intuitive analogy, purely for the sake of an engaging discussion beyond the scope of strict scientific rigor, consider how we evaluate human intelligence (IQ) or brain development. Traditionally, intelligence might be assessed through various tests targeting different dimensions, such as logical reasoning or language skills. These tests, akin to behavioral metrics in LLMs, are task-specific and offer valuable insights into particular intelligent abilities. Now, imagine if we could also obtain precise data about the brain's structure and neural signal transmission. In such a scenario, researchers might aim to develop novel analytical methods that measure the organization, connectivity patterns, and dynamics of neuronal activity in real-time, thereby providing a completely different perspective on intelligence and brain functionality.
> >
> > Similarly, our work aims to take **a first step** toward studying the structure and internal dynamics of large models, providing a theoretical framework supplemented by initial empirical evidence that suggests practical implications. Moreover, our work has various potential application scenarios, including but not limited to:
> >
> >
> > “Regarding the **guidance of model development**, we do not view NeuroMFA merely as a preliminary diagnostic tool but as a theoretical foundation for incorporating higher structural heterogeneity and regularity into the design and training of large models. For example, we could **construct loss functions** that explicitly encourage the increase of structural heterogeneity and regularity during training. Similarly, in **pruning operations**, nodes with homogeneous fractal properties (identified by the identical or similar fractal dimension) could be prioritized for removal to enhance the overall heterogeneity of interactions. These are just two examples, and we believe NeuroMFA can inspire a broader range of strategies and applications, making it a promising foundation for further exploration and innovation in future research.”
> >
> >
> > Finally, we will further discuss why traditional network analysis methods **fail to** effectively meet the requirements of an analytical framework for large models, while also highlighting how our proposed NeuroMFA method enables efficient analysis of neural networks in large models, representing a key contribution of our work.

---

> ### Author Response · Authors · 2024-12-02
> **Cont.**
>
> > **Reviewer's suggestion:** Focusing on the metric’s strengths without overextending its implications would make the paper more credible.
>
> # 3. Why existing network-based analysis methods are almost infeasible for large models and why NeuroMFA is particularly suited for this context.
>
> ### **Key Challenges in Applying Network Analysis to Large Neural Models:**
>
>
>
>
> 1. **Unique Structural Characteristics of Neural Networks:** Neural networks exhibit **directed, weighted, and hierarchical** structures, which make traditional metrics designed for unweighted or undirected networks less applicable.
>
>
>
>
> 2. **Scalability to Large Models:** Large models have billions of parameters, creating networks of enormous size. Many traditional metrics become computationally infeasible for networks exceeding 10 million nodes or edges.
>
>
>
>
> 3. **Need for Multiscale Analysis:** Our objective is to bridge local (microscopic) structural changes in neural networks with their emergent macroscopic capabilities. This requires metrics capable of capturing both local interactions and global multiscale properties.
>
>
>
>
> Based on these points, we analyzed the potential applications of traditional network metrics in neuronal networks and neural networks, as well as their limitations when applied to **large-scale models**.
>
>
> ### **Comparison with Traditional Metrics:**
>
>
> 1. **Centrality Metrics：** Centrality measures such as degree, closeness, and betweenness centrality quantify the importance of nodes within a network.
>     *  **Potential Applications:** Useful for identifying key neurons in neuronal networks that are critical for information propagation or specific task processing.
>     * **Limitations:** Centrality metrics focus on **individual node-level properties**, making them less suitable for capturing global network dynamics or multiscale interactions.
>
>
> 2. **Small Worldness:** Small worldness quantifies the balance between local clustering and global efficiency using metrics such as local clustering coefficients and average shortest path lengths.
>     *  **Potential Applications:**  In neuronal networks, it helps understand the trade-off between local processing and global information integration, as well as network efficiency.
>     * **Limitations:** Primarily designed for undirected and unweighted networks, small worldness (especially for **the calculation of average shortest path**) is computationally expensive for large, weighted networks like those in LLMs.
>
>
> 3. **Modularity:** Modularity measures the degree to which a network can be divided into distinct modules or communities.
>     *  **Potential Applications:**   In neuronal networks, modularity can be used to identify functionally distinct sub-networks or quantify modular brain region connections.
>     * **Limitations:** Modularity identifies large-scale structures but fails to capture **node-level interactions** or multiscale properties. Additionally, it was initially designed for undirected and unweighted networks. Although adaptations exist, they are often computationally expensive to analyze the directed and weighted structures in neural networks.
>
>
>
>
> 4. **Entropy-Based Metrics:** Entropy metrics quantify the diversity or randomness of network structures, often focusing on degree distributions or edge weights.
>     *  **Potential Applications:** Useful for measuring the diversity of synaptic connections or activity patterns in neuronal networks.
>     * **Limitations:**  Entropy metrics analyze **distributions** rather than interactions, making them unsuitable for multiscale analysis. Additionally, entropy calculations for large, directed, and weighted networks are resource-intensive.
>
>
>
>
> 5. **Ricci Curvature:** Ricci curvature provides geometric insights into connectivity and robustness by measuring edge-based properties.
>     *  **Potential Applications:**  Applied in neural networks to identify critical pathways for signal transmission or bottlenecks in artificial neural networks.
>     *  **Limitations:** Ricci curvature is computationally prohibitive for large-scale, weighted networks and focuses on **edge properties**, offering limited insights into global network structure.
>
>
> ### **Why NeuroMFA? (The strength of NeuroMFA)**
>
>
> Compared to the metrics above, NeuroMFA offers several distinct advantages:

---

> ### Author Response · Authors · 2024-12-02
> **Cont.**
>
> Compared to the metrics above, NeuroMFA offers several distinct advantages:
>
>
> 1. **Node-Based Design:** NeuroMFA analyzes nodes directly, making it well-suited for studying local interactions and scaling up to **multiscale analyses**.
>
>
> 2. **Efficiency:** In NeuroMFA, the analysis starts from **node-based local properties** and leverages multiscale characteristics to derive the overall network's multifractal features. This method, which relies on local properties rather than direct computation over the entire network, significantly improves efficiency. Additionally, by leveraging scaling laws, NeuroMFA allows for network sampling without compromising the accuracy of fractal dimension estimation (see Appendix A.1.3, revised version, page 19), making NeuroMFA **computationally efficient**.
>
>
> 3. **Multiscale Analysis:** NeuroMFA captures both local and global properties, enabling us to capture the **heterogeneity** (diversity of local interactions) and **irregularity** (global structure) of the network structure across multiple scales, aligning with natural systems' self-organization and emergence rationale.
>
>
> In summary, although traditional metrics can be useful for analyzing smaller neural networks, they are often limited by computational inefficiency or lack of the ability of multiscale analysis when applied to large-scale neural networks. NeuroMFA overcomes these challenges by providing an efficient, scalable, and multiscale framework tailored to the unique characteristics of large neural models. As a practice example, we attempted to apply **Ollivier Ricci Curvature** to a **1.4B LLM**, but the process took over a week before the server crashed. In contrast, NeuroMFA completes the analysis of the same 1.4B model in approximately **45 minutes**.
>
> ---
>
> > **Q3 :** A consolidated graph that overlays performance across all datasets would also help make this connection clearer, rather than presenting results in five separate figures. (For presentation of the results, should do a single graph that shows all those datasets rather 5 different ones. ).
>
> **A:** Thank you for your suggestion regarding consolidating the performance metrics. In our initial analysis, we attempted to plot all five performance metrics in a single graph using the Pythia-1.4B model as an example, as shown in Figure [metrics_all](https://anonymous.4open.science/r/Neuron_LLM-4DC8/Graphs/metrics_all.png). However, due to the differing ranges of accuracy values across metrics, some detailed variations became less apparent—particularly for metrics with smaller ranges, such as ARC_Challenge and PIQA.
>
> Regarding your valuable suggestion, we normalized all metrics and created a new consolidated visualization, as shown in Figure [metrics_all_normalized](https://anonymous.4open.science/r/Neuron_LLM-4DC8/Graphs/metrics_all_normalized.png). While this normalization results in the loss of some information about the absolute accuracy values, it effectively addresses our previous concerns about scale differences, allowing us to clearly display the detailed variations of all metrics in a single figure. We sincerely appreciate your constructive feedback and will include this new visualization in the appendix of the final version.
>
> ---
>
> We sincerely appreciate the time and effort you have dedicated to reviewing our work and rebuttal. Your feedback is highly valued. We will incorporate the insights (such as the discussion of **the strengths of our NeuroMFA method**) from our discussions into the final version. Our aim is not merely to address your concerns and suggestions but to refine our work to its fullest potential, crafting a submission that resonates with a wider audience. We remain open to any further discussion.

---

> > ### Author Response · Authors · 2024-12-03
> > **Kindly Follow Up**
> >
> > We truly appreciate your thoughtful feedback and the time you’ve dedicated to reviewing our work and rebuttal. As the discussion period is coming to a close, we wanted to kindly follow up to ask if there are any remaining questions or concerns we can address. We remain available to discuss any points in greater detail if needed.
> >
> > Thank you again for your invaluable input, and we look forward to hearing any additional thoughts you may have.

---

### Official Review · Reviewer_oHwU · 2024-11-01

**Soundness:** 4
**Presentation:** 3
**Contribution:** 4
**Rating:** 8
**Confidence:** 5

**Summary:**

In this paper the authors ask the question, is there a better metric than raw parameter number to capture why qualitatively new skills develop when scaling up large transformer models. They introduce the network structural / topological measure of ‘NeuroMFA’ and show that this measure can capture how skills develop throughout training, something which raw parameter numbers necessarily cannot capture. They propose that their measure introduces a new way of studying learning of skills across time in large scale neural network architectures.

**Strengths:**

I genuinely enjoyed reading this paper. It introduces a new measure to study the training of LLMs that I have not seen in this way before. The paper generally is very thorough in its analyses / experimentation. I am sure this will be work that is of interest to the ICLR community.

**Weaknesses:**

I think this paper should definitely appear in the proceedings. I see some weaknesses in the way that methods are introduced, and results are described, which I list in the questions below. I hope my suggestions will be helpful to create more impactful paper.

**Questions:**

*Baseline calculation*

This is my only comment about a possible analysis to add. As you are proposing a new measurement, I think it would be helpful to provide measurements in untrained networks. Of course, Figure 8 shows the effect of training on the metric, but I wonder whether it actually would be zero across all models at the start of training? If it isn’t, should be somehow standardise the metric so that it is zero before training? If this was explore in more detail I would be happy to raise my ‘soundness’ rating to ‘excellent’.

*Communication of methods*

Given the number of results in this paper and the resulting length of the appendix, I think the authors should really aim to communicate the core information in the main text. I think the following would help:

- Given that you use transformers with attention, I imagine the author analyse the effective interaction of model parts conditioned on specific dataset. Naturally the attention weights are going to change how different parts of the models interact and so it seems one could not analyse the structure of the attention weights in general but instead would look at the effective communication when processing specific samples? Or do authors actually just use the weights? It would be great if this could be clarified, ideally at the start of the methods.
- I think I generally got the metric development, but I did find it somewhat hard to follow. I wonder whether this could be improved if every new part of the method introduced would start with a sentence or two on the high-level intuition behind the measure / what the current analyses step is for – take section 3.2 for example. It starts with “To perform multifractal analysis of LLMs, we extend the box-covering and box-growing methods (Song et al., 2005; Evertsz & Mandelbrot, 1992; Salat et al., 2017; Xiao et al., 2021) to capture the local neuron-based fractal properties of NINs.”, followed by plenty of details on methods. If you lead into this section by first giving the reader an idea why it is interesting to decompose the graph into fractals in the first place, then maybe this would be easier to follow. Generally speaking, motivate the method before jumping into details. I know that the paper is already on the page limit but frankly the appendix is so long, moving a bit more of the methods to appendix in lieu of giving readers a better intuition in the main text is probably beneficial.
- Given there already is a bunch of structural measures which have successfully been used for structural analyses in neuroscience and neural network models (e.g. standard ones like modularity / small worldness, or entropy-based ones or Ricci curvature) I wonder whether authors could somehow motivate why their new measure was needed? Given it works, I do not want to say that the new measure was not needed but I would be curious to know why they specifically took the route they took with the development of the metric.

*Stylistic choices*

- I think the font on Figure 6 is very close to unreadable small.
- Across figures, could you either use a serif or sans serif font but not mix that up?
- I wonder whether your strong emphasize of the word ‘emergence’ is actually needed and helpful in communicating your results? In my experience it does put off some researchers and, as you make a link to neuroscience, the development of language and reasoning skills from a newborn to a teenager would simply be considered as part of ‘development’ and ‘learning’. So, I wonder whether you want to deemphasize the concept of emergence a bit here and just call it ‘the development of skills’?

---

> ### Author Response · Authors · 2024-11-26
> **Thank you for your through review and encouraging feedback.**
>
> We would like to thank you for your careful reading and kind words. We are encouraged that you liked our paper. Your suggestions indeed created a more impactful paper and helped improve its narrative and the way our method is described. Please find point-to-point answers to your questions below.
>
>
>
>
> > **Baseline calculation:** This is my only comment about a possible analysis to add. As you are proposing a new measurement, I think it would be helpful to provide measurements in untrained networks. Of course, Figure 8 shows the effect of training on the metric, but I wonder whether it actually would be zero across all models at the start of training? If it isn’t, should be somehow standardise the metric so that it is zero before training? If this was explore in more detail I would be happy to raise my ‘soundness’ rating to ‘excellent’.
>
> **A:** We appreciate your question about the measurements in untrained networks. Regarding the **initial training phase**, as shown in the spectrum plot in Figure 5 (revised version, page 8), the spectra marked as **Epoch 1** represent the initial random network. It is important to note that our method focuses on studying **structural changes**, specifically the transition from randomness to order in the self-organization process. Therefore, our emergence metric is inherently based on the initial state (see Equation (12)). In the initial state, the metric is calculated as: $E = \frac{w(t)}{w(0)} \log\left(\frac{\alpha_0(0)}{\alpha_0(t)}\right) = 1 \cdot 0 = 0, \quad when \quad t=0$ indicating that the metric automatically evaluates to zero at the beginning of training. We hope this explanation resolves any concerns regarding the metric's value in the untrained state.
>
>
> ---
>
> >**Communication of methods 1:** Given that you use transformers with attention, I imagine the author analyse the effective interaction of model parts conditioned on specific dataset. Naturally the attention weights are going to change how different parts of the models interact and so it seems one could not analyse the structure of the attention weights in general but instead would look at the effective communication when processing specific samples? Or do authors actually just use the weights? It would be great if this could be clarified, ideally at the start of the methods.
>
> **A:** We appreciate your insightful question about analyzing attention mechanisms. In self-attention layers, while the interaction between Q and K matrices generates dynamic attention weights that reflect input-dependent relationships between embeddings, our structural analysis deliberately focuses on the static weight matrices (W~Q~, W~K~, and W~V~) rather than these dynamic attention weights.
>
> The rationale behind this choice is that dynamic attention weights vary with input data, making them less suitable for characterizing the inherent properties of a trained model. To investigate the model's **intrinsic structural characteristics**, we utilize only the static weights that represent the model's fundamental learned parameters.
>
> We acknowledge the reviewer's constructive suggestion about analyzing dynamic attention patterns. In future research, we plan to explore how input-dependent dynamic attention weights might provide additional insights into the model's overall characteristics, complementing our current structural analysis approach.

---

> ### Author Response · Authors · 2024-11-26
> **Cont.**
>
> >**Communication of methods 2:** I think I generally got the metric development, but I did find it somewhat hard to follow. I wonder whether this could be improved if every new part of the method introduced would start with a sentence or two on the high-level intuition behind the measure / what the current analyses step is for – take section 3.2 for example. It starts with “To perform multifractal analysis of LLMs, we extend the box-covering and box-growing methods (Song et al., 2005; Evertsz & Mandelbrot, 1992; Salat et al., 2017; Xiao et al., 2021) to capture the local neuron-based fractal properties of NINs.”, followed by plenty of details on methods. If you lead into this section by first giving the reader an idea why it is interesting to decompose the graph into fractals in the first place, then maybe this would be easier to follow. Generally speaking, motivate the method before jumping into details. I know that the paper is already on the page limit but frankly the appendix is so long, moving a bit more of the methods to appendix in lieu of giving readers a better intuition in the main text is probably beneficial.
>
> >**Communication of methods 3:** Given there already is a bunch of structural measures which have successfully been used for structural analyses in neuroscience and neural network models (e.g. standard ones like modularity / small worldness, or entropy-based ones or Ricci curvature) I wonder whether authors could somehow motivate why their new measure was needed? Given it works, I do not want to say that the new measure was not needed but I would be curious to know why they specifically took the route they took with the development of the metric.
>
> **A:** Thank you very much for your valuable suggestion! We completely agree on the importance of providing **motivation** and **intuitive samples** of our methods. Following your suggestions, we have added motivation and high-level intuition for the properties in the revised version.  Please see **General Response Answer A.2**.
>
> ---
>
> >**Stylistic choices 1:** I think the font on Figure 6 is very close to unreadable small.
>
> >**Stylistic choices 2:** Across figures, could you either use a serif or sans serif font but not mix that up?
>
> **A:** Please see General Response answer A.1.
>
> ---
>
> >**Stylistic choices 3:** I wonder whether your strong emphasize of the word ‘emergence’ is actually needed and helpful in communicating your results? In my experience it does put off some researchers and, as you make a link to neuroscience, the development of language and reasoning skills from a newborn to a teenager would simply be considered as part of ‘development’ and ‘learning’. So, I wonder whether you want to deemphasize the concept of emergence a bit here and just call it ‘the development of skills’?
>
> **A:** Thank you for raising this thoughtful question. We understand that the term "emergence" may carry different interpretations across disciplines, and we appreciate your suggestion to reconsider its emphasis. In our work, we use "emergence" as a specific term rooted in the study of complex systems and self-organization, which also serves as the foundation for our proposed metric. In response to your feedback, in the revised version, we have added the section  "Motivation: connecting self-organization to emergence", to more clearly explain our use of the term "emergence" and its relevance to the phenomena we study. We hope this clarification addresses any concerns about overemphasis while maintaining the scientific rigor of our discussion.
>
> **Update:** We sincerely appreciate your valuable suggestions and have made further adjustments to address this concern. Please refer to **General Response Answer A.3** for details.

---

> > ### Comment · Reviewer_oHwU · 2024-11-28
> >
> > Dear authors,
> >
> > thank you so much for your response to my review. I appreciate the structured comments you made to all reviewers' comments. I Thank you for addressing my minor comments. I also appreciate that you took the space to further motivate your metric development. While I am still not *fully onboard* with the strong emphasis on 'emergence' I take this as a scientific disagreement and appreciate that you adapted your wording somewhat to avoid confusion. I do, however, want to ask two clarification questions to the more major issues I raised initially:
> >
> > *(1) Link to other metrics*
> >
> > I appreciate that you now add a general motivation for your metric but I think that explanation somewhat rests on the assumption that 'self-organization' is inherently worth studying with a new metric and that it is not already captured by other metrics. What I would have hoped for was perhaps more an explanation for why the metric you propose captures something that inherently is different from prior network measures. You refer to 'General Response Answer A.2' in your answer to my review but I do not think this point is actually addressed there.
> >
> > *(2) Baseline calculation*
> >
> > I appreciate that the metric automatically evaluates to zero at the start of training but I do not think that is quite addressing my point about baseline measures. What I meant was that, if you had a network that was progressing through training epochs, but the weight changes where somehow fully random as the network is not actually learning anything, what would the curve of your metric look like over training then? Would that just be a line continuously on 0 or very close to 0? Essentially what I am asking for is 'what is the pure noise condition under which the value of your metric would not change over training?'. I think this is important to understand the metric's behaviour overall.
> >
> > I am sorry if there is a trivial answer to this question which I am just somehow overlooking -- or if this is covered in one of your appendix analyses. I do not think that Figure 5 quite addresses this point. As a side note, I find it unintuitive that 'Epoch 1' refers to the untrained network, perhaps it would be better to label it as either 'Epoch 0' or specifically as 'Untrained' in the future.
> >
> > *Side note on static vs dynamic weight matrices*
> >
> > Thank you for clarifying which matrices you use for your analyses. I do think that makes sense (though also agree with you that it would be interesting to look at the dynamic ones in the future). I am however somewhat surprised that after me pointing this misunderstanding out to you, that I do not think you clarified this point in your manuscript -- or perhaps I am overlooking that change? I think other readers might have the same 'task active' expectation at first and just having a quick sentence at the start of the experimental or methods section could easily avoid readers getting confused over it.
> >
> >
> > Thank you again for engaging with my review!

---

> > > ### Author Response · Authors · 2024-11-28
> > > **Further Discussion**
> > >
> > > Thank you very much for your insightful and detailed feedback. We sincerely appreciate the time and effort you have dedicated to reviewing our paper and rebuttal and providing constructive comments. Your suggestions have been immensely helpful in enhancing the clarity of our paper.  We also recognize that certain aspects require more **thorough and precise discussion**, and we value this opportunity to clarify and address your remaining concerns and questions. Below, we provide our responses to your comments and questions questions point by point.
> > >
> > > ---
> > >
> > > > **C1:** While I am still not fully onboard with the strong emphasis on 'emergence' I take this as a scientific disagreement and appreciate that you adapted your wording somewhat to avoid confusion.
> > >
> > >
> > > **A:** We greatly appreciate your suggestions and acknowledge the potential ambiguities with the emphasis on “emergence”. As mentioned in **General Response Answer A.3**, we have made modifications accordingly. Specifically, following your suggestions, in the revised version, we have replaced "emergence" with "development of skills/emergent abilities" when describing the phenomenon of new capabilities arising in large models. For structure-related emergence inspired by "self-organization and emergence in natural systems," we have retained the term "degree of emergence." We highly value your feedback and are more than willing to consider any additional suggestions you may have to further avoid confusion.
> > >
> > > ---
> > >
> > > > **Communication of methods 3:** Given there already is a bunch of structural measures which have successfully been used for structural analyses in neuroscience and neural network models (e.g. standard ones like modularity / small worldness, or entropy-based ones or Ricci curvature) I wonder whether authors could somehow motivate why their new measure was needed? Given it works, I do not want to say that the new measure was not needed but I would be curious to know why they specifically took the route they took with the development of the metric.
> > >
> > > > **Link to other metrics:** I appreciate that you now add a general motivation for your metric but I think that explanation somewhat rests on the assumption that 'self-organization' is inherently worth studying with a new metric and that it is not already captured by other metrics. What I would have hoped for was perhaps more an explanation for why the metric you propose captures something that inherently is different from prior network measures. You refer to 'General Response Answer A.2' in your answer to my review but I do not think this point is actually addressed there.
> > >
> > > **A:** We sincerely appreciate your specific question regarding the rationale behind our choice of NeuroMFA over traditional network metrics. Previously, we primarily discussed the motivation connecting NeuroMFA to self-organization and emergence, but we recognize the need to more explicitly compare NeuroMFA with existing metrics. Below, we provide a detailed explanation and comparison, focusing on the unique challenges of bridging microstructural analysis in large models to their emergent macroscopic properties.
> > >
> > > ### **Key Challenges in Applying Network Analysis to Large Neural Models:**
> > >
> > >
> > > 1. **Unique Structural Characteristics of Neural Networks:** Neural networks exhibit **directed, weighted, and hierarchical** structures, which make traditional metrics designed for unweighted or undirected networks less applicable.
> > >
> > >
> > > 2. **Scalability to Large Models:** Large models have billions of parameters, creating networks of enormous size. Many traditional metrics become computationally infeasible for networks exceeding 10 million nodes or edges.
> > >
> > >
> > > 3. **Need for Multiscale Analysis:** Our objective is to bridge local (microscopic) structural changes in neural networks with their emergent macroscopic capabilities. This requires metrics capable of capturing both local interactions and global multiscale properties.
> > >
> > >
> > > Based on these points, we analyzed the potential applications of traditional network metrics in neuronal networks and neural networks, as well as their limitations when applied to **large-scale models**.
> > >
> > > ### **Comparison with Traditional Metrics:**
> > >
> > > 1. **Centrality Metrics：** Centrality measures such as degree, closeness, and betweenness centrality quantify the importance of nodes within a network.
> > >     *  **Potential Applications:** Useful for identifying key neurons in neuronal networks that are critical for information propagation or specific task processing.
> > >     * **Limitations:** Centrality metrics focus on **individual node-level properties**, making them less suitable for capturing global network dynamics or multiscale interactions.

---

> ### Author Response · Authors · 2024-11-28
> **Cont.**
>
> 2. **Small Worldness:** Small worldness quantifies the balance between local clustering and global efficiency using metrics such as local clustering coefficients and average shortest path lengths.
>     *  **Potential Applications:**  In neuronal networks, it helps understand the trade-off between local processing and global information integration, as well as network efficiency.
>     * **Limitations:** Primarily designed for undirected and unweighted networks, small worldness (especially for **the calculation of average shortest path**) is computationally expensive for large, weighted networks like those in LLMs.
>
> 3. **Modularity:** Modularity measures the degree to which a network can be divided into distinct modules or communities.
>     *  **Potential Applications:**   In neuronal networks, modularity can be used to identify functionally distinct sub-networks or quantify modular brain region connections.
>     * **Limitations:** Modularity identifies large-scale structures but fails to capture **node-level interactions** or multiscale properties. Additionally, it was initially designed for undirected and unweighted networks. Although adaptations exist, they are often computationally expensive to analyze the directed and weighted structures in neural networks.
>
>
> 4. **Entropy-Based Metrics:** Entropy metrics quantify the diversity or randomness of network structures, often focusing on degree distributions or edge weights.
>     *  **Potential Applications:** Useful for measuring the diversity of synaptic connections or activity patterns in neuronal networks.
>     * **Limitations:**  Entropy metrics analyze **distributions** rather than interactions, making them unsuitable for multiscale analysis. Additionally, entropy calculations for large, directed, and weighted networks are resource-intensive.
>
>
> 5. **Ricci Curvature:** Ricci curvature provides geometric insights into connectivity and robustness by measuring edge-based properties.
>     *  **Potential Applications:**  Applied in neural networks to identify critical pathways for signal transmission or bottlenecks in artificial neural networks.
>     *  **Limitations:** Ricci curvature is computationally prohibitive for large-scale, weighted networks and focuses on **edge properties**, offering limited insights into global network structure.
>
> ### **Why NeuroMFA?**
>
> Compared to the metrics above, NeuroMFA offers several distinct advantages:
>
> 1. **Node-Based Design:** NeuroMFA analyzes nodes directly, making it well-suited for studying local interactions and scaling up to **multiscale analyses**.
>
> 2. **Efficiency:** In NeuroMFA, the analysis starts from **node-based local properties** and leverages multiscale characteristics to derive the overall network's multifractal features. This method, which relies on local properties rather than direct computation over the entire network, significantly improves efficiency. Additionally, by leveraging scaling laws, NeuroMFA allows for network sampling without compromising the accuracy of fractal dimension estimation (see Appendix A.1.3, revised version, page 19), making NeuroMFA **computationally efficient**.
>
> 3. **Multiscale Analysis:** NeuroMFA captures both local and global properties, enabling us to capture the **heterogeneity** (diversity of local interactions) and **irregularity** (global structure) of the network structure across multiple scales, aligning with natural systems' self-organization and emergence rationale.
>
> In summary, although traditional metrics can be useful for analyzing smaller neural networks, they are often limited by computational inefficiency or lack of the ability of multiscale analysis when applied to large-scale neural networks. NeuroMFA overcomes these challenges by providing an efficient, scalable, and multiscale framework tailored to the unique characteristics of large neural models. As a practice example, we attempted to apply **Ollivier Ricci Curvature** to a **1.4B LLM**, but the process took over a week before the server crashed. In contrast, NeuroMFA completes the analysis of the same 1.4B model in approximately **45 minutes**.
>
> We truly appreciate your insightful question, as it allows us to clarify these distinctions and highlight why NeuroMFA is particularly suited for analyzing emergent abilities in large-scale models. We will also include this discussion in the final version.
>
> **Update:** We are actively working on the additional experiments to address your other question and will provide updates as soon as possible. Thank you very much for your understanding.

---

> ### Author Response · Authors · 2024-11-29
> **Cont.**
>
> > **Baseline calculation:** This is my only comment about a possible analysis to add. As you are proposing a new measurement, I think it would be helpful to provide measurements in untrained networks. Of course, Figure 8 shows the effect of training on the metric, but I wonder whether it actually would be zero across all models at the start of training? If it isn’t, should be somehow standardise the metric so that it is zero before training? If this was explore in more detail I would be happy to raise my ‘soundness’ rating to ‘excellent’.
>
>
>
>
> > **Baseline calculation:** I appreciate that the metric automatically evaluates to zero at the start of training but I do not think that is quite addressing my point about baseline measures. What I meant was that, if you had a network that was progressing through training epochs, but the weight changes where somehow fully random as the network is not actually learning anything, what would the curve of your metric look like over training then? Would that just be a line continuously on 0 or very close to 0? Essentially what I am asking for is 'what is the pure noise condition under which the value of your metric would not change over training?'. I think this is important to understand the metric's behaviour overall.
> I am sorry if there is a trivial answer to this question which I am just somehow overlooking -- or if this is covered in one of your appendix analyses. I do not think that Figure 5 quite addresses this point. As a side note, I find it unintuitive that 'Epoch 1' refers to the untrained network, perhaps it would be better to label it as either 'Epoch 0' or specifically as 'Untrained' in the future.
>
>
> **A:** Thank you for your thoughtful and detailed questions regarding **baseline calculations**. We realize the importance of discussing this topic, especially regarding the **stability of random initialized models (networks)** in capturing a reliable baseline. We appreciate the opportunity to address these points, as they are critical to demonstrating the robustness and applicability of our methodology. Below, we provide a comprehensive response, including theoretical insights and results from additional experiments.
>
>
> ## **Theoretical Insights**
> In large random networks with a fixed size and generation mechanism, the connection patterns among nodes are **random and homogeneous**, unlike the scale-free networks that introduce uneven distributions in node characteristics. This results in low variability in structural properties, with larger networks exhibiting even greater stability. In smaller random networks, random fluctuations can lead to localized imbalances, such as clusters with higher connectivity. However, in larger random networks, these local variations are averaged out, and the overall properties of the network converge statistically. For instance, as shown in Figure 2c of [1], which uses the **Watts-Strogatz model** to transition from ordered to small-world to random networks (with 1,000 nodes and repeated 100 times), the standard deviation of the multifractal spectra for random networks is significantly lower compared to small-world networks. Given that large language models often contain millions to billions of nodes, their random configurations are expected to exhibit even greater stability due to the statistical convergence of their properties.
>
>
>
>
> ## **Additional Experiments**
>
>
> To further explore the baseline calculation of large models, we conducted two experiments investigating random initialization and training from a pre-trained model. (i) Experiment 1: Following your suggestions, Experiment 1 evaluates the metric’s behavior when the weights of the LLMs undergo **fully random changes**, achieved by re-initializing the weights of the LLM. (ii) Experiment 2: Regarding the baseline, we recognized a **special scenario** where we train a **pre-trained model**, and its performance declines during training due to external factors. In such cases, the baseline for computing the degree of emergence should ideally align with **untrained initialized model**.
>
> We utilized the Hugging Face **pre-trained BERT-large-uncased model** (24 layers, 336M parameters, 1024 hidden dimensions, 16 attention heads) [[Source](https://huggingface.co/google-bert/bert-large-uncased)] [2] and its versions with randomly re-initialized weights.
>
>
> [1] Xiao X, Chen H, Bogdan P. Deciphering the generating rules and functionalities of complex networks. Scientific reports. 2021 Nov 25;11(1):22964.
>
> [2] Kenton JD, Toutanova LK. Bert: Pre-training of deep bidirectional transformers for language understanding. InProceedings of naacL-HLT 2019 Jun 2 (Vol. 1, p. 2).

---

> > ### Author Response · Authors · 2024-11-29
> > **Cont.**
> >
> > ### **Experiment 1: Baseline Analysis with Random Initialization**
> >
> >
> > **Experimental Setup:**
> >
> >
> > We generated 10 random neuron interaction networks (NINs) by **re-initializing the weights** of the BERT-large-uncased model with random values sampled from a truncated normal distribution, a standardized method in the BERT framework [2]. These NINs, denoted as r1–r10, were analyzed using the NeuroMFA metric to compute their degree of emergence. Among them, **r1 was selected as the baseline** for comparison.
> >
> >
> > **Results:**
> >
> >
> > The spectra of the NINs are shown in Figure 1 [Multifractal Spectrum](https://anonymous.4open.science/r/Neuron_LLM-4DC8/Graphs/bert-large-uncased_NMFA.png),  demonstrating that all randomly initialized networks exhibited **consistent multifractal properties**. The results of the degree of emergence are shown in Figure 2 [Degree of Emergence](https://anonymous.4open.science/r/Neuron_LLM-4DC8/Graphs/bert-large-uncased_emergence.png) and detailed in Table 1 below. Across the 10 random NINs, the degree of emergence remained **consistently close to zero** (less than 0.007).
> >
> >
> > | NINs              | r1 | r2 |r3 |r4 | r5 | r6 | r7 | r8 | r9 | r10 |
> > |:---------------------|:-------:|:-------:|:-------:|:-------:|:-------:|:-------:|:-------:|:-------:|:-------:|:-------:|
> > | Degree of Emergence  |0 | 0.00653  | 0.00119  | 0.00291  | 0.00585  | 0.00037 |0.00564  |0.00156 |0.00355  |0.00690  |
> >
> > Table 1. Degree of Emergence under Random Weight Changes.
> >
> >
> >
> >
> > This experiment further confirms that in large models with fixed model parameters and consistent architectural configurations, the multifractal properties remain stable under random weight changes. We appreciate your insightful comments and recognize the potential for **establishing standardized benchmarks** specifically for widely adopted LLMs, which would facilitate broader adoption and application of our method within the research community.
> >
> > ---
> >
> > ### **Experiment 2: Performance Degradation with Contaminated Data**
> >
> >
> > In this experiment, we trained the pre-trained BERT-large-uncased model using a **severely contaminated dataset** to observe how the metric behaves under conditions of performance degradation.
> >
> >
> > **Dataset Construction:**
> >
> >
> > To simulate performance degradation, we created a deliberately contaminated dataset:
> >
> >
> > * Retaining 20% of the original Wikipedia dumps dataset.
> >
> >
> > * Replacing the remaining 80% with random data generated from a normal distribution.
> >
> >
> > **Evaluation Framework:**
> >
> >
> > * **Structural Metric**: we assessed the model's degree of emergence throughout training using NeuroMFA, with the randomly initialized **NIN r1** (introduced in Experiment 1) serving as the baseline.
> >
> >
> > * **Performance Metrics**: model performance was evaluated using two standard metrics: (i) SQUAD 1.1 F1 score for reading comprehension; (ii) Multi NLI Accuracy for natural language inference.
> >
> >
> > **Results:**
> >
> >
> > As shown in Figure [Decreasing Performance](https://anonymous.4open.science/r/Neuron_LLM-4DC8/Graphs/Decreasing_Performance.png), the structure-based metric captured this pattern of performance degradation, decreasing from 0.6 to less than 0.1. This experiment demonstrates that when a large model experiences performance degradation due to severely contaminated datasets, its behavior becomes increasingly random, and the ordered structure of its network is also disrupted. This finding emphasizes the intrinsic connection between a model's emergent capabilities and its network structure.
> >
> > ---
> >
> >
> > **Baseline Label Adjustment:** Thank you for your suggestion regarding the labeling of the baseline. We agree that labeling the baseline as "Epoch 1" could indeed lead to some misunderstanding. We sincerely appreciate your insightful comments and will update all spectral results to label the baseline as "Untrained" for clarity. We will ensure that this adjustment is reflected in the final version of the manuscript.

---

> ### Author Response · Authors · 2024-11-29
> **Cont.**
>
> > **Side note on static vs dynamic weight matrices.** Thank you for clarifying which matrices you use for your analyses. I do think that makes sense (though also agree with you that it would be interesting to look at the dynamic ones in the future). I am however somewhat surprised that after me pointing this misunderstanding out to you, that I do not think you clarified this point in your manuscript -- or perhaps I am overlooking that change? I think other readers might have the same 'task active' expectation at first and just having a quick sentence at the start of the experimental or methods section could easily avoid readers getting confused over it.
>
>
> **A:** Thank you for your valuable suggestion. We have clarified this in Appendix B (Implementation Details, Section B.5) of the revised version (page 30). We value your suggestion and will ensure that this clarification is also included at the beginning of the **Experiments section** in the final version to avoid any potential confusion for readers.
>
> ---
>
> Thank you again for your thorough review, insightful questions, and constructive suggestions! We deeply appreciate your support and the time and effort you have dedicated to helping us improve our manuscript. Once again, we greatly value your feedback and suggestions and welcome any further discussion.

---

> ### Comment · Reviewer_oHwU · 2024-12-02
>
> Dear authors,
>
> thank you so much for your extensive work here -- this is very helpful information and clears up any questions I had. I really hope that this paper will be included in the proceedings. Note that I updated my confidence rating and soundness rating in response to these additional clarifications and experiments.

---

> ### Author Response · Authors · 2024-12-02
> **Thank you for your feedback with the increased confidence score!**
>
> Dear Reviewer oHwU,
>
> Thank you for your kind and encouraging feedback, and especially thanks for **increasing your confidence to 5** and the **soundness to 4** to vote for our work! We are truly delighted to hear that your concerns have been addressed. We appreciate your thorough review and thoughtful comments, which have helped further improve our work. We hope our work can reach a broader audience and contribute valuable insights to the community.
>
> Best,
>
> Authors

---

### Official Review · Reviewer_2tKM · 2024-11-03

**Soundness:** 3
**Presentation:** 3
**Contribution:** 3
**Rating:** 6
**Confidence:** 2

**Summary:**

This paper proposes a new metric to capture emergence within a network by analyzing the structure of neurons within the network. The score depends on the width and peak of the multifractal spectrum overall quantifying the variance in the local structure of the network. This metric is shown to correlate with metrics depending on the behavior of the model, which have traditionally used to measure emergence.

**Strengths:**

1. Writing is clear, explains the new metric clearly.
2. Moving the discussion away from only scaling model size is important
3. The metric is well defined, making it replicable.

**Weaknesses:**

1. Text in Fig 1 is very small making it hard to read and there is a lot of white space that can be used to increase text size.
2. "While our NeuroMFA framework demonstrates correlations between the emergence metric and
downstream performance by studying the self-organization of LLMs, it has not yet established a
clear causal relationship. Future work should focus on identifying specific linguistic phenomena
captured by LLMs and demonstrating how our analysis methods can reveal the learning process of
these phenomena during training, thereby establishing a stronger causal connection." This should be in the main paper.
3. While figure 6 shows the correlation between this metric and previous ones (based on the behavior), I find little reason to use this metric over the behavior (encoded by the previous metrics) of the model itself. At best this metric is correlated with those behavioral methods, at worst they disagree where the behavior would still be more convincing.
4. We see no reason why this property isn't just linked to a further amount of training. For example, if there is a dataset that when an LLM is trained on it, there is a low performance on the behavior based metrics would we still expect to see the proposed metric increase?
5. In figure 6 as the accuracy goes up, the proposed metric increases (across models). This is no longer always true in Appendix D.2. For example in fig 19, the 410M model has lower performance but the proposed metric is higher than both the 1B and 1.4B models. This makes sense as the score is normalized with initial training but then you cannot compare across models. This is possible for all of the traditional behavioral metrics.
6. Fig 8 shows that the scores are lower in the beginning of training than the end for all metrics, and that the model size is only recovered at the later end. How does this address the problem in Fig 1b? As the other metrics have a similar low score after 500 epochs do they also not only rely on the model size?

Overall it is unclear to me the benefit that this metric provides, especially when the paper itself acknowledges that the metric is only validated through its correlation.

**Questions:**

1. What are the behaviors for Pythia 14m and 31m fig 6? Do they also agree with Lambada/PIQA?
2. Are there examples where the performance on behavioral metrics decreases significantly and the proposed metric also decreases?

---

> ### Author Response · Authors · 2024-11-26
> **Thanks for your review and insightful comments.**
>
> We thank the reviewer for the insightful feedback. We hope our detailed response addresses your concerns and highlights the framework’s motivation and rationale. We answer to your comments and questions step-by-step below.
>
> > **W1:** Text in Fig 1 is very small making it hard to read and there is a lot of white space that can be used to increase text size.
>
> **A:** Please see **General Response Answer A.1**.
>
> ---
>
> > **W2:** "While our NeuroMFA framework demonstrates correlations between the emergence metric and downstream performance by studying the self-organization of LLMs, it has not yet established a clear causal relationship. Future work should focus on identifying specific linguistic phenomena captured by LLMs and demonstrating how our analysis methods can reveal the learning process of these phenomena during training, thereby establishing a stronger causal connection." This should be in the main paper.
>
> **A:** Thank you for the suggestion. In response to this concern, we have included the limitations in the revised version (page 10).

---

> ### Author Response · Authors · 2024-11-26
> **Cont.**
>
> > **W3:** While figure 6 shows the correlation between this metric and previous ones (based on the behavior), I find little reason to use this metric over the behavior (encoded by the previous metrics) of the model itself. At best this metric is correlated with those behavioral methods, at worst they disagree where the behavior would still be more convincing.
>
> **A:** We acknowledge your concerns regarding comparing behavior-based metrics and our proposed structural metric. While behavior-based metrics offer insights into a model's external performance, they are inherently limited in the following ways as we write in the paper (page 1): “(i) observation-based methods often require exhaustively testing various outputs, making it challenging
> to establish a unified standard for evaluating the full scope of a large model’s emergence; (ii) the
> approaches treat the model as a black box, limiting the understanding of the internal mechanisms
> that drive the model’s behavior and emergent abilities; (iii) relying solely on model size to measure
> performance overlooks other critical factors that significantly influence the model’s capabilities,
> such as architecture design, data quality and training dynamics.” Moreover, we would like to give more examples here:
>
> 1. **Model Illusions and Deceptive Behaviors:** Large language models (LLMs) can exhibit illusions in their behavior. For instance, models often produce outputs aligned with training biases but fail in edge cases or when transferred to new tasks. These limitations underscore the need for metrics that analyze internal structures rather than relying solely on external behavior.
>
> 2. **From "What" to "Why" in Understanding LLMs:** LLMs are often criticized as "black boxes" due to the difficulty of understanding the origins of certain outputs or behaviors. To deploy LLMs in high-stakes scenarios, it is crucial to understand not only what models can do but also why they exhibit certain behaviors. Behavior-based metrics alone are insufficient to build this level of trust. Structural analysis aims to enhance interpretability by linking structural properties to behaviors. This connection provides a framework to explain the origins of complex capabilities, making model behavior more predictable and trustworthy.
>
> We aim to address the critical limitations of behavior-based metrics by developing methods that quantify **internal structures** and link them to observable behaviors. This effort seeks to move beyond simply evaluating what models can do, to understanding why they behave in specific ways. NeuroMFA is our attempt to bridge this gap by providing a framework for structural analysis, paving the way for broader applications and deeper insights. Based on this work, we are actively exploring ongoing projects and envision multiple future directions, including but not limited to:
>
> 1. **Comparing architectures and training strategies:** The NeuroMFA framework enables researchers to analyze and compare the self-organization processes across different LLM architectures and training setups. By examining metrics like α0 and w, one can gain insights into which designs facilitate the emergence of complex patterns and potentially identify factors that promote the development of advanced capabilities.
>
> 2. **Guiding model development:** The insights from NeuroMFA could inform the design of future LLMs. For example, if higher values of α0 and w consistently correlate with better performance, researchers could explore architectures or training techniques that encourage these properties. The graph-based analysis thus provides a new set of metrics to optimize and guide the development process.
>
> 3. **Fundamental understanding of emergent intelligence:** More broadly, the NeuroMFA framework contributes to the ongoing research into the fundamental principles underlying the emergence of intelligence in artificial systems. By drawing parallels to self-organization in natural systems and characterizing the complex dynamics of neuron interactions, this work offers a fresh perspective on how sophisticated behaviors arise from simple components. It opens up new questions and directions for investigating the nature of emergent intelligence.
>
> We believe this work is only the beginning of a broader effort to understand and quantify the structural dynamics underlying emergent behaviors in large-scale models. By addressing the key limitations of behavior-based metrics and providing a robust framework for structural analysis, we hope to inspire further research and collaborations in this field.

---

> ### Author Response · Authors · 2024-11-26
> **Cont.**
>
> > **W4:** We see no reason why this property isn't just linked to a further amount of training. For example, if there is a dataset that when an LLM is trained on it, there is a low performance on the behavior based metrics would we still expect to see the proposed metric increase?
>
> > **Q2:** Are there examples where the performance on behavioral metrics decreases significantly and the proposed metric also decreases?
>
> **A:** We appreciate the reviewer's concern about validating our proposed metric in models where **performance decreases** during training. To address this, we conducted experiments using BERT-large in both optimal and degraded training scenarios.
>
> **Methodology:**
>
> 1. **Model Selection:** We utilized a pre-trained BERT-large-uncased model from Hugging Face (24 layers, 336M parameters, 1024 hidden dimensions, 16 attention heads).
> 2. **Dataset Construction:** We created a deliberately contaminated dataset by:
>    * Preserving 20% of the original Wikipedia dumps dataset
>    * Replacing 80% with randomly generated data from a normal distribution
> 3. **Evaluation Framework:** We assessed model performance using two standard metrics:
>    * SQUAD 1.1 F1 score (reading comprehension)
>    * Multi NLI Accuracy (natural language inference)
>
> **Results:**
>
> As illustrated in **Figure** [Decreasing Performance](https://anonymous.4open.science/r/Neuron_LLM-4DC8/Graphs/Decreasing_Performance.png), the results revealed a strong correlation between our proposed metric and performance measures. When training with contaminated data, all metrics showed:
>
> * Sharp initial decline within the first few hundred epochs
> * Subsequent stabilization at reduced performance levels
>
> Our proposed metric effectively captures the pattern of model ability degradation caused by severe data contamination, highlighting its capability to track structural changes in the model under different training scenarios. Moreover, this consistency further validates that our structure-based metric can reflect the evolution of emergent properties in LLMs throughout the training process, regardless of training quality or the direction of change. This part has been added to the appendix C.3 of the revised version (page 34). For the motivation of the metrics, please see **General Response A.2.**

---

> ### Author Response · Authors · 2024-11-26
> **Cont.**
>
> > **W5:** In figure 6 as the accuracy goes up, the proposed metric increases (across models). This is no longer always true in Appendix D.2. For example in fig 19, the 410M model has lower performance but the proposed metric is higher than both the 1B and 1.4B models. This makes sense as the score is normalized with initial training but then you cannot compare across models. This is possible for all of the traditional behavioral metrics.
>
> **A:** Thank you for raising the question regarding the comparison of metrics across models. First, we would like to clarify that the degree of emergence for untrained models is initially 0. As training progresses, we estimate the structural degree of emergence for each model. Different model sizes exhibit varying speeds and extents of **structural self-organization**, making comparisons across models meaningful. In fact, we have demonstrated cross-model comparisons in Figure 8 (page 10), where it can be observed that the degree of emergence increases logarithmically with model size. Notably, there is indeed an inflection point at the **410M** model size. Interestingly, similar small inflection points are also observed on LAMBADA and ARC Challenge benchmarks. As shown in Figure 20 (revised version, page 39), the structural self-organization of the 410M model continues to increase throughout the training process, while other models appear to plateau once their metrics reach a threshold. We tend to believe this may indicate that structural self-organization undergoes significant enhancement when the model size falls within a specific range, making it a phenomenon worth further investigation in future work.
>
> ---
> > **W6:** Fig 8 shows that the scores are lower in the beginning of training than the end for all metrics, and that the model size is only recovered at the later end. How does this address the problem in Fig 1b? As the other metrics have a similar low score after 500 epochs do they also not only rely on the model size?
>
> **A:** Thank you for your question. First, it is important to clarify that 500 epochs represent an extremely **early stage** of training compared to the full 143,000 epochs. At this stage, as shown in Figure 2, the outputs from larger models are almost nonsensical, highlighting that the model remains in a near-randomized state. This aligns with the lower two points near the x-axis in Figure 1b (bottom graph), shown as two **yellow points** in this figure [Early Stage](https://anonymous.4open.science/r/Neuron_LLM-4DC8/Graphs/Illustration.png). These points correspond to the model’s early training phase when it has not yet developed meaningful structural organization. This observation underscores the fact that relying solely on Figure 1b (top graph) to evaluate emergent abilities is not comprehensive. It highlights the necessity of studying self-organization structures and their relationship with emergence from a training perspective.

---

> > ### Author Response · Authors · 2024-11-26
> > **Cont.**
> >
> > > **Q1:** What are the behaviors for Pythia 14m and 31m fig 6? Do they also agree with Lambada/PIQA?
> >
> > **A:** Thank you for the question. It is important to emphasize that our NeuroMFA analysis focuses on capturing structural changes (i.e., regularity and heterogeneity), and the proposed metric is specifically designed based on the principles of **self-organization**. These metrics must be analyzed based on the **self-organization phenomenon**. However, as we have indicated in the manuscript, we did not observe self-organization phenomena in the 14M and 31M models.
> >
> > As discussed in the paper (revised version, Figure 5, page 8), our multifractal analysis provides the following insight: “This indicates that, in the 14M model, the network’s **regularity does not increase**. Similarly, the 31M model displays a subtle leftward shift at the 5,000th epoch, but this trend does not continue significantly thereafter. In contrast, for larger models (exceeding 100M parameters), a consistent leftward shift of the network spectra is observed from the 0th to the 35,000th epoch. This shift suggests an increase in regularity and a corresponding enhancement in self-organization as training progresses."
> >
> > According to the principles and theories of self-organization and emergence,  emergent properties arise only when a system reaches sufficient complexity, with numerous components and interactions leading to the formation of ordered structures (**self-organization**) [1,2]. As a result, the 14M and 31M models do not exhibit self-organization phenomena during the training, making them unsuitable for the proposed metrics. Compared to larger models, they also exhibit inferior performance and are considered to **lack emergent capabilities** [3]. To address potential confusion, we have added the following clarification to the revised version of the manuscript: “Notably, since the degree of emergence is fundamentally tied to self-organization phenomena, we did not compute this metric for the 14M and 31M models in subsequent experiments.” (See revised version, page 8)
> >
> > While our proposed metric may not be applicable to smaller models lacking observable self-organization, we believe that NeuroMFA analysis remains valuable, offering insights into the structural dynamics of these models. It’s interesting that this finding also aligns with understandings from natural systems, where emergence occurs only when individual interactions are sufficiently complex and abundant [1].
> >
> > [1] Jeffrey Goldstein. Emergence as a construct: History and issues. Emergence, 1(1):49–72, 1999.
> >
> > [2] James P Crutchfield. The calculi of emergence: computation, dynamics and induction. Physica D: Nonlinear Phenomena, 75(1-3):11–54, 1994.
> >
> > [3] Biderman, Stella, et al. Pythia: A suite for analyzing large language models across training and scaling. International Conference on Machine Learning. PMLR, 2023.

---

> > > ### Comment · Reviewer_2tKM · 2024-11-27
> > > **Thanks for addressing comments**
> > >
> > > Overall, your additions have addressed a good amount of my concerns, especially the Bert training, which does give me more confidence about the submission. I am still concerned about the use of this metric when it is only correlated with behavioral metrics, and there are still some cases where it seems to diverge (Supp Fig 20-23). Especially in cases like fig 20 1B where the emergence has clearly plateaued and the accuracy continues to improve.
> > >
> > > # The use of a correlational metric
> > > > 1. **Comparing architectures and training strategies:** The NeuroMFA framework enables researchers to analyze and compare the self-organization processes across different LLM architectures and training setups. By examining metrics like α0 and w, one can gain insights into which designs facilitate the emergence of complex patterns and potentially identify factors that promote the development of advanced capabilities.
> > > 2. **Guiding model development:** The insights from NeuroMFA could inform the design of future LLMs. For example, if higher values of α0 and w consistently correlate with better performance, researchers could explore architectures or training techniques that encourage these properties. The graph-based analysis thus provides a new set of metrics to optimize and guide the development process.
> > > 3. **Fundamental understanding of emergent intelligence:** More broadly, the NeuroMFA framework contributes to the ongoing research into the fundamental principles underlying the emergence of intelligence in artificial systems. By drawing parallels to self-organization in natural systems and characterizing the complex dynamics of neuron interactions, this work offers a fresh perspective on how sophisticated behaviors arise from simple components. It opens up new questions and directions for investigating the nature of emergent intelligence.
> > >
> > > I agree these are all things the NeuroMFA framework **could** be used for these aspects but I don't see how any of these are fully answered with only a correlational metric. Why would I use this to guide model development instead of looking at benchmark performance? Maybe it can be used as a first check **if** NeuroMFA is faster than the benchmarks but I don't think this is the case (could be wrong here).
> > >
> > > I think this is a good contribution to understanding how training changes structures within a network. But, I also feel that this does not answer the question of how large models are (from the abstract):
> > >
> > > >handling complex tasks that small-scale models are unable to perform
> > >
> > > without an explicit link of the structures NeuroMFA describes, to the complex tasks that the larger models can complete I am unsure of its usefulness.
> > >
> > > # Emergence or something else?
> > > Maybe the problem is still the use of the word emergence that both other reviewers have taken issue with as well:
> > >
> > > >I disagree with the article’s use of “emergence,” although I understand the rationale. You state that your measure is intended to quantify network regularity. The main conclusion of the article should be that this measure appears sufficient as a proxy for the emergent properties of LLMs.
> > >
> > > >I wonder whether your strong emphasize of the word ‘emergence’ is actually needed and helpful in communicating your results? In my experience it does put off some researchers and, as you make a link to neuroscience, the development of language and reasoning skills from a newborn to a teenager would simply be considered as part of ‘development’ and ‘learning’. So, I wonder whether you want to deemphasize the concept of emergence a bit here and just call it ‘the development of skills’?
> > >
> > > I do think the added section does address some of these concerns but if you again go back to the definition in the abstract of large models "handling complex tasks that small-scale models are unable to perform" at best it is a proxy.
> > >
> > > # Final Thoughts
> > > I do think the paper has strayed a little from my expertise so I might be missing aspects of the usage. Therefore I will increase my score from a 3 to a 5 (as I do like the changes but I still think this doesn't answer the question posed) but also reduce my confidence to a 2. I will defer to the other reviewers who may be more experienced with this topic.

---

> ### Author Response · Authors · 2024-11-28
> **Thank you for your prompt feedback and raising the score.**
>
> We sincerely appreciate your thorough review and are encouraged to hear that the rebuttal, including the **BERT training experiments**, has addressed a good amount of your concerns and provided you with more confidence about the submission. We deeply value the time and effort you have invested in providing constructive feedback, which has greatly helped us improve our manuscript. We also welcome this opportunity to address your remaining concerns and have provided a one-to-one response below. We look forward to any further discussions.

---

> > ### Author Response · Authors · 2024-11-28
> > **Cont.**
> >
> > > **C1:**  I am still concerned about the use of this metric when it is only correlated with behavioral metrics, and there are still some cases where it seems to diverge (Supp Fig 20-23). Especially in cases like fig 20 1B where the emergence has clearly plateaued and the accuracy continues to improve.
> >
> > **A:** Thank you for your thoughtful question. We would like to first highlight that evaluating the emergent abilities of LLMs requires a multifaceted approach involving multiple benchmarks to study scaling laws and related phenomena comprehensively [1]. We do not expect the performance metrics to exhibit consistent variation. We are particularly focused on the **emergent abilities** exhibited by large models, which are absent in smaller models [1]. This phenomenon aligns with the "emergence" observed in natural systems. By studying the structural self-organization of large models, we propose a **structural metric** and compare it against several established benchmarks that evaluate emergent capabilities, such as high-level comprehension and scientific reasoning (see Figure 7, revised version, page 9), showing the correlations between these benchmarks and our proposed metric.
> >
> > Regarding your specific observation in Figure 20 (1B model), where the emergence metric plateaus while accuracy continues to improve, we believe this reflects the complex relationship between structural self-organization and task performance. The plateau in the emergence metric suggests that the structural self-organization of the model has reached saturation, indicating that the structure has formed a stable and orderly pattern. However, task performance may continue to improve due to fine-tuning and optimization for specific benchmarks, which does not necessarily require further changes in the overall underlying structure. This highlights the need to use structural metrics in conjunction with behavioral metrics to gain a comprehensive understanding of LLM emergent abilities.
> >
> > **Intuitive Analogy:** To provide a more straightforward and intuitive analogy, purely for the sake of an engaging discussion beyond the scope of strict scientific rigor, consider how we evaluate human intelligence (IQ) or brain development. Traditionally, intelligence might be assessed through various tests targeting different dimensions, such as logical reasoning or language skills. These tests, akin to behavioral metrics in LLMs, are task-specific and offer valuable insights into particular intelligent abilities. Now, imagine if we could also obtain precise data about the brain's structure and neural signal transmission. In such a scenario, researchers might aim to develop novel analytical methods that measure the organization, connectivity patterns, and dynamics of neuronal activity in real-time, thereby providing a completely different perspective on intelligence and brain functionality.
> >
> > Similarly, our work aims to take **a first step** toward studying the structure and internal dynamics of large models, providing a theoretical framework supplemented by initial empirical evidence that suggests practical implications.
> >
> > [1] Wei, Jason, et al. Emergent abilities of large language models. Transactions on Machine Learning Research, 2022a.
> >
> >
> > ---
> >
> >
> > > **C2:** I agree these are all things the NeuroMFA framework could be used for these aspects but I don't see how any of these are fully answered with only a correlational metric. Why would I use this to guide model development instead of looking at benchmark performance? Maybe it can be used as a first check if NeuroMFA is faster than the benchmarks but I don't think this is the case (could be wrong here).
> >
> > **A:** Thank you for raising this specific question. Regarding the **guidance of model development**, we do not view NeuroMFA merely as a preliminary diagnostic tool but as a theoretical foundation for incorporating higher structural heterogeneity and regularity into the design and training of large models. For example, we could **construct loss functions** that explicitly encourage the increase of structural heterogeneity and regularity during training. Similarly, in **pruning operations**, nodes with homogeneous fractal properties (identified by the identical or similar fractal dimension) could be prioritized for removal to enhance the overall heterogeneity of interactions. These are just two examples, and we believe NeuroMFA can inspire a broader range of strategies and applications, making it a promising foundation for further exploration and innovation in future research.

---

> ### Author Response · Authors · 2024-11-28
> **Cont.**
>
> > **C3:** I think this is a good contribution to understanding how training changes structures within a network. But, I also feel that this does not answer the question of how large models are (from the abstract): handling complex tasks that small-scale models are unable to perform without an explicit link of the structures NeuroMFA describes, to the complex tasks that the larger models can complete I am unsure of its usefulness.
>
> **A:** We appreciate your insightful comments. While our future work aims to establish a causal link between specific network structures and task-specific capabilities, we believe our analysis already provides novel and valuable insights into understanding how large model scale supports **the formation of self-organized structures**. For example, our NeuroMFA method reveals significant structural dynamic differences regarding the multifractal spectra between models of varying scales, particularly distinguishing between small models (e.g., tens of MB) and larger models (e.g., hundreds of MB and above) (please refer to Section 4.1 page 8), indicating that achieving a certain scale is a prerequisite for forming self-organized structures. This observation aligns with existing research on scaling language models, which demonstrates that model performance remains near random below a certain scale but significantly improves to well above random levels once this threshold is exceeded, reflecting the **emergent capabilities** in larger models [1] [2]. By capturing the multifractal **structural dynamic differences** across models of varying scales, our NeuroMFA method provides a structural analysis perspective for understanding the gap between small and large models.
>
> [1] Wei, Jason, et al. Emergent abilities of large language models. Transactions on Machine Learning Research, 2022a.
>
> [2] Biderman, Stella, et al. Pythia: A suite for analyzing large language models across training and scaling. International Conference on Machine Learning, 2023.
>
> ---
>
>
> > **C4:** Emergence or something else? I do think the added section does address some of these concerns but if you again go back to the definition in the abstract of large models "handling complex tasks that small-scale models are unable to perform" at best it is a proxy.
>
> We sincerely appreciate your thoughtful feedback on the usage of the term “**emergence**” in our manuscript. Upon further reflection and considering your comments and those of other reviewers, we recognize that we did not sufficiently distinguish between the definitions of “emergence” in complex systems and LLMs, potentially causing ambiguity for readers. In our work, the metric is inspired by the principles and definitions of self-organization and emergence in natural complex systems, where emergence represents the **new properties** appearing due to structural self-organization. However, in the LLM domain, “emergence” often merely refers to “**the appearance of emergent abilities**,” which may lead to misunderstandings.
>
> We are grateful for your insight and have made the following adjustments in the revised manuscript to address this issue:
>
> 1. **Abstract adjustment:** We have revised the abstract from “a novel structure-based metric for evaluating the degree of emergence in large models” to “a novel structure-based metric as a proxy for emergent abilities of large models.”
>
> 2. **Clarification of the "degree of emergence" definition:** We have clarified that the "degree of emergence" in our method refers specifically to emergence following structural self-organization. This clarification has been added where the term  "degree of emergence" first appears (see revised version, page 3), explicitly stating that:“ degree of emergence (this is a structure-based metric that can be viewed as a proxy for the emergent abilities of large models)”.
>
> 3. **Terminology modification:** To reduce ambiguity, we have unified the references to “emergence” associated with large model capabilities, replacing them with “**emergent abilities**” or “**development of skills**” as the reviewer oHwU suggested.
>
> We believe these adjustments will mitigate potential confusion while maintaining the scientific rigor of the manuscript. Once again, thank you for your valuable input, which has significantly helped improve the clarity and presentation of our work.

---

> > ### Comment · Reviewer_2tKM · 2024-11-30
> >
> > Thanks for making those further changes.
> >
> > >**A:** Thank you for raising this specific question. Regarding the guidance of model development, we do not view NeuroMFA merely as a preliminary diagnostic tool but as a theoretical foundation for incorporating higher structural heterogeneity and regularity into the design and training of large models. For example, we could construct loss functions that explicitly encourage the increase of structural heterogeneity and regularity during training. Similarly, in pruning operations, nodes with homogeneous fractal properties (identified by the identical or similar fractal dimension) could be prioritized for removal to enhance the overall heterogeneity of interactions. These are just two examples, and we believe NeuroMFA can inspire a broader range of strategies and applications, making it a promising foundation for further exploration and innovation in future research.
> >
> > I like the ideas suggested here and hope to see future work on the effect of using the metric in these ways during the training process.
> >
> > Combining that with the adjustments and clarifications to the abstract and some of the terminology used I will increase my score one more time. Thanks for the engagement!

---

> ### Author Response · Authors · 2024-11-30
> **Thank you for your thorough review and further increasing the score!**
>
> We sincerely thank you for recognizing our efforts in addressing your concerns and for increasing your score following our further discussions and clarifications. Your thoughtful review and constructive feedback have been invaluable in improving our work.
>
> We are particularly encouraged by your recognition of our ideas, especially the potential of NeuroMFA in guiding model development and expanding its applications in LLM research. This aligns closely with our vision of contributing **valuable insights** and **practical methods** to the community.
>
> We also appreciate your positive feedback on the adjustments made to enhance the clarity and readability of our manuscript. Your thoughtful engagement throughout this review process has been instrumental in refining our work.

---

### Author Response · Authors · 2024-11-26
**General Response 1**

We would like to thank all the reviewers for their time, thoughtful comments, and suggestions for our paper. We are profoundly encouraged by the fact that the reviewers have found our work **interesting**, **novel** and **well-written**. We have conducted additional experiments and revised the manuscript in response to the reviewers' questions and insightful suggestions. While these experiments and improvements took more time than we initially anticipated, they have significantly enhanced the readability and clarity of the paper. In the revised version, all changes have been marked in blue. We are looking forward to a fruitful discussion and greatly appreciate the reviewers' responses.

Here, we summarized common questions and comments below. For other comments, we have provided **one-to-one responses** to each reviewer.

**Q.1. Improvements to Figures:** [Reviewer **2tKM** and **oHwU**]

> Reviewer 2tKM: Text in Fig 1 is very small making it hard to read and there is a lot of white space that can be used to increase text size.

> Reviewer oHwU:  I think the font on Figure 6 is very close to unreadable small.

> Reviewer oHwU: Across figures, could you either use a serif or sans serif font but not mix that up?

**A.1:** We sincerely thank Reviewer **2tKM** and Reviewer **oHwU** for their careful reading and feedback regarding the readability of the figures. Reviewer 2tKM pointed out that the text in Figure 1 was too small and there was excessive white space, while Reviewer oHwU noted that the font in Figure 6 was very small and suggested using a consistent font style across all figures. In response, we have increased the text size in Figures 1 and 6, reduced unnecessary white space and ensured consistent use of font across all figures. These changes have been implemented in the revised version of the manuscript.

---

> ### Author Response · Authors · 2024-11-26
> **General Response 2**
>
> **Q.2. Motivation and illustration of the proposed metrics:** [Reviewer **2tKM**, **oHwU**, and **VqLs**]
> > Reviewer 2tKM: We see no reason why this property isn't just linked to a further amount of training. For example, if there is a dataset that when an LLM is trained on it, there is a low performance on the behavior-based metrics would we still expect to see the proposed metric increase?
>
> > Reviewer oHwU: I think I generally got the metric development, but I did find it somewhat hard to follow. I wonder whether this could be improved if every new part of the method introduced would start with a sentence or two on the high-level intuition behind the measure / what the current analyses step is for – take section 3.2 for example. It starts with “To perform multifractal analysis of LLMs, we extend the box-covering and box-growing methods (Song et al., 2005; Evertsz & Mandelbrot, 1992; Salat et al., 2017; Xiao et al., 2021) to capture the local neuron-based fractal properties of NINs.”, followed by plenty of details on methods. If you lead into this section by first giving the reader an idea why it is interesting to decompose the graph into fractals in the first place, then maybe this would be easier to follow. Generally speaking, motivate the method before jumping into details. I know that the paper is already on the page limit but frankly the appendix is so long, moving a bit more of the methods to appendix in lieu of giving readers a better intuition in the main text is probably beneficial.
>
>
> > Reviewer oHwU: Given there already are a bunch of structural measures which have successfully been used for structural analyses in neuroscience and neural network models (e.g. standard ones like modularity / small worldness, or entropy-based ones or Ricci curvature) I wonder whether authors could somehow motivate why their new measure was needed? Given it works, I do not want to say that the new measure was not needed but I would be curious to know why they specifically took the route they took with the development of the metric.
>
>
> > Reviewer VqLs: I feel like your "degree of emergence" is itself one dimensional as well, which makes it way less interesting as if it was not. It would be a bit outrageous if we were able such a complex reorganisation with only one global metric. It would much more interesting if you were able to link certain network properties with specific emerging properties.
>
>
> > Reviewer VqLs: Emergence is, by definition, a property observable at a higher scale that is not explainable by its components. You are measuring network properties, which is interesting, but is it truly the appropriate scale to discuss emergence? I am not convinced the term “emergence” is wisely used in this article. Instead, the focus should be on providing a quantification of network structure that enables or correlates with emergent properties
>
>
> **A.2:** We sincerely thank the reviewers for their insightful feedback regarding the need to better explain the theoretical foundation, motivation, and intuition behind our proposed metrics. We recognize that the reviewers 2tKM and VqLs’ misunderstandings of our metrics may stem from our initial manuscript’s lack of clarity in articulating the motivation and intuition before introducing the methods. Taking this into account, and based on the reviewers’ feedback, we have added a new section to the revised version, which discusses the motivation for our metrics at the beginning of the **Methodology** section (see **Section 3.1 page 4**). This section is grounded in the definitions of **self-organization** and **emergence** and explains how these principles guide the design of our metrics.
>
> First, it is essential to clarify that our metrics are not simply general network analysis tools but are specifically designed based on and directly tied to the phenomena of self-organization and emergence. As described in Supplementary A.3.3 (revised version, page 26), we draw an **analogy** between self-organization and emergence in natural systems and their counterparts in large models. The core design of our metrics is inspired by two critical aspects of self-organization and emergence:
>
> 1. **Development of new interaction patterns**: Captures the increasing diversity and complexity of local interactions among components.
> 2. **Formation of orderly structures**: Reflects the system’s evolution towards global order and structured interactions.

---

> ### Author Response · Authors · 2024-11-26
> **General Response 3**
>
> When self-organization reaches critical thresholds in these two aspects, emergent capabilities are observed. To measure these two aspects of self-organization, we employ **neuron-based multifractal analysis (NeuroMFA)**. This framework provides a unified approach to analyzing:
>
> - **Heterogeneity**: Quantified through the spectral width of the multifractal spectrum, representing the diversity of local interaction patterns (corresponding to the development of new interaction patterns).
>
> - **Regularity**: Quantified through the peak position of the multifractal spectrum, reflecting the degree of global order in the network (corresponding to the formation of orderly structures).
>
> By **simultaneously capturing** heterogeneity and regularity, NeuroMFA offers a comprehensive tool for analyzing self-organization in neuron interaction networks (NINs). Combined with **information-theoretic** principles, these observations of self-organization enable us to compute emergence quantitatively [1]. Detailed explanations of these metrics are provided in Supplementary A.2.1–A.2.3.
>
> To address these points, we have added the following explanation in the revised manuscript:
> "Self-organization is a foundational concept for understanding emergent phenomena in natural systems. It describes how systems gradually evolve through local interactions to develop **more diverse interaction patterns** and **achieve overall order**, ultimately leading to emergence. This process involves two key aspects: the development of new interaction patterns (reflecting increased **heterogeneity** in interactions) and the formation of orderly structures (representing enhanced **regularity** in global structure). Emergence, in this context, refers to the novel macro-level properties formed during the process of self-organization—characteristics that arise through collective interactions and cannot be fully explained by the dynamics of individual components. Inspired by these principles, we extend the concept of self-organization to study emergence in large artificial models  (please refer to Supplementary A.3.3) and design network-based metrics to quantify the two key aspects of self-organization: **heterogeneity** (capturing the diversity of neuron interactions) and **regularity** (reflecting the order in global network structure). By introducing neuron-based multifractal analysis (NeuroMFA), we provide a unified framework to investigate these dimensions, enabling a quantitative analysis of the relationship between structure and emergent capabilities in LLMs."
>
> We are deeply grateful for the reviewers’ constructive suggestions, which prompted us to improve the clarity and accessibility of our manuscript. In the revised version, we have expanded the motivation section, reorganized the methodology chapter, and added explanations to address the reviewers’ concerns.
>
> [1] Prokopenko, Mikhail, Fabio Boschetti, and Alex J. Ryan. "An information‐theoretic primer on complexity, self‐organization, and emergence." Complexity 15.1 (2009): 11-28.

---

> ### Author Response · Authors · 2024-11-26
> **General Response 4**
>
> To further illustrate, according to the reviewers’ suggestions to provide intuitive samples, we have incorporated the use of toy examples to demonstrate our emergent calculations, especially the heterogeneity metric $w$ and regularity metric $\alpha_0$. Below, we offer detailed specifics of our experimental design with the results showcased in this **Figure** [NeuroMFA Metrics](https://anonymous.4open.science/r/Neuron_LLM-4DC8/Graphs/NeuroMFA%20Metrics.png) :
>
> **Exp 1: Increasing Heterogeneity** (From a homogeneous network to a network with a higher degree of heterogeneity). The **homogeneous network**, as seen in fig. **(a)** (comprising 3 layers with 32 nodes each), is initialized, such that all nodes and their interactions are identical, resulting in every node exhibiting the same fractal properties. Consequently, the neuroMFA spectrum of this homogeneous network is represented by **a single point** in fig. **(g)**, indicating **a single interaction pattern**. Subsequently, we transform the network by randomly selecting a quarter of the nodes and replacing them with nodes that possess two other distinct fractal properties, creating the network shown in fig. **(b)**. Due to the **increased heterogeneity** of the network, the **spectral width broadens**. We then further modify the network by selecting a third of the remaining homogeneous nodes at random, and replacing them with nodes of two different fractal properties, leading to the network depicted in fig. **(c)**. As shown in fig. **(g)**, the network's heterogeneity increases even more, resulting in a wider spectral width. Through this simple example, we aim to demonstrate how the heterogeneity metric in the spectrum can infer the formation of new interaction patterns by measuring the heterogeneity of the Neuron Interaction Network (NIN) through NeuroMFA.
>
> **Exp 2: Increasing regularity** (From a random network to a network with a higher degree of regularity). We initiate this phase of the experiment by generating a **completely random network**. Subsequently, we adopted the principle of **introducing regularity** into an unweighted network, employing the **Preferential Attachment Mechanism** (PAM), and simply extended this concept to our weighted directed hierarchical network to progressively enhance its regularity. The PAM describes a process of creating a scale-free network characterized by the emergence of hubs or highly connected nodes. When applied to NIN, the weighted PAM (wPAM) (see Algorithm 1) involves introducing specific rules, such as strengthening strong connections and weakening the weaker ones, to regularize interactions.
>
> Starting with a randomly generated network (fig. **(d)**), we then implement the wPAM rules (as described in Algorithm 1), iterating N=5 times to obtain network **(e)**. Following another 5 iterations, we produce network **(f)**. Observing the spectrum shift to the left and a decrease in $\alpha_0$ indicates an increase in the network structure's regularity, as depicted in fig. **(h)**. This progression illustrates the variation of the regularity of the dynamic network can be captured by the regularity metric $\alpha_0$ through NeuroMFA.
>
>
> ---
> **Algorithm 1:** Weighted Preferential Attachment Mechanism (wPAM) for NIN
>
> 1. Initialize graph $G$ with a total of 128 nodes and denote the set of nodes as $V = \{v_1, v_2, \ldots, v_{128}\}$.
>
> 2. Divide $V$ into 4 layers: $L_1 = \{v_1, \ldots, v_{32}\}$, $L_2 = \{v_{33}, \ldots, v_{64}\}$, $L_3 = \{v_{65}, \ldots, v_{96}\}$, $L_4 = \{v_{97}, \ldots, v_{128}\}$. Each node in $L_i$ is fully connected to all nodes in $L_{i+1}$.
>
> 3. Initialize edge weights between nodes using $W_{ij} = \text{random.uniform}(A, B)$ for each edge $(v_{i}, v_{j})$, where $W_{ij}$ represents the weight of the edge from node $v_{i}$ to node $v_{j}$ in the next layer.
>
> 4. Compute the median weight of all edge weights in $G$ as $median(W)$.
>
> 5. Define decrease factor $\delta < 1$ to selectively reduce smaller weights and increase factor $\iota = 1 / \delta$ to selectively increase larger weights.
>
> 6. Set the number of iterations $N$ for multiple rounds of weight adjustment.
>
>     For $n = 1$ to $N$ do
>
>     $\quad$ For $i = 1$ to $128$ do
>
>     $\quad$$\quad$ For $j = 1$ to $32$ do
>
>     $\quad$$\quad$$\quad$ If $W_{ij} < median(W)$ then
>
>     $\quad$$\quad$$\quad$$\quad$ $W_{ij} \gets W_{ij} \cdot \delta$ $\quad$ // Decrease weight
>
>     $\quad$$\quad$$\quad$ Else
>
>     $\quad$$\quad$$\quad$$\quad$ $W_{ij} \gets W_{ij} \cdot \iota$ $\quad$ // Increase weight
>
>     $\quad$$\quad$$\quad$ EndIf
>
>     $\quad$$\quad$ EndFor
>
>     $\quad$ EndFor
>
>     EndFor
> ---
> We have included this in the revised version of the manuscript. By incorporating these updates, we believe the manuscript now presents a clearer and more **intuitive understanding** of our proposed metrics and their connection to the **foundational principles** of self-organization and emergence.

---

> > ### Author Response · Authors · 2024-11-28
> > **General Response 5  - Further Discussion**
> >
> > **Q.3. The use of term “Emergence”:** [Reviewer **2tKM**, **oHwU**, and **VqLs**]
> >
> > > Reviewer 2tKM: Emergence or something else? I do think the added section does address some of these concerns but if you again go back to the definition in the abstract of large models "handling complex tasks that small-scale models are unable to perform" at best it is a proxy.
> >
> > > Reviewer oHwU:  I wonder whether your strong emphasize of the word ‘emergence’ is actually needed and helpful in communicating your results? In my experience it does put off some researchers and, as you make a link to neuroscience, the development of language and reasoning skills from a newborn to a teenager would simply be considered as part of ‘development’ and ‘learning’. So, I wonder whether you want to deemphasize the concept of emergence a bit here and just call it ‘the development of skills’?
> >
> > > Reviewer oHwU: I disagree with the article’s use of “emergence,” although I understand the rationale. You state that your measure is intended to quantify network regularity. The main conclusion of the article should be that this measure appears sufficient as a proxy for the emergent properties of LLMs.
> >
> >
> > **A.3:**  After further discussions with Reviewer 2tKM, we realized that the cross-disciplinary usage of the term “emergence” in our manuscript—drawing from both complex systems and LLM domains—might cause potential misunderstanding among readers. The definition of "emergence" in the context of complex systems and its usage in the LLM domain have subtle differences. In our work, the metric is inspired by the principles and definitions of **self-organization and emergence** in natural complex systems, where emergence represents the **new properties appearing** due to structural self-organization. However, in the LLM domain, “emergence” often merely refers to “**the appearance of emergent abilities**.” Therefore, the simultaneous use of "emergence" to refer to structural self-organization in complex systems and to the development of new capabilities in LLMs may lead to potential misunderstandings. To eliminate this ambiguity, and in alignment with the valuable suggestions from reviewers oHwU and VqLs, we have made the following adjustments in the revised manuscript to address this issue.
> >
> > 1. **Abstract adjustment:** We have revised the abstract from “a novel structure-based metric for evaluating the degree of emergence in large models” to “a novel structure-based metric as a proxy for emergent abilities of large models.”
> >
> > 2. **Clarification of the "degree of emergence" definition:** We have clarified that the "degree of emergence" in our method refers specifically to emergence following structural self-organization. This clarification has been added where the term  "degree of emergence" first appears (see revised version, page 3), explicitly stating that:“ degree of emergence (this is a structure-based metric that can be viewed as a proxy for the emergent abilities of large models)”.
> >
> > 3. **Terminology modification:** To further reduce ambiguity, we have unified the references to “emergence” associated with large model capabilities, replacing them with “**emergent abilities**” or “**development of skills**” as the reviewer oHwU suggested. For structural aspects, we continue to use "emergence" (e.g. “degree of emergence”) to align with the definitions in complex systems, where the term specifically describes the arising of novel, coherent properties or patterns resulting from structural self-organization.
> >
> > We believe these adjustments will mitigate potential confusion while maintaining the scientific rigor of the manuscript. We sincerely appreciate the efforts and insightful suggestions of the reviewers, which have greatly contributed to improving the clarity of our manuscript.

---

### Meta-Review · Area_Chair_Pxhk · 2024-12-23

**Metareview:**

This paper proposes a new metric and new analytical framework for the improvement in LLMs as a function of neural network structure, which captures scaling behavior better than raw parameter count.  The reviewers agreed that this paper addresses a timely and important problem, and were impressed by the paper's novelty, thoroughness, and rigor.   I'm pleased to report that it has been accepted to ICLR.  Congratulations!  Please revise the manuscript to address all reviewer comments and discussion points.

**Additional Comments On Reviewer Discussion:**

The reviewers asked for a number of clarifications and quantitive comparisons.  The authors provided a very thorough and compelling set of explanations and additional results, and in the end 2/3 reviewers were strongly in favor of acceptance. (One reviewer -- VqLs -- gave a 5 and felt that the paper needed a bit more work, but the enthusiasm of the other two convinced me that it ought to be accepted).

---

### Decision · Program_Chairs · 2025-01-22

Accept (Poster)